# Ocean carbonate system variability in the North Atlantic Subpolar surface water (1993-2017)

**Coraline Leseurre[1], Claire Lo Monaco[1], Gilles Reverdin[1], Nicolas Metzl[1], Jonathan Fin[1], Solveig Olafsdottir[2] and Virginie Racapé[3]**

[1]Laboratoire d'Océanographie et du Climat: Expérimentation et Approches Numériques (LOCEAN-IPSL), Sorbonne Université-CNRS-IRD-MNHN, Paris, 75005, France
[2] Marine and Freshwater Research Institute (MFRI), Reykjavik, Iceland
[3]Laboratoire d'Océanographie Physique et Spatiale (LOPS), CNRS-IFREMER-IRD-UBO, Plouzané, 29280, France

*Correspondence to*: Coraline Leseurre (coraline.leseurre@locean-ipsl.upmc.fr)

**Abstract.** The North Atlantic is one of the major ocean sinks for natural and anthropogenic atmospheric $CO_2$. Given the variability of the circulation, convective processes or warming/cooling recognized in the high latitudes in this region, a better understanding of the $CO_2$ sink temporal variability and associated acidification needs a close inspection of seasonal, interannual to multidecadal observations. In this study, we investigate the evolution of $CO_2$ uptake and ocean acidification in the North Atlantic Subpolar Gyre (50°N-64°N) using repeated observations collected over the last three decades in the framework of the long-term monitoring program SURATLANT (SURveillance de l'ATLANTique). Over the full period (1993-2017) pH decreases (-0.0017 yr$^{-1}$) and fugacity of $CO_2$ (f$CO_2$) increases (+1.70 µatm yr$^{-1}$). The trend of f$CO_2$ in surface water is slightly less than the atmospheric rate (+1.96 µatm yr$^{-1}$). This is mainly due to dissolved inorganic carbon (DIC) increase associated with the anthropogenic signal. However, over shorter periods (4-10 years) and depending on the season we detect significant variability investigated in more detail in this study. Data obtained between 1993 and 1997 suggest a rapid increase in f$CO_2$ in summer (up to +14 µatm yr$^{-1}$) that was driven by a significant warming and an increase in DIC for a short period. Similar f$CO_2$ trends are observed between 2001 and 2007 during both summer and winter but, without significant warming detected, these trends are mainly explained by an increase in DIC and a decrease in alkalinity. This also leads to a pH decrease but with contrasting trends depending on the region and season (between -0.006 yr$^{-1}$ and -0.013 yr$^{-1}$). On the opposite, data obtained during the last decade (2008-2017) in summer show a cooling of surface waters and an increase in alkalinity, leading to a strong decrease of surface f$CO_2$ (between -4.4 and -2.3 µatm yr$^{-1}$; i.e. the ocean $CO_2$ sink increases). Surprisingly, during summer, pH increases up to +0.0052 yr$^{-1}$ in the southern subpolar gyre. Overall, our results show that, in addition to the accumulation of anthropogenic $CO_2$, the temporal changes of the uptake of $CO_2$ and ocean acidification in the North Atlantic Subpolar Gyre present significant multiannual variability, not clearly directly associated with the North Atlantic Oscillation (NAO). With such variability it is uncertain to predict the near-future evolution of air-sea $CO_2$ fluxes and pH in this region. Thus, it is highly recommended to maintain long-term observations to monitor these properties in the next decade.

## 1 Introduction

The ocean plays an important role in climate regulation by absorbing between one quarter and one third of anthropogenic carbon dioxide ($CO_2$) emitted to the atmosphere (Le Quéré et al., 2018; Gruber et al., 2019). The North Atlantic (NA) is one of the strongest ocean sinks for natural (Takahashi et al., 2009) and anthropogenic atmospheric $CO_2$ (Khatiwala et al., 2013; Sabine et al., 2004; Perez et al, 2008). Although many observational studies were conducted in the NA to evaluate how the $CO_2$ sink varies from seasonal to multi-decadal scales and modeling studies attempted to reproduce the observed changes, there are still open questions regarding the processes that control the contemporary $CO_2$ sink variability in this region (e.g. Schuster et al., 2013, for a synthesis). The uptake of $CO_2$ in the NA is mainly due to extensive biological activity during spring-summer and considerable heat loss during winter, both processes being subject to significant spatio-temporal variability related to climate mode such as the North Atlantic Oscillation (NAO) and the Atlantic Multidecadal Variability (AMV), also called Atlantic Multidecadal

Oscillation (AMO). The recent decades have indeed witnessed large variations of the NAO index (Hurrell index, 2013) which displayed strong positive values in the early 1990s, then from 1996 to the early 2000 tended to be positive, and has then been more neutral, albeit with extreme anomalies, such as the negative 2010 event, until again witnessing large positive anomalies in the mid-2010s. The NA has also being at the core of large decadal to multi-decadal variability of the AMV with a positive peak by the mid-2000s and a decrease since then (Robson et al., 2018).

In the North Atlantic Subpolar Gyre (NASPG, around 50-60°N) where the oceanic $CO_2$ sink is particularly strong (e.g. Watson et al., 2009), recent years in the mid-2010s have also witnessed particularly large negative surface temperature anomalies (Josey et al., 2018) and an extreme freshening (Holliday et al., 2020). Associated changes in the ocean circulation and water masses in the NASPG are for example presented in Chafik et al. (2014, 2019); Desbruyères et al. (2015); Nigam et al. (2018). These physical processes linked to NAO and/or AMV directly impact the sea surface $fCO_2$ and air-sea $CO_2$ fluxes through warming/cooling or deeper convection as this has been observed in the NASPG for specific

periods (Corbière et al., 2007; Fröb et al., 2019). However, a direct link of the $CO_2$ uptake variability with the NAO depends on the period investigated and this is not always clearly revealed from observations (Metzl et al., 2010; Schuster et al., 2013).

Ocean models and Earth System Models (ESM) can help to understand the link between NAO and biogeochemical cycles, but results from models in the NA are still controversial (Thomas et al., 2008; Ullman et al., 2009; Keller et al., 2012; Tjiputra et al., 2012). Keller et al. (2012)

investigated simulations from 6 ESMs and found that the inter-annual variability of the $CO_2$ sink in the North Atlantic is the largest in the subpolar gyre. They also conclude that for winter (i.e. not the productive season) on-site entrainment in the NASPG (mixing and upwelling) is the main driver of carbon sink variability as opposed to advection (Thomas et al., 2008), but the magnitude and responses of the carbon uptake to the NAO significantly differ between the ESM models (Keller et al., 2012). Thomas et al. (2008) suggest that negative or neutral NAO conditions result in a substantial decline of the $CO_2$ uptake for the years 1997-2004 along the North Atlantic Current region (NAC) and in the

eastern subpolar gyre. On the opposite Ullman et al. (2009) conclude that the $CO_2$ uptake increased over 1992–2006. During the transition of NAO (from positive to neutral), their model simulates a decline of the convection and vertical DIC supply to the surface in the subpolar region, counteracting the increase of $pCO_2$ due to warming. This leads to a relatively small $pCO_2$ increase compared with the atmospheric $CO_2$ trend and an increasing $CO_2$ sink, while observations in 1993-2003 suggest a reduced $CO_2$ sink in the NASPG after the NAO shift in the mid-90s (Corbière et al., 2007).

For longer timescales, two decades or more, observations in the NASPG show a gradual $fCO_2$ increase slightly smaller or indistinguishable from trends in atmospheric $CO_2$ and that the long-term oceanic $fCO_2$ trend is not correlated with NAO (Takahashi et al., 2009; McKinley et al., 2011; Fay and McKinley, 2013). On decadal scale the link between AMO and $fCO_2$ variability in the NASPG appears more robust (Breeden and McKinley, 2016; Landschützer et al., 2019). When AMO enters in a positive phase these studies indicate a reduction of DIC in the NASPG due

to reduced mixing, a process that dominates the effect of warming on $fCO_2$ (Breeden and McKinley, 2016). Consequently, since the mid-1990s the $fCO_2$ long-term trend lagged the one in the atmosphere and the $CO_2$ uptake is increasing in this region. Such signal is not captured by current ESM CMIP5 models (Tjiputra et al., 2014; Lebehot et al., 2019) likely due to inadequate representation of biogeochemical cycles (here DIC and/or TA) and in part to bad representation of mixing leading to $pCO_2$ seasonality that is opposed to observations (Goris et al., 2018). This leads to uncertainties on the evolution of the oceanic $CO_2$ uptake in the North Atlantic in the future (Lebehot et al., 2019). A better knowledge

of DIC and TA trends (not only $fCO_2$) is needed to correct and validate biogeochemical representation in ESM 35 models.

The responses of the marine biological processes to NAO and AMO are much less studied than their physical counterpart. However, and somehow, as for $fCO_2$, the biology link seems also better identified with AMO than with NAO. Based on the biomass variability detected from SeaWIFS over 1998-2007 in the North Atlantic and supported by biogeochemical simulations McKinley et al. (2018) conclude that "nowhere is the NAO correlated with biomass variability". The same is true, i.e. no correlation with NAO, when analyzing long-term in-situ observations

of marine species in the NA since the 60s (Beaugrand et al., 2013; Rivero-Calle et al., 2015). However, these authors detect significant increase of calcifying species (coccolithophores, foraminifera) associated to the warming in the NA since the mid-90s and correlated with AMO. This increase in calcification would impact the DIC and TA concentrations in summer and thus the $fCO_2$ trend and carbon uptake in the NA. On the

other hand, this also suggests that these phytoplanktonic groups are not yet altered by acidification, a phenomenon known as the other $CO_2$ problem (Doney et al., 2009). Indeed, the accumulation of anthropogenic $CO_2$ in the ocean has led to a decrease in pH in surface waters by 0.1 unit since the industrial revolution (IPCC, 2013 - Hartmann et al., 2013). It is projected that surface pH would be reduced by 0.2 to 0.4 unit by the end of the century depending on the anthropogenic emission scenario (Orr et al., 2005; Doney et al., 2014; Jiang et al., 2019). This would be associated with a dramatic reduction of the aragonite saturation horizon at high latitudes, from 2000 m at present up to 100 m by 2100 in the North Atlantic (north of 50°N). Indeed, these northern surface waters (north of 50°N) have initially low $CaCO_3$ saturation because of high natural (preindustrial) DIC concentrations due to low temperatures giving high $CO_2$ solubility. Such fast lysocline shoaling based on a simulation (Orr et al., 2005) is started to be observed in this region (e.g. Vázquez-Rodríguez et al., 2012). Due to the potential threat on marine life, the decrease in pH is now recognized as a true indicator of global change, similarly to warming and sea level rise (World Meteorological Organization, 2018). The urgent need to document and understand changes in oceanic pH and its impact on marine life has motivated more studies in recent years. Among them, the data syntheses from Lauvset et al. (2015) and Bates et al. (2014) indicated that pH is decreasing in most of the surface ocean as a result of the increase in oceanic $CO_2$. In the North Atlantic they reported a decrease of pH ranging between -0.001 $yr^{-1}$ in the subtropical region and -0.0026 $yr^{-1}$ in the Irminger Sea. These results have been confirmed at a regional scale. Based on 12 cruises conducted in summer between 1991 and 2015 in the Irminger and Iceland basins, García-Ibáñez et al. (2016) identified a pH decrease in all water masses, with largest change in surface and intermediate waters of between -0.0016 to -0.0018 unit $yr^{-1}$ and conclude that it is mainly due to anthropogenic $CO_2$ and modulated by change in alkalinity. Recently Omar et al. (2019) analyzed regular surface observations during 2004-2015 in the North Sea; when selecting winter data in the North Atlantic surface waters they found significant long-term trends of $fCO_2$ and pH of +2.4 µatm $yr^{-1}$ and -0.0024 $yr^{-1}$, consistent with results obtained in the subpolar North Atlantic (Bates et al., 2014; Lauvset and Gruber, 2014). A driver analysis indicates that $fCO_2$ and pH trends are almost entirely explained by increasing DIC due to anthropogenic $CO_2$ uptake (Omar et al., 2019).

As anthropogenic $CO_2$ accumulates each year in the surface and interior ocean with direct impact on both air-sea $CO_2$ exchange and pH, it is important to conduct and maintain regular observations to follow the evolution of the ocean carbonate system properties and interpret how they vary from inter-annual to multi-decadal scales in relation with physical and biological changes. The aim of this study is to document the recent evolution of the surface $CO_2$ system parameters in the NASPG and to evaluate the drivers for the evolution of surface $fCO_2$ and pH for winter and summer. We mostly use Total Alkalinity (TA) and Dissolved Inorganic Carbon (DIC) discrete observations, while other studies are based on observations of partial pressure of $CO_2$ (pCO2) and TA/Salinity relations (Lauvset and Gruber, 2014; Lauvset et al., 2015). The observations were collected in 1993-2017 between Iceland and Newfoundland and this extends previous analyses based on winter SURATLANT observations in 1993-2008 to both summer and winter time (Corbière et al., 2007; Metzl et al., 2010).

The NASPG as defined here extends from the east coast of Canada (65°W) to the west of Northern Europe (5°W) mostly north of 45°N (Figure 1). It consists mostly of the Labrador, Irminger and Iceland basins, and the Rockall Plateau and Trough. The NASPG is a region of mixing between subtropical, subpolar and polar surface waters. The inflow of these surface waters is dominated by three surface currents: The North Atlantic Current (NAC), also known in this sector as the North Atlantic Drift, mostly of subtropical origin, the East Greenland Current (EGC) and the Labrador Current (LC) which are the conduits that bring waters from further north and the Arctic (Fig. 1). The North Atlantic Drift delimits the NASPG in the south and transports warm (8°C to 15°C) and salty (35 to 36) subtropical surface waters to the north. These waters are then cooled as they circulate in the NASPG due to a loss of heat to the atmosphere, while they become less salty due to excess local precipitation and freshwater inputs from the Arctic and the ice sheet. The East Greenland Current (EGC) and Labrador Current (LC) are the western branches of the NASPG and carry cold (<4°C) and fresh (<34.6) waters of polar origin. The study area is mostly in the central part of the NASPG between Iceland and Newfoundland.

## 2 Material and Methods

### 2.1 Data collection and measurements

This study is based on observations collected in the framework of the long-term monitoring program SURATLANT (SURveillance de l'ATLANTique) initiated in 1993. The main objective of this program is to monitor hydrological and biogeochemical properties in surface waters of the NASPG, notably Sea Surface Salinity (SSS) to improve the understanding of the role of salinity on the variability and predictability of climate as well as the water cycle (Reverdin et al., 2018a; Holliday et al., 2020). To this end, two to four cruises per year are conducted between Reykjavik (Iceland) and Newfoundland (Canada) on board merchant ships. Seawater samples are collected every three to four hours from a pumping system (at approximately 5m deep) in order to measure salinity, as well as alkalinity (measured since 2001), DIC, silicate, nitrate and phosphate concentrations.

Underway measurements of Sea Surface Temperature (SST) are obtained using a Seabird Thermosalinograph, with a precision of ±0.1°C. SSS values in this paper are from sample measurements of conductivity using a Guidline AUTOSAL salinometer done since 1997 at the Marine and Freshwater Research Institute (MFRI) in Reykjavik (the associated error is estimated at ±0.005). For TA and DIC, the 500 ml glass bottles are rinsed three times before introducing seawater (avoiding introducing air bubbles) with an overflow in order to remove the water in contact with air during filling. The samples are then poisoned with mercuric chloride and stored in a cool, dark place. Since 2001, these samples are measured within three months of collection at the SNAPO-$CO_2$ (Service National d'Analyse des Paramètres Océaniques du $CO_2$) located at the Laboratory LOCEAN in Paris. The accuracy is on the order of ±3 µmol kg$^{-1}$ for both DIC and TA based on CRMs analysis and occasional intercomparisons as explained by Reverdin et al. (2018b). Samples for the analysis of nutrients concentrations are frozen just after sampling. Spring or summer samples are filtered before analysis. The measurements are carried out by the MFRI team in Reykjavik according to the standard colorimetric method described by Ólafsson et al. (2010) with an accuracy of ±0.2 µmol l$^{-1}$ for nitrate and silicate and ±0.03 µmol l$^{-1}$ for phosphate. Details of the sampling and analyses for all properties are provided in Reverdin et al. (2018b). The SURATLANT dataset is freely available and is accessible at http://www.seanoe.org/data/00434/54517/. In the SURATALANT data file, each sample and data is associated with a quality flag (based on WOCE flag criteria, i.e. same as used in GLODAP data-base, Olsen et al. (2016). In this study we use only data qualified with flag 2 (acceptable data).

### 2.2 Calculations of the carbonate system parameters and contributions

Carbonate system parameters such as pH, the fugacity of $CO_2$ (f$CO_2$) and the saturation states for the calcium carbonate minerals calcite ($\Omega_{Ca}$) and aragonite ($\Omega_{Ar}$) are calculated using SST, SSS, TA, DIC, silicate and phosphate data described by Reverdin et al. (2018b). For the latter two, we used monthly climatological values derived from the SURATLANT data over the period 1993-2017 because nutrients inter-annual variability does not have a significant impact on the carbonate system parameters calculations (and because nutrients were missing for some cruises). The calculation program used is CO2SYS originally developed by Lewis et al. (1998) available in MATLAB version (van Heuven et al., 2011) that now includes error propagation (Orr et al., 2018). The constants of the thermodynamic equilibrium of $CO_2$ in seawater used are: $K_1$ (for the dissociation of carbonic acid) and $K_2$ (for the bicarbonate ion) defined by Mehrbach et al. (1973), refitted by Dickson and Millero (1987). The total boron value is calculated according to Uppström (1974) and the $KHSO_4$ dissociation constant is from Dickson (1990). The adopted pH scale is total scale.

When TA was not measured (notably before 2001), it was calculated from salinity data. The correlation between sea surface alkalinity and salinity in the open ocean can be described with an empirical linear relationship (Millero et al., 1998; Friis et al., 2003). For the NASPG, the original TA/SSS relation used to analyze f$CO_2$ variability and trend in 1993-2007 (e.g. Corbière et al., 2007; Metzl et al., 2010; McKinley et al., 2011) was revisited by Reverdin et al. (2018b) based on SURATLANT data over the period 2001-2016 (Eq. 1). This equation was obtained for samples collected north of 50°N with SSS > 34.

$$TA = 45.5337 \times SSS + 713.58,\tag{1}$$

To evaluate the air-sea $CO_2$ difference (delta-$fCO_2$) for each sample we used atmospheric $fCO_2$ values for the period 1993-2017 calculated from the molar fraction ($xCO_2$) data at Mace Head provided by the Cooperative Global Atmospheric Data Integration Project (Dlugokencky et al., 2018). The $xCO_2$ data are available at http://www.esrl.noaa.gov/gmd/dv/iadv/ last access 06/03/2019). $xCO_2$ data were converted to $fCO_2$ at 100% humidity following Weiss and Price (1980), with standard atmospheric pressure (i.e. 1013.25 hPa).

The trends in sea surface pH, $fCO_2$, and $\Omega$ are driven by changes in SST, SSS, TA and DIC (Keeling et al., 2004; Fröb et al., 2019). For DIC, this includes anthropogenic $CO_2$ concentration (Cant) and the DIC natural component (DIC-nat = DIC-Cant). The contribution of each term for a specific period and season is evaluated by allowing a change in only one parameter according to their observed trend, while setting the other parameters to their climatological seasonal values (Eq. 2). The uncertainty on the contributions was evaluated by performing 1000 random perturbations within the range of the standard deviation of the observed trends in SST, SSS, TA and DIC. The same method was applied when separating the effect of DIC-nat and anthropogenic $CO_2$ (Cant).

$$\frac{dX}{dt} = \frac{\partial X}{\partial SST}\frac{dSST}{dt} + \frac{\partial X}{\partial SSS}\frac{dSSS}{dt} + \frac{\partial X}{\partial DIC}\left(\frac{\overline{sDIC}}{SSS_0}\frac{dSSS}{dt} + \frac{\overline{SSS}}{SSS_0}\frac{dsDIC}{dt}\right) + \frac{\partial X}{\partial TA}\left(\frac{\overline{sTA}}{SSS_0}\frac{dSSS}{dt} + \frac{\overline{SSS}}{SSS_0}\frac{dsTA}{dt}\right)$$

$$\tag{2}$$

Here, X corresponds to pH, $fCO_2$, and $\Omega$. $SSS_0$ is the reference salinity, which is set to 35 (Normal Standard Seawater, Millero et al., 2008) and very close to the mean salinity observed in the NASPG (SSS = 34.84). sDIC and sTA were computed by normalizing DIC and TA to a salinity of 35 and assuming a nonzero freshwater end-member DIC and TA concentration (Friis et al., 2003; Reverdin et al., 2018b) . The trends (dX/dt) were estimated for each season, each boxes and periods; and $\overline{SST}, \overline{SSS}, \overline{TA}, \overline{DIC}$ correspond to their climatological values calculated over the same period.

The uncertainties of measured parameters (SST, SSS, DIC and TA; Fig. 4a to 4f) used in random calculations were based on the standard deviation associated with the averaging of the data in each box and month. For the calculated parameters (pH, $fCO_2$, $\Omega_{Ar}$; Fig. 4d to 4j), the uncertainties consider the errors associated with each measurement of the calculation parameters (Orr et al., 2018). The uncertainty on the trends is linked to the discrete sampling of spatio-temporal variability within the season that we wish to analyze. The uncertainty on the trends is linked either to first, the discrete sampling of a spatial-temporal variability within the season analyzed, or second, to the interannual variability (real or linked to insufficient sampling). If the uncertainty associated with the annual averages is low, then the trend uncertainty will be low and will correspond to the first part. On the other hand, in the case where the interannual (real) variability is high, the uncertainty will correspond to the second part which will decrease the trend significance.

## 2.3 Data selection: regions and seasons

The sampled region was separated into five 4° latitude boxes, from 46°N to 64°N (Fig. 1) according to Reverdin et al. (2018b). The southern box (box A: 46°N-50°N) covers the shelf and continental margin of North America and is excluded in the present long-term trend analysis because of insufficient sampling and large inter-annual variations due to fresh waters inputs from continental runoff. Box B (50°N-54°N) incorporates only samples where SSS is between 34 and 35 (to avoid including continental shelf or NAC waters). Thus, whereas over 470 samples with DIC data are found in Box-B for 1993-2017, only 399 data in the salinity range 34-35 are considered (we excluded 38 data for S<34 and 33 data for S>35). For boxes C (54°N-58°N), D (58°N-62°N) and E (62°N-64°N) we used all available data.

The trend analysis and contributions calculations (Eq. 2) was first performed for each box but results of trends and drivers for boxes C, D, E were found quasi identical. Therefore, we present and discuss the results by combining the data collected north of 54°N (i.e., boxes C, D and E). The trends in box B are evaluated separately because properties and their trends (SST, SSS, DIC, TA) differ significantly from the other three boxes. We separate seasons, and the trends for winter (January to March) and summer (June to August) are analyzed separately as they correspond to seasonal extrema of surface carbonate properties (described below in Section 3.1) and these seasons were more regularly sampled along the SURATLANT line (Table 1). Finally, we choose February and July as a reference for winter and summer, respectively. When February or July data were missing for a specific year, the data collected in January or March (for winter) and in June or August (for summer) were used after correcting the observed seasonal anomalies from the climatological cycles described in the next section (3.1). We tested this reconstruction for February and July data using SST and SSS data based on the Binned products from Reverdin et al 2018a and averaging the bins in the same Boxes (B and CDE) used for discrete sampling (see Supp. Figure S1 and Table S1). As expected, the differences for SST and SSS are most pronounced for summer and box B (Fig. S1 a,c) due to higher variability observed during this season, both in time and at small-scale. For winter the reconstruction based on discrete sampling appears more robust when compared to the Binned products, especially for box CDE (Fig. S1, b,c). To interpret our results, we thus are more confident with the trends based on winter data than summer and for box CDE. No attempt to reconstruct data was performed when the SURATLANT line was not sampled for at least 3 months (e.g. winter 2016 and 2017 not included in this analysis and trends limited to 1993-2015 for this season).

## 3 Results

### 3.1 Seasonal cycle

As mentioned above, the SURATLANT sampling is not available each year for the reference months of February and July selected for the summer and winter trend analysis (Table 1). When no data are available in February or July, we adjust the representative winter and summer data based on the deviations from the seasonal cycle observed in January or March (for winter) and June or August (for summer) according to equation (3).

$$SST^{year}_{month(A)} = SST^{clim}_{month(A)} + ( SST^{year}_{month(B)} - SST^{clim}_{month(B)})$$

(3)

Here, *month(A)* corresponds to February or July, *month(B)* to January or March (for winter), June or August (for summer) and *clim* to climatology.

The mean seasonal cycles constructed from data collected over the period 1993-2017 (only 2001-2017 for TA) are portrayed in Figure 2. For nutrients (not shown here) the seasonal cycle was described by Reverdin et al. (2018b). The seasonal cycles deduced here from the SURATLANT data represent classical variability observed in the North Atlantic north of 50°N and are coherent with other climatology of the oceanic carbonate system constructed from different data and methods (Lee et al., 2006; Takahashi et al., 2009, 2014; Becker et al., 2018). Except for salinity and TA, all properties show a marked seasonality in all regions with maxima and minima identified at the same period for boxes B, C, D and E. For the northern box E (62°N-64°N, black lines in Figure 2), we note some short time variability during summer for salinity and TA (minimum in July) that impact the $fCO_2$, pH and $\Omega$ cycles in this region.

The mean monthly DIC concentration (Fig. 2d) is maximum in winter due to deep vertical mixing, presents a steep decline from April to May when phytoplankton blooms start to occur, and a minimum at the end of summer. The mean DIC concentrations are very similar each month north of 54°N (boxes C, D, E) and significantly lower in the southern region (box B, red line in Figure 2) because salinity is also lower and the temperature higher from June to November in the south. The seasonal DIC amplitude of around 70-80 µmol kg$^{-1}$ in the north is slightly more

pronounced in the south of around 90 µmol kg$^{-1}$. Given the variability from year to year, the mean seasonal cycles of fCO$_2$, pH and $\Omega_{Ar}$ (Fig. 2e,f,g) show fairly similar variations in all boxes. As the alkalinity seasonality is low, the seasonal changes in fCO$_2$ and pH are anti-correlated and the seasonal variability of $\Omega$ is relatively similar to that of pH.

Although the temperature is +3°C to +6 °C warmer in summer compared to winter (Fig. 2a), the seasonal fCO$_2$ variability is mainly driven by DIC, i.e. the biological production in spring-summer dominates the effect of warming and deep mixing in winter dominates the cooling effect. The seasonal amplitude of fCO$_2$ that vary between 50 and 70 µatm (Fig. 4e) is much larger than in the atmosphere (around 14 µatm at these latitudes). Consequently, in a climatological view, the NASPG is near equilibrium in winter and a strong CO$_2$ sink in summer. How and why this CO$_2$ sink is changing over 1993-2017 will be investigated in the next sections.

**3.2 Long-term trend and anthropogenic CO$_2$**

When describing the SURATLANT data-set for 1993-2017, Reverdin et al. (2018b) evaluated the long-term trend of properties over a broad region (50°N-63°N). Using data over 24 years (and all seasons) they estimate an increase in DIC of +0.77 (±0.11) µmol kg$^{-1}$ yr$^{-1}$, a fCO$_2$ trend of +1.95 (±0.12) µatm yr$^{-1}$ and a pH decrease of -0.0021 (±0.0001) yr$^{-1}$. Here we re-evaluate these trends in each box (B, CDE merged) and only for summer (July) or winter (February) using observations and reconstructed data as described in section (2.3). The trends for 1993-2017 in each

box and season are listed in Table 2 (first lines). The DIC increasing rate ranges between +0.5 to +0.9 µmol kg$^{-1}$ yr$^{-1}$ depending on the season and region. For fCO$_2$ we evaluate trends ranging between +1.5 and +1.7 µatm yr$^{-1}$ and for pH between -0.0016 and -0.0019 yr$^{-1}$. Not surprisingly, because the same original data are used, the summer and winter trends for each box (B, CDE) are consistent with preliminary results from Reverdin et al. (2018b). From 1993 to 2017, we also observed a warming, most pronounced in summer (up to +0.05 °C yr$^{-1}$, Table 2), but the fCO$_2$ increase and pH decrease are mainly explained by DIC increase. The long-term fCO$_2$ and pH changes are thus mainly attributed to

anthropogenic CO$_2$ uptake. This is also supported by a decrease in $\delta^{13}C_{DIC}$ (Suess effect) along SURATLANT track but for a shorter period, 2005-2017 (Reverdin et al., 2018b), a signal observed at depth in the Irminger Basin between 1981 and 2006 and linked to anthropogenic CO$_2$ (Racapé et al., 2013).

To gain insight in the DIC trends and separate natural versus anthropogenic contributions to the fCO$_2$ and pH trends, we evaluate the

anthropogenic DIC (hereafter noted C-ant) in the NASPG region. Here we use the TrOCA method (Tracer combining Oxygen, inorganic Carbon and total Alkalinity, Touratier et al., 2007) applied to data available in the GLODAPv2 data-base (Key et al., 2015; Olsen et al., 2016). In the NASPG, repeated observations since 1997 were mostly conducted during summer (13 cruises in June-September). Because indirect methods such as TrOCA are not suitable to evaluate C-ant concentrations in surface waters (due to biological activity and gas exchange) we calculate C-ant in the layer 150-200m only. For the period 1997-2010, we estimate a C-ant trend of +0.6 µmol kg$^{-1}$ yr$^{-1}$ in the NASPG. The anthropogenic

signal would explain 65% of the DIC trend of +0.9 µmol kg$^{-1}$ yr$^{-1}$ observed in GLODAPv2 subsurface data for the same period in the region 54°N-64°N / 40°W-20°W. Note that we obtain similar results when selecting different periods (C-ant = +0.53 µmol kg$^{-1}$ yr$^{-1}$ for 1997-2007 whereas it is +0.58 µmol kg$^{-1}$ yr$^{-1}$ when restricted to 2001-2007).

Recently, a new data-based method, eMLR(C∗), developed by Clement and Gruber, (2018) was used to evaluate the accumulation of C-ant from

1994 to 2007 in the global ocean (Gruber et al., 2019a). To compare with our estimates based on TrOCA, we explore the C-ant concentrations from eMLR(C∗) in the NASPG (data extracted from Gruber et al., 2019b). Along the SURATLANT line, the accumulated C-ant between 1994 and 2007 is +8.5 (±1.7) µmol kg$^{-1}$ in the layer 150-200m. This would correspond to a C-ant trend of +0.65 µmol kg$^{-1}$ yr$^{-1}$ close to our C-ant estimate based on GLODAPv2 and TrOCA method in the same layer. This is not surprising as Gruber et al. (2019a,b) used GLODAPv2 data as well. In the surface layer, the eMLR(C∗) method leads to a C-ant accumulation in the layer 0-50m along the SURATLANT line (55°N-64°N),

of +10.1 (± 0.8) µmol kg$^{-1}$ between 1994 and 2007, i.e. about +0.8 µmol kg$^{-1}$ yr$^{-1}$. Again, this is in the range of the long-term (1993-2017) DIC

surface trends that we report (Table 2) between +0.7 µmol kg$^{-1}$ yr$^{-1}$ and +0.9 µmol kg$^{-1}$ yr$^{-1}$ in boxes CDE (depending on the season). Interestingly, in the region 55°N-64°N winter DIC from SURATLANT averaged 2134.1 (± 2.8) µmol kg$^{-1}$ in 1994 and 2145.1 (± 2.1) µmol kg$^{-1}$ in 2007, i.e. an increase of + 11 µmol kg$^{-1}$ almost equal to the C-ant accumulation deduced from eMLR(C$^*$) over the same period. These independent estimates confirm that in the NASPG the observed long-term DIC trend and derived trends for fCO$_2$ and pH in surface waters (Table 2, 1993-2017) are mostly linked with anthropogenic CO$_2$ uptake. The same conclusion was drawn for the subpolar mode waters found in subsurface layers in the NASPG (Vázquez-Rodríguez et al., 2012; García-Ibáñez et al., 2016; Fröb et al., 2018). For example, based on data collected in 1981-2008, Vázquez-Rodríguez et al. (2012) estimated that 75% of the observed pH decrease (-0.0019 yr$^{-1}$) in the Subarctic intermediate waters (SAIW) in the Irminger Basin is due to anthropogenic DIC. García-Ibáñez et al. (2016) incorporating 4 additional cruises (2010-2015) estimated a pH decrease of around -0.0018 yr$^{-1}$ over 1991-2015 in the subpolar mode waters (SPMW) found around 200-300m in the Irminger basin. They also conclude that in the SPMW the pH decrease is dominated by DIC increase (+0.82 µmol kg$^{-1}$ yr$^{-1}$) though anthropogenic CO$_2$ (C-ant trend of +0.9 µmol kg$^{-1}$ yr$^{-1}$) but this is modulated by a pH increase over time due to an increase in alkalinity (about +0.11 µmol kg$^{-1}$ yr$^{-1}$). From the SURATLANT time series (1993-2017) we also observe a small increase in TA during winter (+0.1 µmol kg$^{-1}$ yr$^{-1}$, Table 2) coherent with the TA signal observed in the SPMW layer from data collected in summer (García-Ibáñez et al., 2016). The TA increase in the SPMW layer in the Irminger basin is directly linked with salinity increase (García-Ibáñez et al., 2016) attributed to advection of higher salinity of subtropical origin and also associated with the contraction of the subpolar gyre since the mid-90s (Häkkinen and Rhines, 2004). However, we observed much stronger TA increase in surface water during summer, up to +0.4 µmol kg$^{-1}$ yr$^{-1}$ in the northern region (Table 2, box CDE). Such TA increase would lead to a pH change of around +0.0009 yr$^{-1}$, counteracting the impact of DIC increase. In addition, when TA is normalized in salinity, we evaluate summer sTA trends of +0.2 µmol kg$^{-1}$ yr$^{-1}$ in the north and +0.3 µmol kg$^{-1}$ yr$^{-1}$ in the south (Table 2) suggesting that the TA changes are not solely linked to salinity and advective process.

At high latitudes in the open ocean such long-term positive TA trend was only observed (to our knowledge) in the Pacific Western Subarctic Gyre (Wakita et al., 2013, 2017). There, in the winter mixed-layer, Wakita et al. (2017) estimate positive TA and sTA trends of +0.4 and +0.34 µmol kg$^{-1}$ yr$^{-1}$ over 1999-2015. This leads to a long-term pH decrease in this region (-0.0008 yr$^{-1}$) much slower than observed in other open ocean times-series stations (Bates et al., 2014) and in the NASPG for 1993-2017 (Table 2). The TA increase also leads to slow fCO$_2$ increase +0.9 µatm yr$^{-1}$ (Wakita et al., 2017). Wakita et al. (2017) suggested the TA increase might be due to weakened calcification (but this process was not quantified). Therefore, in addition to advection or convective processes, biological process such as calcification cannot be ruled out to explain TA and sTA long-term positive trends observed during summer in the NASPG (Table 2). At the high latitudes of the North Atlantic blooms of the coccolithophorid (*Emiliania huxleyi*) are well captured from space (Brown and Yoder, 1994) and there is indication that these blooms were more pronounced in the 1990s compared to the last two decades, 2000-2017 (Loveday and Smyth, 2018). Thus, in addition to DIC and anthropogenic CO$_2$, a better knowledge of temporal TA dynamics is relevant for an understanding of fCO$_2$ and pH changes. It is worth noting that Ocean BioGeochimical Model (OBGM) used to quantify the carbon cycle at large scale are not able to fully capture TA variability when calcifying phytoplankton is not explicitly included (Ullman et al., 2009, who used 1993-2005 SURATLANT data for model comparison). When multiple phytoplankton functional groups including coccolithophores are explicitly parameterized in ecosystem carbon models, the seasonal cycle of both DIC and TA are better represented in the NASPG (Signorini et al., 2012). The same is likely true to interpret inter-annual to decadal TA variability and explain the drivers of observed fCO$_2$ and pH trends along SURATLANT line.

Because interannual variability is more pronounced in summer than in winter because of the added influence of biological activity  most studies focused on winter to analyze and interpret the decadal trends of the carbonate system in surface waters (Olafsson et al., 2009; Metzl et al., 2010; Fröb et al., 2019; Omar et al., 2019). Only few observational studies conducted at high latitudes (Olafsson et al., 2009; Munro et al., 2015; Wakita et al 2017) showed that the trends of DIC, fCO$_2$ and pH are seasonally different and thus driven by different processes not yet fully explained. In a global context, taking into account a seasonal view for the trend analysis is also relevant to better understand the strengthening of fCO$_2$ seasonality observed over the last 3 decades (Landschützer et al., 2018) and how this would change the degree of acidification in the

future (Kwiatkowski and Orr, 2018). Note that the observed trend of $fCO_2$ seasonality (winter minus summer) in the North Atlantic appears less pronounced than in other regions with significant sub-decadal variability (see Fig. 1b and Fig. 4a in Landschützer et al., 2018). In this context we now investigate the observed trends for winter and summer in more detail and over specific periods to quantify the effect of temperature, salinity, DIC, TA that drive the $fCO_2$ and pH variations in the NASPG.

The periods 1993-1997, 2001-2007 and 2008-2017 are selected for several reasons. First, as the trends and quantification of drivers are sensitive to the data selection, we only select periods when time series for carbonates data were conducted (e.g. no DIC and TA data available in 1998-2000, see Table 1). This is in contrast with previous work where trends based on SURATLANT data were evaluated over 1993-2003 (Corbière et al., 2007; Metzl et al., 2010). Second, we use a new binned products based on more regular SST and SSS observations (Reverdin et al., 2018a)

10 to separate periods that present significant inter-annual variability and different trends in temperature and salinity (Fig. 3a,b). The three selected periods present also contrasting SST trends well identified at regional scale in the North Atlantic, at monthly or seasonal scales (Supp. Figure S2). Finally, in the NASPG it is well recognized that decadal and multi-decadal variability in temperature, salinity, winter convection and large-scale gyre circulation are associated with the phase of NAO or AMO (Fig. 3c).

During the first observational period 1993-1997 (rather short), NAO shifted from a positive to negative phase while AMO was in a transitional stage and regularly increased during the nineties (Fig. 3c). During the positive NAO in the early 90s, the NASPG experienced deep convection regime in the western subpolar gyre (Pickart et al., 2003). During the second period 2001-2007, NAO was negative or neutral and AMO reached a high index. The convection in the western NASPG was relatively shallow during this period and the ocean surface warmed (Fig. 3a). In the last decade, 2008-2017, NAO was highly variable compared to previous periods, with lowest phase in 2010 and highest phase in 2015 (Fig. 3c).

On the opposite, AMO strongly decreases since the late-2000s. During this period, both SST and SSS anomalies present clear negative trends: the NASPG becomes colder and fresher (Fig. 3a,b) during this period also associated with several deep convection events occurring in the western subpolar gyre during the winters 2008, 2012 and 2015 (Våge et al., 2008; de Jong and de Steur, 2016; Fröb et al., 2016; Piron et al., 2017).

**3.3 Winter trends for different periods**

The trends and drivers in winter are identified in blue symbols in Figures 4 and 5. During the first period, 1993-1997, the results are contrasted between the southern and northern regions. The strongest change occurs in the south (box B), where we observe a rapid cooling over 4 years (-0.19 °C yr$^{-1}$), an increase in salinity (+0.045 yr$^{-1}$) and an increase of TA (+2 µmol kg$^{-1}$ yr$^{-1}$) that is directly linked to salinity (recall that for 1993-1997 TA was not measured and the TA values are based on Salinity, Eq. 1). Over this short period, this leads to a strong decrease of $fCO_2$ (-7

µatm yr$^{-1}$) and an increase in pH both attributed to the cooling and TA changes (Fig. 5a,d), the effect of DIC being minor. In the north (boxes CDE), we do not observe any significant trend of winter properties in 1993-1997 for SST, sTA, sDIC, pH and $fCO_2$. As a result, in the NASPG (all region), the oceanic $fCO_2$ trend does not follow the atmospheric $CO_2$ increase and the air-sea disequilibrium increases, i.e. the region was a $CO_2$ source in winter 1994 and reached near-equilibrium afterwards (Fig. 4h).

As opposed to the nineties, data collected in winter during the second period (2002-2007) show a gradual DIC increase in the two regions (+1.7 and +2.1 µmol kg$^{-1}$ yr$^{-1}$, Table 2) higher than expected from anthropogenic $CO_2$ increase. A decrease in TA is also observed in the northern region but not directly related to salinity. This leads to significant pH decrease (-0.006 to -0.008 yr$^{-1}$) and fast $fCO_2$ increase (+6 to +7 µatm yr$^{-1}$). For this period, the DIC increase explains the temporal changes of pH and $fCO_2$ in box B, whereas both DIC and TA drive these changes in box CDE (Fig. 5b,e). In contrast to 1993-1997, the winter oceanic $CO_2$ source increases during 2002-2007. It is near equilibrium in 2002 and a

source in 2004-2007 (Fig. 4h) a result previously confirmed with independent sea surface $fCO_2$ observations (Metzl et al., 2010). At the end of this period, in winter 2007, oceanic $fCO_2$ was higher than 400 µatm and pH was low, near 8.04.

The last period, 2008-2015, was not investigated in previous studies (Corbière et al., 2007; Metzl et al., 2010). During 2008-2015, the DIC concentrations in winter continue to rise (+1.0 to +1.3 µmol kg$^{-1}$ yr$^{-1}$, Table 2) but at a slower pace compared to previous years and still higher than the anthropogenic DIC trend. Contrary to 2001-2007, we do not observe large changes in TA. In the northern region (box CDE) the sea surface cools (-0.05 °C yr$^{-1}$), which is not found in the southern region, inducing different fCO$_2$ and pH trends in the southern and northern sectors. In the north, the low fCO$_2$ and pH trends are due to DIC increase compensated by cooling and a small TA increase, whereas in the south the fCO$_2$ trend (+2.5 µatm yr$^{-1}$) and pH trend (-0.0024 yr$^{-1}$) are mainly driven by DIC increase (Fig. 5c,f). These trends deviate from the ones in 2001-2007 with different processes possibly related to NAO presenting much more variability during the last period (Fig. 3c). At the end of our time-series and in both regions fCO$_2$ in winter 2015 reaches the highest value (410 µatm) recorded since 1993, pH the lowest values (8.03) and the same for $\Omega_{Ar}$ (1.7). This is mainly explained by DIC concentrations approaching 2160 µmol kg$^{-1}$ in February 2015 (Fig. 4e). These high DIC concentrations observed in winter 2015 correspond to a high positive NAO phase (Fig. 3c) and to negative SST anomalies (Fig. 3a). It was also associated with strengthening of fresh water coming from the western NASPG (Holliday et al., 2020), and likely resulted in increased vertical mixing over the Reykjanes Ridge (De Boisséson et al., 2012). The high DIC and sDIC concentrations observed in winter 2015 are not unique to the Irminger Sea. Recently, based on sea surface fCO$_2$ measurements conducted in winter 2004-2017 and DIC calculated from fCO$_2$ data and reconstructed TA (using adapted TA/SSS relation), Fröb et al. (2019) report average winter sDIC concentrations of around 2160 µmol kg$^{-1}$ in 2012 and 2014. These authors also observed high sDIC concentrations in winter 2017, above 2165 µmol kg$^{-1}$. Unfortunately, we cannot compare these values as we have no winter data after 2015 along the SURATLANT line (Table 1). Although the periods are not exactly the same and methods to evaluate trends are different, the sDIC trends reported by Fröb et al. (2019) are in the same range as here: between +1.3 to + 1.75 µmol kg$^{-1}$ yr$^{-1}$ for the period 2004-2017, when we find (i.e. boxes CDE including the eastern Irminger Sea) a sDIC trend of + 1.8 µmol kg$^{-1}$ yr$^{-1}$ for 2001-2007 and +1.3 µmol kg$^{-1}$ yr$^{-1}$ for 2008-2015 (Table 2). Both results show a clear increase of sDIC in the NASPG or Irminger Sea since 2001 (or 2004), and the differences in trends might be also modulated by inter-annual variability.

During 2008-2015, we also observe significant inter-annual variability in winter, with marked DIC minima in 2010 and 2013 also found in sDIC concentrations as well as in TA and sTA (Fig. 4c,d,e,f). Interestingly, Fröb et al. (2019) also report lower sDIC concentrations in 2010 and 2013 in the Irminger Sea. In 2010, NAO shifted to a negative phase (Fig. 3c) supporting relatively shallow convection in the NASPG and inducing positive SST anomalies (Fig. 3a); the absence of deep mixed layers during these years would explain the relatively low DIC and TA concentrations observed in winter (i.e. less input of high DIC and TA from subsurface layers). In 2013, although the NAO was neutral we also observed lower DIC and TA in winter. It is suggested that mixing was relatively shallow in the Irminger Sea and near the Reykjanes Ridge in 2013, which could drive less renewal of the surface layers by the enriched sDIC deeper waters, and thus negative winter DIC and TA anomalies for this particular year. However, as both DIC and TA decrease, fCO$_2$ and pH are not strongly changing for these particular years (2010 and 2013) and associated air-sea CO$_2$ disequilibrium remains stable (Fig. 4h). The strong negative DIC anomalies observed in 2010 and 2013 nonetheless influence the DIC trend for the period 2008-2015: not including those two years increases the trend in boxes CDE from +1.0 µmol kg$^{-1}$ yr$^{-}$ to +1.3 µmol kg$^{-1}$ yr$^{-1}$. However, this is still lower compared to 2002-2007 (+1.7 µmol kg$^{-1}$ yr$^{-1}$) and thus much slower trend for fCO$_2$ and pH are estimated in recent years. As a result, in 2008-2015 we interpret the fCO$_2$ and pH winter changes equally due to the contribution of anthropogenic CO$_2$ (C-ant) and the natural component (DIC-nat), whereas in 2001-2007 the natural component dominates attributed to the dynamics in the NASPG (Fig. 6c,d).

### 3.4 Summer trends for different periods

We examine the trends and drivers in summer over the same periods selected for winter (except for the last period, with 2001-2017 in summer and 2002-2015 in winter). Results for summer are identified by red symbols in Figures 4 and 5. As mentioned earlier, a trend analysis in summer was not performed in previous work based on SURATLANT data (Metzl et al., 2010). Here we attempt for the first time to detect trends and processes occurring during this season and how they impact fCO$_2$, air-sea CO$_2$ equilibrium and pH changes. Not surprisingly, the temporal

variability of the carbonate properties is much more pronounced during summer when biological processes generally starting in spring imprint large DIC variations at seasonal scale (Fig. 2). The biological bloom and its timing may also lead to significant inter-annual variability but as we use July as a reference for summer, we expect to record each year the low DIC concentrations due to accumulated carbon uptake though production occurring in spring-summer.

During the first period, 1993-1996, the most remarkable feature is a very rapid increase of DIC in the northern and southern sectors (+4 and + 9 $\mu$mol kg$^{-1}$ yr$^{-1}$). This contrasts with the DIC trends in winter over 1994-1997 (Table 2). Because the northern box (CDE) also experienced a sharp warming of +0.6 °C yr$^{-1}$, there the fCO$_2$ increase (+12 to +14 $\mu$atm yr$^{-1}$) and pH decrease (-0.015 to -0.017 yr$^{-1}$) are particularly fast. However, the period is too short to clearly interpret which are the main processes at play and derive any conclusion on the trends. We notice that in 1993-1996, NAO changed from a positive to negative phase and AMO progressively increased in the nineties (Fig. 3c) but no particular anomalies were revealed in the winter observations. Thus, we have no straight explanation for the fast DIC, fCO2 and pH changes observed during the summers 1993-1996, except that relatively higher DIC in 1996 might have resulted from a decrease in primary production compared with previous years. This is rather speculative as we have no other information on nutrients or on the biological activity (e.g. no nutrient data for summer 1996 and no remote sensing data at the beginning of the nineties). However, using CPR data over 1960-2010 in both western and eastern NASPG regions, Martinez et al. (2016) identified negative anomalies of Chl-a around 1996-1997 after a sharp increase of Chl-a during 1986–1995. The fCO$_2$ data show that the region was a large CO$_2$ sink in summer 1993-1995 but abruptly rised to near-equilibrium values in 1996 (Fig. 4h). In the northern region (box CDE), the mean delta-fCO$_2$ value in summer 1996 was close to 0 $\mu$atm, i.e. about the same as observed during the winter 1996-1997.

The SURATANT regular sampling for DIC, TA and nutrients was restarted in June 2001 (Table 1). During 2001-2007, and as in summer 1993-1996, we observe a rapid increase of fCO$_2$ (+8 to +12 $\mu$atm yr$^{-1}$) and pH decrease (-0.010 to -0.013 yr$^{-1}$). Despite having reconstructed July from the data obtained in August (for boxes B and CDE) both for the starting (2001) and ending (2007) years, this result is fairly robust. Indeed, if we do not normalize summer to July, but directly include the June / July / August data, we also observe a rapid increase in fCO$_2$ of +6.2 $\mu$atm yr$^{-1}$. The summer DIC and TA trends in the northern and southern areas have the same sign (similar as during winter), positive for DIC and negative for TA but not directly related to salinity (Table 2). However, the DIC increasing and TA decreasing rates are regionally significantly different. The DIC increase is most pronounced in the north (+5 $\mu$mol kg$^{-1}$ yr$^{-1}$) and the TA decrease more pronounced in the south (-3.9 $\mu$mol kg$^{-1}$ yr$^{-1}$). Although the fCO$_2$ and pH trends are similar in the two areas, this suggests different drivers for boxes B and CDE (Fig. 5b,e). In the north, DIC explains most of the fCO$_2$ and pH change, whereas in the south the TA contribution dominates. In both regions, as in winter, temperature and salinity changes have a small effect for this period (Fig. 5b,e). The rapid oceanic fCO$_2$ increase strongly impacts the variations of air-sea CO$_2$ disequilibrium (Fig. 4h). In the north, the region was a sink in 2001-2004 but reaches equilibrium in 2005 and 2007 (no data in July 2006 in the north). In July 2007, temperature was relatively low (around 10°C) and the high fCO$_2$ at that period was mainly due to high DIC concentrations either linked to low productivity that year.

For the last period, 2008-2017, results in summer are very different compared with previous decades as we now observe a decrease of fCO$_2$ and an increase of pH in both regions (Fig. 4g,i; Table 2). This also contrasts with the winter trends in 2008-2015. The variation of fCO$_2$ leads to a strong ocean CO$_2$ sink with low delta-fCO$_2$ in 2010-2017 (-40 to -80 $\mu$atm, Fig. 4h). During the summers 2008-2017 sea surface properties are very variable that might be related to the NAO variability in this period (Fig. 3c), but also to changes in the ship's route in some years or to productivity occurring at meso-scale. This was the case in July 2013 where the sampling took place further north-west (Fig. 1), impacting SST (colder, Fig. 4a) and DIC (stronger, Fig. 4 e,f). However, these anomalies are less noticeable on fCO$_2$ and pH (Fig. 4g, i) because the effects of SST and DIC partially compensate. Thus, we chose to not consider July 2013 for calculating trends and drivers.

Although the inter-annual variability is large in summer, the trends are evaluated over the period 2008-2017 using all available data (except July 2013). For this period all terms (SST, DIC and TA) contribute to the $fCO_2$ and pH changes (Fig. 5c,f). The DIC increase is only revealed in the north (box CDE, +1.2 µmol $kg^{-1}$ $yr^{-1}$); this is slightly higher than in winter (+1.0 µmol $kg^{-1}$ $yr^{-1}$) and thus higher than anthropogenic signal (around +0.6 µmol $kg^{-1}$ $yr^{-1}$). However, as the surface ocean cools (-0.11 °C $yr^{-1}$) and TA increase (+1.7 µmol $kg^{-1}$ $yr^{-1}$), the net effect is a
5      decrease of $fCO_2$ (-2.3 µatm $yr^{-1}$) and pH increase (+0.0026 $yr^{-1}$). During 2008-2017, the progressive cooling in the NASPG, found here in both winter and summer data (Fig. 4a) is a large-scale signal (Fig. 3a, Robson et al., 2016; Reverdin et al., 2018a). An intriguing signal in 2008-2017 is the increase of TA (Fig. 4c) opposed to the decease of salinity (Fig. 3e, Fig. 4b); this leads to positive sTA trend opposite to summer 2001-2007. Therefore, we cannot interpret the observed TA increase directly linked to salinity. The shift of the sTA trend, negative in 2001-2007 and positive in 2008-2017, is an important signal that drives opposite trends for $fCO_2$ and pH between the two periods (Fig. 5b,c,e,f).

This contrasting TA signal in summer cannot be attributed to error or drift in laboratory analyses as the measurements were performed using the same methods for the whole time series (Reverdin et al., 2018b) and no such signal is identified for winter cruises in 2008-2015. In addition, during OVIDE cruises conducted in summer 2006-2014 in the North Atlantic (e.g. García-Ibáñez et al., 2016), we performed regular TA and DIC inter-comparisons with the ICM/CSIC group in Vigo (F. Pérez) and certified our results to within around ±4 µmol $kg^{-1}$ both for TA and
DIC. Results based on the surface TA samples measured at LOCEAN for the OVIDE cruises in summer 2006-2018 (data in Metzl et al., 2018) also suggest that sTA increases in the NASPG at a rate of around +1.5 µmol $kg^{-1}$ $yr^{-1}$.

To support these results, we have evaluated sTA trends in the NASPG from independent data available in the most recent GLODAPv2.2019 version (Olsen et al., 2019). For this we selected the data in the layer 0-20m for all cruises conducted in June-August in 1997-2014. We found
a significant difference of sTA trends during this period: -0.52 µmol $kg^{-1}$ $yr^{-1}$ in 1997-2006 against +0.54 µmol $kg^{-1}$ $yr^{-1}$ in 2006-2014. Although the periods are not exactly the same and sTA trends from GLODAPv2.2019 are smaller than those deduced from the SURATLANT time-series, the changing sTA trends from negative to positive in recent years also features in that dataset. We are thus confident with TA data over time and need to find a process that explains why the sTA trend was negative in summer 2001-2007 (-1.9 µmol $kg^{-1}$ $yr^{-1}$) and positive in summer 2008-2017 (+2.3 µmol $kg^{-1}$ $yr^{-1}$).

The variability of calcification through the production of calcifying species (e.g. coccolithophores, foraminifera) is a possible mechanism that would impact TA trends as suggested by Wakita et al. (2017) in the Pacific Western Subarctic Gyre. In the North-East Atlantic, long-term in-situ CPR observations (Continuous Plankton Recorder) showed a significant increase of the calcifying species starting in the mid-90s (Beaugrand et al., 2013). These authors suggest that the temperature (a warming) was the main driver of the positive trend of calcifying plankton. In addition
they show that no correlation was identified between the NAO and species variability but that positive trend of the calcifying species was correlated with AMO, a result also confirmed by Rivero-Calle et al. (2015). The TA decrease in summer we observe during the period 2001-2007 after the AMO moved to its positive state (Fig. 3c), might be explained by the increase of calcifying species as identified from in-situ CPR observations in the North Atlantic (Beaugrand et al., 2013; Rivero-Calle et al., 2015). Unfortunately, we don't have yet direct in-situ observational evidence of a reduced calcifying species after 2010. However, if one follows the proposed scenario for 2001-2017, the TA increase
in summer 2008-2017 during the cooling phase in the NASPG could be linked to a weakening of calcification. In this region, this seems supported by the absence of coccolithophores blooms in recent years (2010-2017) as identified from remote-sensing reflectance records (Loveday and Smyth, 2018).

## 4 Discussion

Our observations collected over the last three decades show an abrupt change in the evolution of hydrological and biogeochemical properties in
the NASPG around the year 2007 (Fig. 4) when AMO reached a maximum and NAO was around neutral (Fig. 3c). Following a warming since

the mid-90s, the region experienced a large cooling and freshening after 2005 (Fig 3a,b; Robson et al., 2016; Holliday et al., 2020). This change is associated with an increase in the size of the gyre and increased currents along the gyre southern rim (Chafik et al., 2014, 2019; Desbruyères et al., 2015), as well as by a very large heat loss during positive NAO years particularly in 2015 (Fig 3c, Josey et al., 2018). Indeed, NAO was previously recognized as a possible cause of the rapid $fCO_2$ increase when NAO shifted from positive to negative phase in 1995-1996 (Fig. 3c; Corbière et al., 2007). However, this was not confirmed for the period 2001-2007 when NAO did not vary so much around a neutral value (Fig. 3c; Metzl et al., 2010). For the whole NA, Schuster et al. (2009) synthetized $fCO_2$ observations over 1990-2006, i.e. before 2007 when we observe abrupt changes in property trends. North of 45°N, Schuster et al. (2009) evaluate a trend of $fCO_2$ exceeding +3 µatm $yr^{-1}$ and a significant decrease of the $CO_2$ sink in the NA by over 50% between 1990 and 2006. They also predict an increasing sink in the subpolar regions following the increasing NAO index in 2007. Although the impact of the climatic modes (NAO and/or AMO) on the oceanic physical properties and circulation has been well established, their effects, if any on the $fCO_2$ and pH trends need to be clarified.

Whatever the NAO variability, the long term $fCO_2$ and pH trends we evaluate over 1993-2017 in the NASPG for summer or winter are mainly explained by the increase in DIC associated with the uptake of anthropogenic $CO_2$ (Fig. 6), that is the DIC or sDIC trends (Table 2) are not significantly different from the anthropogenic DIC trend estimated between 1994 and 2007 in this region (Gruber et al., 2019a). For $fCO_2$, here calculated from DIC/TA pairs, the long-term trends between +1.5 to +1.7 µatm $yr^{-1}$ in the NASPG (Table 2) are slightly higher than the mean trend of +1.47 (±0.06) µatm $yr^{-1}$ evaluated for 1992-2014 in the whole NA (Lebehot et al., 2019) based on monthly reconstructed $fCO_2$ using a Multiple Linear Regression (MLR) approach and SOCAT-v4 $fCO_2$ data (Bakker et al., 2016). Lebehot et al. (2019) show that the $fCO_2$ trends based on observations are much lower than those derived from 19 ESM CMIP5 models, +1.90 (±0.09) µatm $yr^{-1}$. By performing several sensitivity test analyses on the ocean ESM models, Lebehot et al. (2019) conclude that the discrepancy between observed and simulated $fCO_2$ trends originates mainly in model's biogeochemistry, e.g. biases in simulated TA and also related to the way ESM models do or do not represent winter mixing (Goris et al., 2018). This might be especially relevant for the NA subpolar region where some ESM models project faster change of ocean $fCO_2$ in the future (2061-2100) with 60% due to the DIC increase and up to 29% due to the TA changes (Tjiputra et al., 2014). We also suspect that ESM models used to predict future change of the oceanic $CO_2$ sink and ocean acidification would produce faster pH trends than what we observed.

If the long-term $fCO_2$ and pH trends could be mainly explained by anthropogenic $CO_2$ uptake in the NASPG, at shorter time scales (4-10 years) the trends are very different (Table 2). Indeed, the gradual changes of $fCO_2$ and pH caused each year by the uptake of anthropogenic $CO_2$ can be significantly masked by the natural variability of DIC (DIC-nat), temperature and/or TA. To better identify when and why the DIC-nat dominates we separately compute the impact of anthropogenic $CO_2$ (C-ant) and DIC-nat on pH and $fCO_2$ trends for each season and periods. The results for pH are presented in Figure 6. The same results are obtained for $fCO_2$ (not shown), with an opposite sign for each bar plotted on Figure 6. For C-ant we adopt a value of +0.6 µmol $kg^{-1}$ $yr^{-1}$ as described in section 3.2.

For all sub-periods and both seasons, the effect of DIC-nat on pH trend is significant and with similar or higher magnitude than the effect of C-ant (Fig. 6). Occasionally it opposes C-ant, i.e. DIC-nat decreases with a positive effect on pH trend (Box B, 2008-2017 in summer, Fig. 6a and Box CDE, 1994-1997 in winter, Fig. 6d). Over 1993-1997 the DIC-nat effect on pH trends for this short period is opposed in summer and winter, suggesting that this is not linked to changes in regional circulation, e.g. less or more input of DIC from different water masses. For 1993-1997 the largest DIC-nat effect being observed in summer (in the northern and southern boxes), one might suggest that it is linked to primary productivity; unfortunately, prior to SeaWIFS in 1998, we have no direct or indirect information on biological changes to explain why DIC-nat increased in summer 1993-1997. However, the length of the period is short and trend results are very sensitive to interannual anomalies, especially in the first and last years.

Apart for the first short period (1993-1997) the largest effect of DIC-nat is observed in summer 2001-2007 in the northern region (Fig 6b). This period includes a decade (1995-2005) where the strength of the subpolar gyre circulation decreased (Häkkinen and Rhines, 2004, 2009; Häkkinen et al., 2011, 2013) suggesting that more water of subtropical origin penetrates in the NASPG. This would have decreased the DIC surface concentrations (and increase TA as well), but for 2001-2007 we observed the opposite, including in winter (Table 2). We thus eliminate

the effect of advection to interpret the DIC-nat increase in the NASPG that impacts significantly on pH trend. As mixed-layer depths are not deeper than 30-40m in summer and present no significant inter-annual changes (according to the dataset Armor which presents the same variations as the data of product Global Reanalysis PHY 001 030 distributed on http://marine.copernicus.eu/), the variation of the vertical mixing during this season is not a likely candidate to explain the changes of DIC-nat. A possible explanation for the large contribution DIC-nat on pH (and $fCO_2$) trend in 2001-2007 could be a decline in biomass identified in SeaWIFS data in the eastern subpolar region for years 1998-2007

although the trends for net primary productivity (NPP) appears not statistically significant over 10 years (McKinley et al., 2018). For the NPP subject to high frequency variability the evaluation of trends over nearly 10 years is challenging. Annual biomass anomalies (based on SeaWIFS and MODIS sensors) changed from positive values in 1998-2004 to negative ones in 2005-2009 (McKinley et al., 2018). To explain the biomass decline McKinley et al. (2018) used a coupled physical biogeochemical ocean model (OBGM) to reproduce these changes and found that nutrient concentrations decline significantly in the region north of 50°N due to reduced physical supply through horizontal and vertical fluxes; their

model suggests that enhanced phosphate and silicate limitation over time dominates the light limitation in this region.

Due to limited sampling months and the high variability of nutrients in spring-summer we have not been able to detect such mechanism from the SURATLANT nutrient data. However, the reduced productivity due to nutrient limitations (McKinley et al., 2018) is supported by independent observations of silicate concentrations in the subpolar NA over 25 years (McKinley et al., 2018). They showed a decline in pre-

bloom silicate concentrations in the winter mixed layer since the 1990s until 2010 and attributed it to the decrease in winter convection depth and the weakening of the NASPG. This decline of silicates observed over 25 years would negatively impact diatom blooms in spring and favor coccolithophore blooms occurring in summer. Such scenario is coherent with the increase in calcifying species since the mid-90s as observed from the CPR data in the NA during a warm period associated to the AMO (Beaugrand et al., 2013; Rivero-Calle et al., 2015). The shift of phytoplankton species and local or regional intensified calcification might also explain the low sTA concentrations occasionally observed in

2003 and 2005, i.e. lower than for summer 2001 and also for winter (Fig. 4d). The inter-annual variability of TA of around $\pm 8$ µmol kg$^{-1}$ in 2001-2007 is twice the TA change of $\pm 4$ µmol kg$^{-1}$ due to coccolithophores in a biological model applied at 60°N-30°W (Signorini et al., 2012). The TA and sTA negative trends observed in 2001-2007 (Table 2) contribute less to pH and $fCO_2$ trends than with DIC (Fig. 5b,e), but nearly the same magnitude compared to the contribution of C-ant. This is not the case after 2007.

During the last decade, 2008-2017, observations were obtained during a strong negative NAO in 2010 and a positive NAO phase in 2015 (Fig. 3c). NAO presents large inter-annual variation compared to 1995-2007 (Fig. 3c) and a decline in AMO index after 2010 (Fig. 3c). The 2010 event was associated with a warming and freshening (Fig. 4a,b) found in both SURATLANT discrete sampling (in August 2010) and monthly reconstructed Binned products (Reverdin et al., 2018a). In August 2010 the DIC concentrations were very low in the north (< 2070 µmol kg$^{-1}$) compared with other years since 1993 (Fig. 4e). This was not associated with particular signals in TA but with high $\delta^{13}C_{DIC}$ as reported in Racapé

et al. (2014) and also with higher Chl-a concentration identified from MODIS data (McKinley et al., 2018). Indeed, Henson et al. (2013) showed that the physical forcing caused by the very negative NAO recorded in winter 2009-2010 stimulated spring blooms and not the eruption of the volcano Eyjafjallajökull in Iceland which erupted in spring 2010, depositing large amounts of iron in the North Atlantic Subpolar. The effect of the higher productivity in summer 2010 is to lower $fCO_2$ and increase pH compared to previous summers. Thus, there was a rapid drop in delta-$fCO_2$ in 2009-2010, such that summer 2010 was a strong $CO_2$ sink. In 2015, when NAO was in a positive phase, the SST anomaly was on the

order of -1°C (Fig. 3a): for that year, DIC was high in winter, close to the maximum observed in winter in our time series (Fig 4 e, f). As the temperature also lowers $fCO_2$, the $fCO_2$ values (and pH) were not so different from other winters, illustrating the competing effects of cooling and deeper vertical mixing on $fCO_2$. Although significant changes are observed on DIC variability during NAO events in 2010 and 2015, they

seem to have a small impact on the trends. After 2007 the positive DIC trends are less important than in 2001-2007 (Table 2) and the contributions of the natural and anthropogenic parts of DIC have a similar magnitude in summer and winter (Fig. 6). This result alone does not explain the decrease in $fCO_2$ and the increase in pH observed during the last period in summer. Indeed, there is also a significant impact of cooling and as opposed to 2001-2007, an intriguing increase in TA, both leading to decreasing $fCO_2$ and increasing pH trends (Fig. 4; Fig. 5).

5 **5 Conclusion and perspectives**

Based on sea surface observations of DIC and TA collected in the NASPG over 1993-2017 we have analyzed the variability and trends of the carbonate system properties including $fCO_2$, pH, $\Omega_{Ar}$ and $\Omega_{Ca}$ calculated from the DIC/TA pairs. This study extends to summer and for pH and $\Omega_{Ar}$ previous work based on winter data (Corbière et al., 2007; Metzl et al., 2010). It also extends the analysis for the last decade, 2008-2017, after the AMO reached a maximum and then decline and when the NAO was highly variable.

In the last decade we observed a continuous surface DIC increase in winter in the northern and southern NASPG. In February 2015 when the NAO was in a positive state, DIC reached a maximum concentration (DIC > 2150 µmol kg$^{-1}$, i.e. more than +20 µmol kg$^{-1}$ higher than in the 90s). In 2015 pH was at minimum (8.03) and $fCO_2$ exceeded 400 µatm and close to the atmospheric $fCO_2$. Such high $fCO_2$ was also observed in the NASPG in Jan-Feb 2015 from direct underway measurements (range 405-415 µatm for cruises AGFO20150115 and AGFO20150212
15 (measured by Wanninkhof NOAA/AOML on the M/V Skogafoss) in SOCAT-V5, Bakker et al., 2016). As opposed to the wintertime DIC continuous increase observed since the 90s, the TA decadal variability is not uniform. In 2001-2007 the decreasing of TA, added to the DIC changes, reinforced a rapid $fCO_2$ increase in up to +7 µatm yr$^{-1}$ and a strong pH decline of around -0.007 yr$^{-1}$ confirming previous studies (Metzl et al., 2010). On the opposite, in 2008-2015 the increasing TA and the cooling in the northern NASPG compensated the effect of DIC increase, leading to much smaller winter trends for $fCO_2$ and pH, on the order of +1 µatm yr$^{-1}$ and -0.001 yr$^{-1}$ respectively.

During summer, the inter-annual variability of all properties is much more pronounced due to active primary productivity in the NASPG in spring-summer and shallow mixed-layers. Consequently, the decadal trends of the carbonate system properties in summer are more difficult to detect compared with winter or other oceanic regions such as the subtropics (Bates et al., 2014; Ono et al., 2019). In addition, primary production often occurs at small spatio/temporal scales and the DIC/TA heterogeneous sampling may have occasionally missed planktonic blooms. In 2001-
25 2007 the summer trends of SST, DIC and TA has the same sign as in winter. This confirms the fast increase of $fCO_2$ and the strong decline of pH observed during this period. The natural variability of DIC dominates the effect of the anthropogenic uptake on $fCO_2$ and pH trends, especially in the northern region (Fig. 6). The DIC increase in summer is likely due to a reduced productivity during this period (Tilstone et al., 2014; McKinley et al., 2018). After 2007, the $fCO_2$ and pH trends for summer are drastically different from the previous decade. In both southern and northern parts of the transect, the $fCO_2$ trend becomes negative while the pH trend is positive (up to +0.0052 yr$^{-1}$ in the south). This is driven by
30 a complex interplay of cooling (-0.1 °C yr$^{-1}$), DIC increase and significant increase of TA more pronounced in summer than in winter.

Before 2007, we evaluate a rapid $fCO_2$ increase faster than in the atmosphere. As a result, the ocean $CO_2$ uptake decreased in the NASPG from 1993 to 2007, in agreement with other studies (Schuster et al., 2009; Landschützer et al., 2013). However, at larger scale, here for the NA-SPSS regional biome (Fay and McKinley, 2014), the decreasing $CO_2$ sink in the NASPG for this period is not always resolved from $fCO_2$ data-based
methods that evaluate an increasing $CO_2$ sink in the North Atlantic Subpolar region (Rödenbeck et al., 2015; Denvil-Sommer et al., 2019). This disagreement might be in part due to missing $fCO_2$ data for the period 1994-2003 in the NASPG (Bakker et al., 2016) and need to be investigated in further studies. After 2007, our results show that winter $fCO_2$ increased at a lower rate than in the atmosphere whereas in summer we observe a decrease in $fCO_2$. Thus, the ocean $CO_2$ sink increases over 2008-2017 (Fig. 4h). For this period this is coherent with the results derived from data-based methods in the NA-SPSS biome but only for 2007-2013. Indeed after 2013 the indirect methods produce either an increasing $CO_2$

sink or a decreasing $CO_2$ sink and with a very high variability noticeable in 2015 when the NAO was positive (four indirect methods present on Fig. S5h in Denvil-Sommer et al., 2019).

The observed change of all sea surface properties around 2007 in the NASPG, as well as a decreasing $CO_2$ sink before 2007 and an increasing $CO_2$ sink after 2008 (Fig. 4h) seems rather linked to the AMO than directly to NAO (Fig. 3c) as also suggested from data-based reconstructed $fCO_2$ fields in the North Atlantic (Landschützer et al., 2019).

The temporal change of $fCO_2$ and $CO_2$ uptake around 2007 that we deduce from SST, DIC and TA, is also clearly observed in pH variability. In summer, the rapid pH decline in 2001-2007 is followed by a significant pH increase in 2008-2017. This is due to a changing trend in TA that was negative before 2007 and positive after 2008, a signal observed in both seasons. This unexpected decadal change in TA, also observed in the North-Western Subpolar Pacific might be linked to changes in calcification processes (Wakita et al., 2017). Indeed, it has been recognized that calcifying species in the North Atlantic present significant variations since the mid-90s (Beaugrand et al., 2013). To quantify how this process impacts $fCO_2$ and pH variability deserves further studies to investigate the coupling of chemical measurements with species observations such as obtained from CPR in the NASPG.

Although we identified significant inter-annual to decadal variability in surface carbonate system properties over the full time-series (1993-2017) the long-term trends of $fCO_2$ and pH in winter and summer are almost entirely explained by the DIC increase and anthropogenic $CO_2$ uptake (Fig. 6). The long-term trend of $fCO_2$ in the NASPG (between +1.5 and +1.7 µatm yr$^{-1}$) is slightly higher than the mean trend for 1992-2014 (+1.47 µatm yr$^{-1}$) evaluated at large scale in the NA (Lebehot et al., 2019). Our results now extended to 2017 confirm the $fCO_2$ trends evaluated by McKinley et al. (2011) for the period 1981-2009, and our new estimate for 1993-2017 in summer (+1.7 µatm yr$^{-1}$) is not significantly different from the trend of +1.8 µatm yr$^{-1}$ estimated for 1981-2002 based on a few observations in August (Corbière et al., 2007). However, it is worth noting that in other NA sectors and periods, such $fCO_2$ increase is not always observed. In a recent study focused in the mid-latitude NA (40-50°N), Macovei et al. (2019) showed that $fCO_2$ is highly variable with a small trend of +0.37 (±0.22) µatm yr$^{-1}$ in 2002-2016 implying a significant increase of the $CO_2$ uptake in this region especially after 2010 when we also observed a sudden drop of $fCO_2$ in summer in the NASPG (Fig. 4g,h). Our results added to those from Macovei et al. (2019) suggest the ocean $CO_2$ sink increased in the North Atlantic (north of 40°N) at least since 2007 and until 2017, contributing to the increase of the global ocean CO2 sink (Friedlingstein et al., 2019). At the NA basin scale this is consistent with results deduced from $fCO_2$ reconstructed data-based methods (Denvil-Sommer et al., 2019; Gregor et al., 2019) but opposed to the ocean $CO_2$ sink variability generated by current ESM models suggesting uncertainties for predicting the evolution of the NA $CO_2$ sink in the future (Lebehot et al., 2019).

For pH the long-term trend in the NASPG of -0.0017 yr$^{-1}$ is in the range of what is observed in other oceanic regions but we note some differences. This trend is equal to the trend estimated at BATS station but is lower than in the Irminger Sea and higher than in the Iceland Sea (Bates et al., 2014). It is also lower than the pH trend of -0.0020 yr$^{-1}$ estimated in the North Atlantic Subpolar region in 1991-2011 (Lauvset et al., 2015) and -0.0024 yr$^{-1}$ recently observed in the NA waters in the North Sea in 2004-2015 (Omar et al., 2019).

Given the large differences of $fCO_2$ and pH trends at regional scale listed above, our results among many others highlight the need for acquiring sustained time series of ocean carbonates system in different regions as has been strongly recommended at the recent Ocean-Obs19 conference (Tilbrook et al., 2019; Wanninkhof et al., 2019).

An understanding of these differences also calls for a comprehensive analysis including a synthesis of all DIC and TA sea surface observations collected in different regions. This should be achieved at an international level as is done for sea surface $fCO_2$ in the frame of SOCAT (Bakker et al., 2016) or for CLIVAR/GO-SHIP cruises assembled in GLODAP (Olsen et al., 2019). In addition to ship-based observations, the analysis

of data from BGC-Argo floats equipped with pH sensors (together with temperature and salinity sensors, from which TA, DIC and fCO$_2$ can be estimated, e.g. Williams et al., 2017) will help to better constrain spatial, seasonal and inter-annual variability. Such data synthesis would also help to validate ocean and earth system models that at present do not represent correctly the temporal change of marine biogeochemistry as demonstrated by Lebehot et al. (2019) for the North Atlantic.

*Data availability.* The data set is freely available and is accessible at http://www.seanoe.org/data/00434/54517/ (http://doi.org/10.17882/54517, Reverdin et al. (2018b).

*Author contributions.* CL produced the data analyses and wrote the manuscript with inputs from NM, GR and CLM. GR, NM and VR produced the data synthesis. SO provided the nutrients data. JF provided the DIC and TA data.

*Competing interests.* The authors declare that they have no conflict of interest.

*Acknowledgments.* The SURATLANT project is supported by the French institute INSU (Institut National des Sciences de l'Univers), the Lamont-Doherty Earth Observatory (LDEO), the National Oceanic and Atmospheric Administration (NOAA) - Atlantic Oceanographic and Meteorological Laboratory (AOML) and the Climate Program Office (CPO). We thank the EIMSKIP Company and the MFRI team, both based in Reykjavik (Iceland), for their cooperation in sea water sampling and analysis. We also thank the numerous scientific volunteers who worked at sea, as well as the crew and captain of the vessels for their help. Support from the European Integrated Projects CARBOOCEAN (511176) and CARBOCHANGE (264879) is also acknowledged. We thank the reviewers, Are Olsen and an anonymous reviewer for their helpful comments on this work. We have an emotional thought for our late colleague Taro Takahashi who contributed to initiate this sampling in 1993 and was a strong source of motivation for maintaining this long-time monitoring.

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

**Table 1**. SURATLANT sampling since 1993 for each box in Fig. 1.

| | Jan | Feb | Mar | Apr | May | Jun | Jul | Aug | Sep | Oct | Nov | Dec |
|---|---|---|---|---|---|---|---|---|---|---|---|---|
| **1993** | | | | | | | A-B-C-D-E | | | | | |
| **1994** | A-B-C-D-E | | | A-B-C-D-E | | A-B-C-D-E | | | A-B-C-D-E | | | |
| **1995** | A-B-C-D-E | | | | A-B-C-D-E | | A-B-C-D-E | | | | | A-B-C |
| **1996** | C-D-E | | A-B-C-D-E | | | | A-B-C-D-E | | | | | |
| **1997** | | A-B-C-D-E | | | | | | | | | | |
| **2001** | | | | | | A-B-C-D-E | | A-B-C-D-E | | | | A-B-C-D-E |
| **2002** | | A-B-C-D-E | | | A-B-C-D-E | | | | A-B-C-D-E | | | A-B-C-D |
| **2003** | | | A-B-C-D-E | | | | A-B-C-D-E | | | B-C-D-E | | |
| **2004** | B-C-D-E | | | B-C-D-E | | | A-B-C-D-E | | | | B-C-D-E | |
| **2005** | | | | A-B-C-D-E | A | A-B-C-D-E | | | | | B-C-D-E | |
| **2006** | | B-C-D-E | | A-B-C-D-E | | | A-B-C | | | | A-B-C-D-E | |
| **2007** | | B-C-D-E | | | | | | A-B-C-D-E | | A-B-C-D-E | | |
| **2008** | | | A-B-C-D-E | | B-C-D-E | | B | C-D-E | | | A-B-C-D-E | |
| **2009** | | A-B-C-D-E | | | | A-B-C-D-E | | | B-C-D-E | | | A-B-C-D-E |
| **2010** | | | B-C-D-E | | | A-B-C-D-E | | A-B-C-D-E | | | | A-B-C-D-E |
| **2011** | | | B-C-D-E | | | | | | | | | B-C-D-E |
| **2012** | | | A-B-C-D-E | | | E | A-B-C-D-E | | A-B-C-D-E | A | | A-B-C-D |
| **2013** | | | B-C-D-E | | | | A-B-C-D-E | | B-C-D-E | A | | |
| **2014** | A-B-C-D-E | | | A-B-C-D-E | | | A-B-C-D-E | | | A-B-C-D-E | | |
| **2015** | A-B-C-D-E | | | | | A-B-C-D-E | | | A-B-C-D-E | | | A-B-C-D-E |
| **2016** | | | A-B-C-D-E | | | | A-B-C-D-E | | | B-C-D-E | | |
| **2017** | | | | | | | B-C-D-E | C-D | | | | |

**Table 2.** Trends (per year) evaluated from data presented in Figure 4. In bold are represented the significant trends (Student test).

| period | season | box | SST (°C yr⁻¹) | SSS (yr⁻¹) | TA (µmol kg yr⁻¹) | sTA (µmol kg yr⁻¹) | DIC (µmol kg yr⁻¹) | sDIC (µmol kg yr⁻¹) | pH (yr⁻¹) | fCO₂ (µatm yr⁻¹) | ΩAr (yr⁻¹) | ΩCa (yr⁻¹) |
|---|---|---|---|---|---|---|---|---|---|---|---|---|
| 1993-2017 | summer | B | **0.05 ±0** | **-0.003 ±0** | 0.1 ±0.1 | **0.3 ±0** | **0.5 ±0** | **0.7 ±0** | **-0.0017 ±0.0001** | **1.5 ±0.1** | **-0.003 ±0** | **-0.005 ±0.001** |
| | | C.D.E | **0.03 ±0** | **0.003 ±0** | **0.4 ±0** | 0.2 ±0 | **0.9 ±0** | **0.8 ±0** | **-0.0019 ±0.0001** | **1.7 ±0.1** | **-0.005 ±0** | **-0.008 ±0** |
| | winter | B | **0.02 ±0** | **0.003 ±0** | 0.1 ±0.1 | 0.0 ±0.1 | **0.6 ±0.1** | **0.5 ±0.1** | **-0.0016 ±0.0002** | **1.6 ±0.2** | **-0.004 ±0.001** | **-0.006 ±0.001** |
| | | C.D.E | **0.01 ±0** | **0.005 ±0** | **0.1 ±0.1** | -0.1 ±0 | **0.7 ±0** | **0.4 ±0** | **-0.0016 ±0.0001** | **1.7 ±0.1** | **-0.005 ±0** | **-0.007 ±0.001** |
| 1993-1997 | summer | B | **-0.09 ±0.02** | **0.042 ±0.001** | 1.9 ±1.2 | 0.0 ±0.5 | **9.2 ±0.5** | **7.4 ±0.5** | **-0.0146 ±0.0015** | **12.2 ±1** | **-0.070 ±0.006** | **-0.110 ±0.009** |
| | | C.D.E | **0.60 ±0.01** | **0.004 ±0.001** | 0.8 ±0.9 | **0.6 ±0.3** | **4.0 ±0.3** | **4.0 ±0.3** | **-0.0166 ±0.001** | **14.1 ±0.8** | **-0.024 ±0.004** | **-0.042 ±0.007** |
| | winter | B | **-0.19 ±0.02** | **0.045 ±0.001** | 2.2 ±1.7 | 0.2 ±0.6 | 0.1 ±0.6 | -2.0 ±0.6 | 0.0076 ±0.0024 | -7.4 ±2 | 0.017 ±0.008 | 0.028 ±0.013 |
| | | C.D.E | 0.00 ±0.01 | **-0.018 ±0.001** | -0.9 ±0.9 | -0.1 ±0.3 | -1.0 ±0.3 | -0.2 ±0.3 | 0.0006 ±0.0012 | -0.7 ±1 | 0.001 ±0.004 | 0.002 ±0.006 |
| 2001-2007 | summer | B | **0.04 ±0.01** | **-0.027 ±0** | **-3.9 ±0.2** | **-2.7 ±0.2** | **1.3 ±0.2** | **2.6 ±0.2** | **-0.0105 ±0.0008** | **8.4 ±0.6** | **-0.050 ±0.003** | **-0.079 ±0.005** |
| | | C. D. E | -0.01 ±0 | **0.013 ±0** | **-1.3 ±0.1** | **-1.9 ±0.1** | **5.0 ±0.1** | **4.5 ±0.1** | **-0.0133 ±0.0005** | **11.8 ±0.4** | **-0.060 ±0.002** | **-0.095 ±0.003** |
| | winter | B | **0.04 ±0.01** | **-0.019 ±0.001** | -0.2 ±0.5 | 0.7 ±0.5 | **2.1 ±0.5** | **3.0 ±0.5** | -0.0061 ±0.0018 | 6.2 ±1.5 | -0.019 ±0.006 | -0.031 ±0.01 |
| | | C. D. E | **-0.05 ±0.01** | -0.002 ±0 | **-1.9 ±0.2** | **-1.8 ±0.2** | **1.7 ±0.2** | **1.8 ±0.2** | **-0.0076 ±0.0007** | **7.2 ±0.7** | **-0.033 ±0.003** | **-0.052 ±0.004** |
| 2008-2017 | summer | B | **-0.09 ±0** | **-0.013 ±0** | **1.4 ±0.1** | **2.0 ±0.1** | **-0.4 ±0.1** | 0.1 ±0.1 | **0.0052 ±0.0004** | **-4.4 ±0.3** | **0.017 ±0.002** | **0.028 ±0.003** |
| | | C. D. E | **-0.11 ±0** | **-0.013 ±0** | **1.7 ±0.1** | **2.3 ±0.1** | **1.2 ±0.1** | **1.8 ±0.1** | **0.0026 ±0.0003** | **-2.3 ±0.2** | **0.005 ±0.001** | **0.008 ±0.002** |
| | winter | B | **0.02 ±0.01** | **-0.004 ±0** | **0.4 ±0.2** | **0.6 ±0.2** | **1.3 ±0.2** | **1.5 ±0.2** | **-0.0024 ±0.0008** | **2.5 ±0.7** | -0.007 ±0.002 | -0.011 ±0.004 |
| | | C. D. E | **-0.05 ±0** | **-0.006 ±0** | **0.3 ±0.1** | **0.6 ±0.1** | **1.0 ±0.1** | **1.3 ±0.1** | -0.0009 ±0.0004 | 0.9 ±0.3 | **-0.006 ±0.001** | **-0.010 ±0.002** |

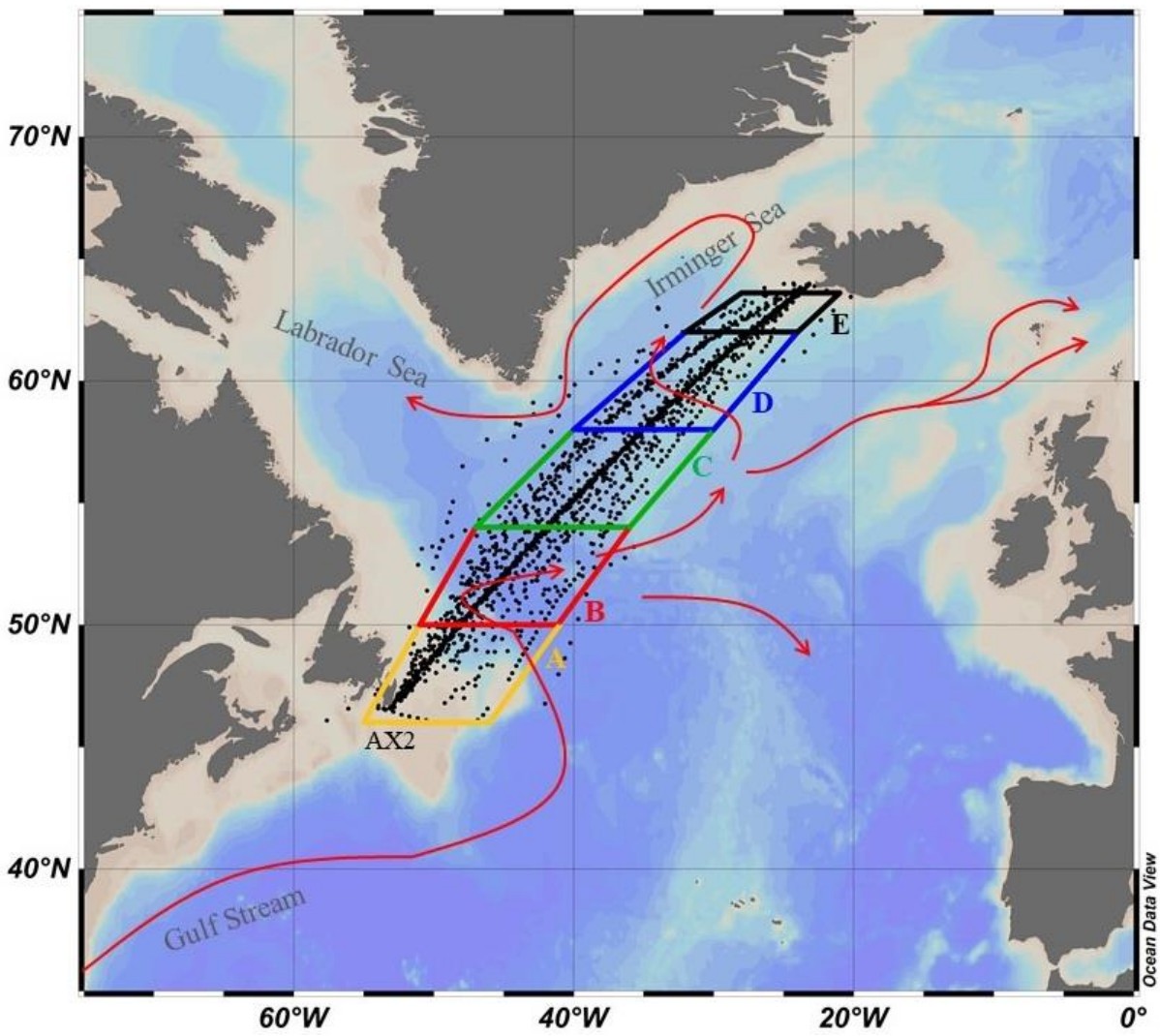

**Figure 1.** SURATLANT cruises track (in black, TA/DIC data over the period 1993 2017 with the fives boxes [BOX A: 46°N-50°N (in yellow), Box B: 50°N-54°N (in red), Box C: 54°N-58°N in green), Box D: 58°N-62°N (in blue), Box E: 62°N-64°N (in black)] and surface currents in the North Atlantic Subpolar Gyre (NASPG) circulation resulting from the Gulf Stream and the North Atlantic Current (in red).

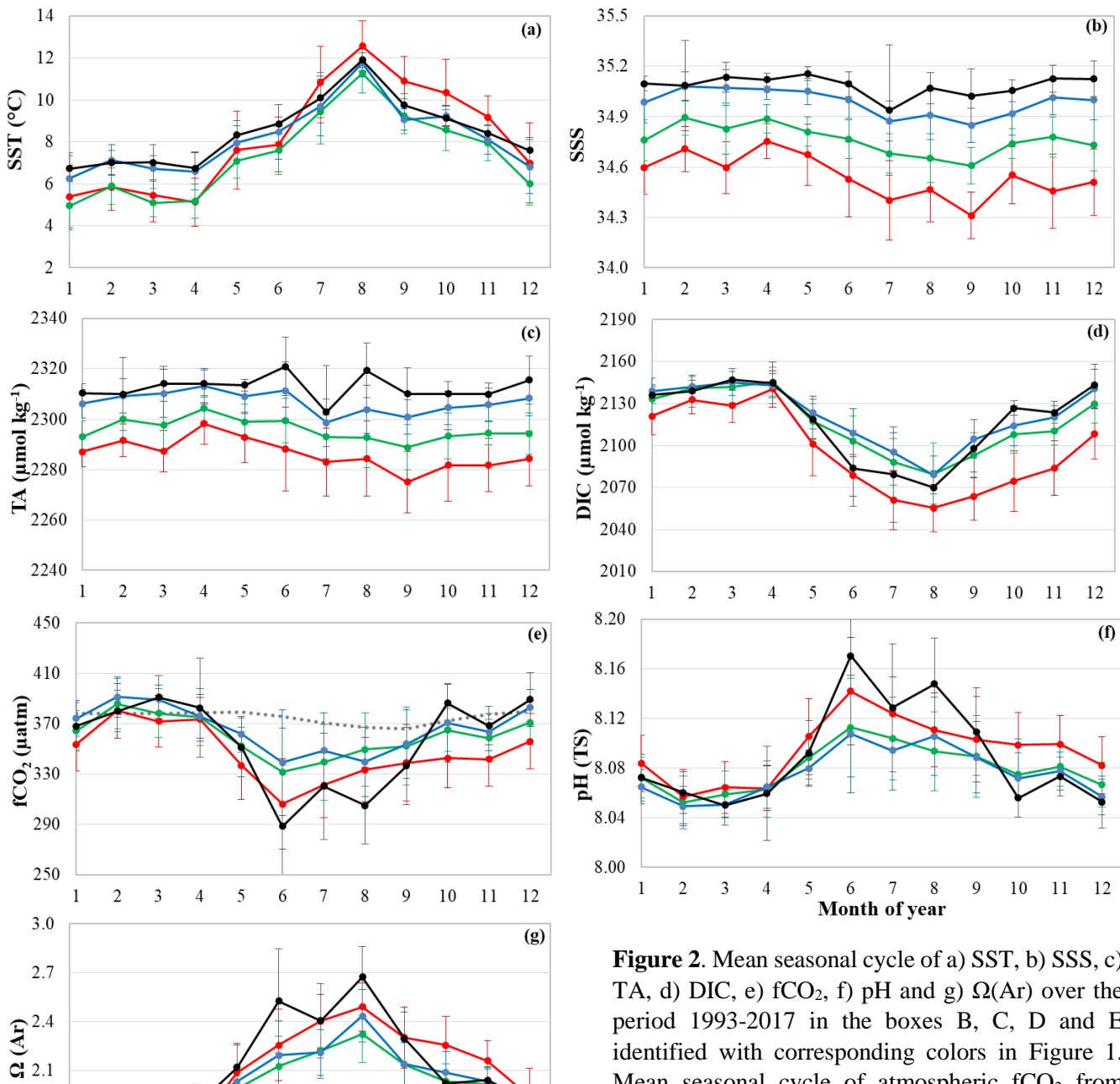

**Figure 2**. Mean seasonal cycle of a) SST, b) SSS, c) TA, d) DIC, e) $fCO_2$, f) pH and g) $\Omega$(Ar) over the period 1993-2017 in the boxes B, C, D and E identified with corresponding colors in Figure 1. Mean seasonal cycle of atmospheric $fCO_2$ from Mace Head station is represented by grey dotted in plot (e). Errors bars result from both interannual and spatial variability within a box.

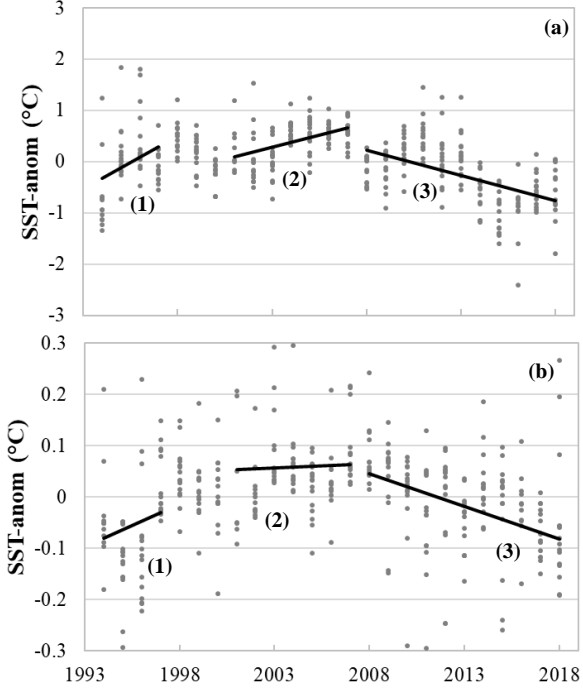

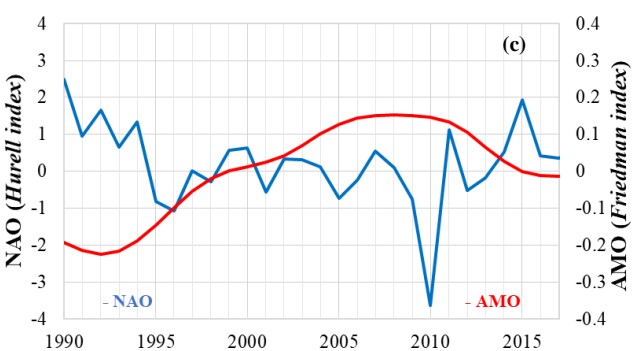

**Figure 3**. SST (a) and SSS (b) anomalies for winter (jan-feb) along Suratlant track (north of 50°N), based on the Binned products constructed by Reverdin et al. (2018a) and updated data to 2018, available at https://doi.org/10.6096/SSS-BIN-NASG, last access 25/03/2020). The mean trends for each period selected for the carbonate data analysis (when available) are represented in black lines (1: 1993-1997, 2: 2001-2007, 3: 2008-2017). The NAO index (in blue, Hurrell data available at https://climatedataguide.ucar.edu/climate-data/hurrell-north-atlantic-oscillation-nao-index-station-based, last access 30/09/2019) and AMO index (in red, Friedman ERSST V5 data available at https://www.ncdc.noaa.gov/data-access/marineocean-data/extended-reconstructed-sea-surface-temperature-ersst-v5, last access 20/11/2019) are represented (c).

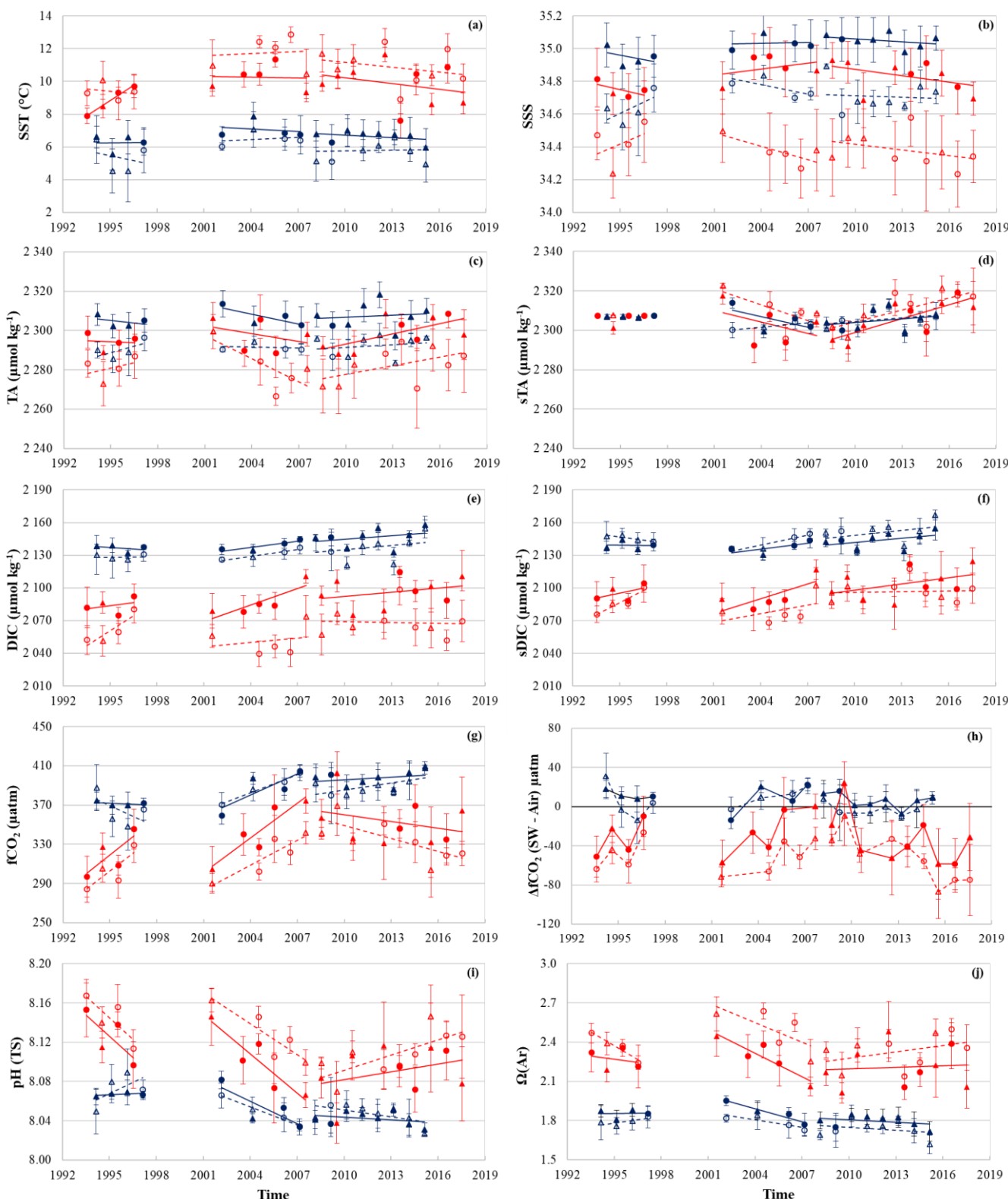

**Figure 4**. Evolution of a) SST, b) SSS, c) TA, d) sTA, e) DIC, f) sDIC, g) fCO₂, h) ΔfCO₂, i) pH and j) $\Omega_{Ar}$ between 1993 and 2017, are obtained in box B (open symbols) and by combining all data in boxes C, D, E (filled symbols) during summer (in red) and winter (in blue). Data from February and July are indicated with circles and the reconstructed data are depicted with triangles.

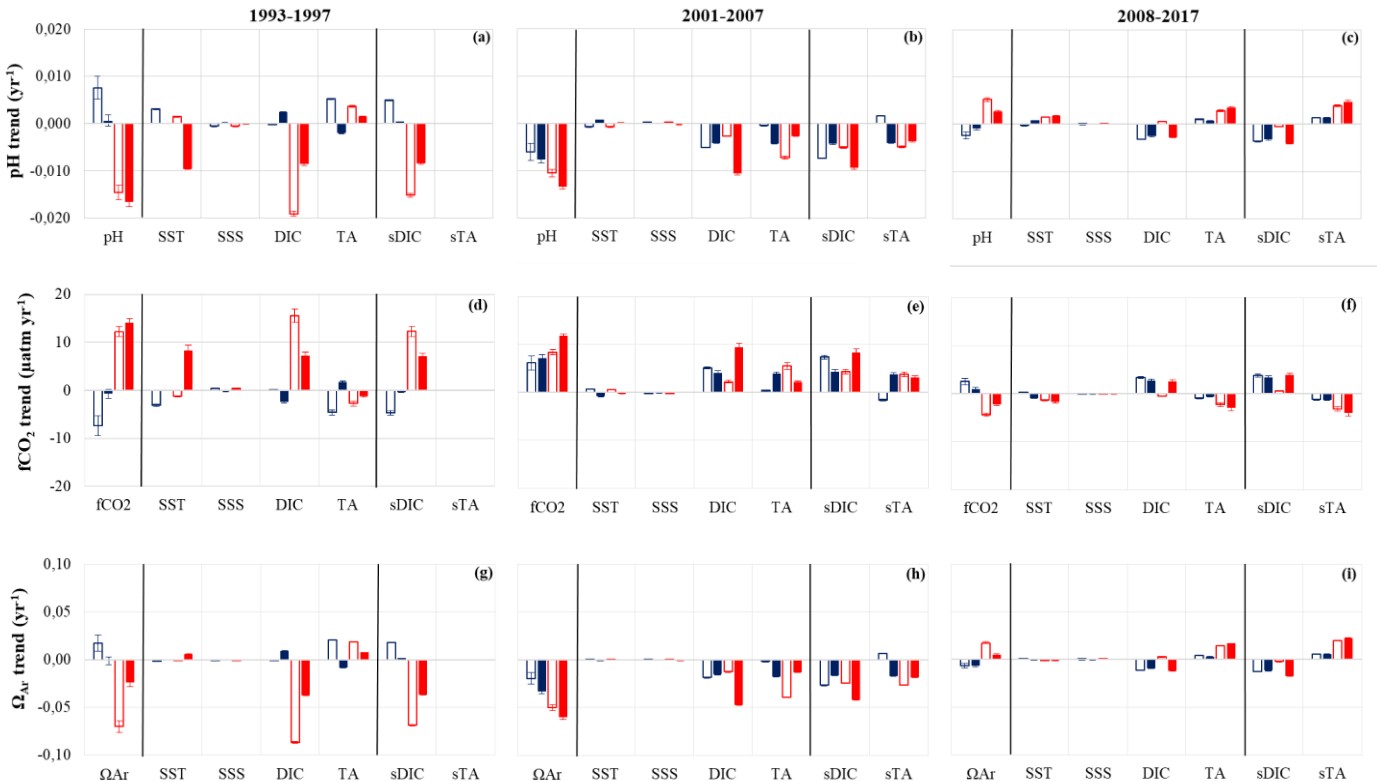

**Figure 5**. Decomposition of the trends in surface pH (a-c), fCO$_2$ (d-f) and $\Omega_{Ar}$ (g-i) according to equation (2). The effect of the changes in SSS, SST, DIC, TA and the normalized part (sDIC and sTA) is shown for the three periods: 1993-1997 (left column), 2001-2007 (middle column) and 2008-2017 (right column). Color coding is the same as in Figure 4. Here, DIC (or TA) represent the total carbon (or total alkalinity) part, whereas sDIC (or sTA) is the part of DIC (or TA) driver not related to salinity (see explanations for Eq. 2).

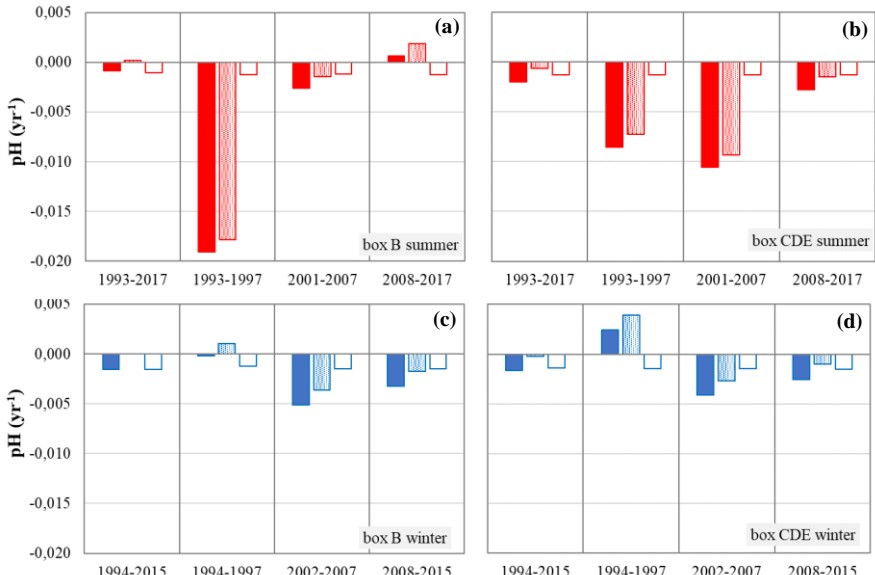

**Figure 6**. Decomposition of the trends in surface pH according to the effect of the changes in DIC-tot (filled), DIC-nat (dotted) and C-ant (open) during summer (a,b) and winter (c,d), for the long term period and the three periods.