# Peer review of "Ocean carbonate system variability in the North Atlantic Subpolar surface water (1993-2017)"

_Biogeosciences, 2019_

## Referee Comment (RC1) · Anonymous Referee #1 · 18 Apr 2019

Reference: bg-2019-119 Ocean carbonate system variability in the North Atlantic Subpolar surface water (1993-2017) Coraline Leseurre, Claire Lo Monaco, Gilles Reverdin, Nicolas Metzl, Jonathan Fin, Solveig Olafsdottir and Virginie Racapé.

General Comments This short manuscript describes the different trends in $CO_2$ fugacity (fCO2) in the North Atlantic Subpolar Gyre (50°N-64°N) considering three different periods based on the observations of the long-term monitoring program SURATLANT. Reverdin et al. (2018) have previously described these observations in an article in ESSD. Changes in pH and aragonite saturation, variables quasilinear dependent on $CO_2$ fugacity, are also described. As shown in the manuscript itself (p.2 l.25) is this an extension of the previous analyses made by Corbière et al., 2007; Metzl et al., 2010, there was practically nothing new. The new data given in figure 3 only represents 1/3

of the complete serie. A large part of the data and the half of figures come from the article by Reverdin et al. (2018). The description of the seasonal cycles shown by Reverdin et al. (2018) are different from those shown here without showing the reason for it. This puts in doubt that the interannual changes given for winter and summer cannot be affected by a poor quantification of the annual cycle in part due to low temporal coverage that can generate aliasing problems. The manuscript tries to describe the main drivers associated with CO2 chemistry using the same methodology shown in Metzl et al. (2010) for fCO2 and also García-Ibañez et al (2016) for pH. However, the relationship between the drivers and the main processes occurring in North Subpolar gyre is unfortunately very poorly developed. Partly because the authors seem to be unaware of key articles that have demonstrated the main patterns of variation linked to the NAO (Thomas et al. 2008, Keller et al. 2012, Schuster et al. 2013 and Pérez et al. 2013). In terms of acidification rates, the article by Garcia-Ibañez et al. (2016) is also ignored. Very imprecise in the description of the processes involved, mixing anthropogenic factor with natural processes without a clear target. The manuscript cites the well-known key processes of water mass transformations. Besides, the drivers are disjointed way with any change with the water column chemistry. It continuously mixes the anthropogenic and non-anthropogenic changes or flows of CO2. Methodology poorly described. For interannual estimates the seasonal variability is not eliminated, which calls into question whether rates of change can be affected by changes in the annual cycle or biases in the relatively low frequency of observations.

Specific remarks P.1 Line 25 "As a consequence, the future evolution of air-sea CO2 fluxes, pH and the saturation state of surface waters with regards to aragonite and calcite remain highly uncertain in this region". This is a very weak point of the manuscript, since despite analyzing the drivers does not allow them to make future evolutions. P.2 Lines 3-4 " covering 5% of the global surface ocean, is responsible for 20% of the oceanic uptake of anthropogenic CO2 (Khatiwala et al., 2013), with a mean annual air-sea CO2 flux estimated at 0.27 PgC/yr (Takahashi et al., 2009)." This is misleading. North Atlantic accumulates 20% anthropogenic CO2. The uptake can be produce in

the subtropical regions and transported northwards. The rate given by Takahashi et al. 2009 included a big component of natural CO2 mostly due to the cooling of northward advected subtropical water and by biological carbon fixation in the subpolar gyre (Thomas et al. 2008; Perez et al. 2013) P.5 Line 25 "The seasonal changes in fCO2 and pH are anticorrelated" This a consequence of the marine carbonic system when the alkalinity variability is so low as occurs in the North Atlantic. P.6 Line 27 "but it is due to a large increase in DIC rather than warming (Table 2), and as a consequence, it is accompanied by a rapid decrease in $\Omega$" Why? Reduction of the vertical winter missing by cooling typical of the Irminger? P.6 Line 30 "a need to further investigate the drivers of TA variability, which seem partially decoupled from surface" : However, Reverdin et al. 2018 show a perfect linear regression between TA and salinity, and the manuscript used this relationship to fill the gap of many observations without a second carbonic system variable. P.8 Line 20 "saturation with respect to calcium carbonate ($\Omega$)". This is a speculative addendum given the strong uncertainties and possible aliasing due to the seasonal coverture of the data. Fig 1 and Fig 2 come from Reverdin et al. 2018.

References : Thomas, et al. 2008. Changes in the North Atlantic Oscillation influence CO2 uptake in the North Atlantic over the past 2 decades, Global Biogeochem. Cycles, 22, GB4027, doi:10.1029/2007GB003167 Keller et al. 2012 Variability of the ocean carbon cycle in response to the North Atlantic Oscillation. Tellus B 2012, 64, 18738, http://dx.doi.org/10.3402/tellusb.v64i0.18738 Schuster et al. 2013 An assessment of the Atlantic and Arctic sea–air CO2 fluxes, 1990–2009. Biogeosciences, 10, 607-627, https://doi.org/10.5194/bg-10-607-2013. Perez et al. 2013, Atlantic Ocean CO2 uptake reduced by weakening of the meridional overturning circulation Nature Geoscience, 6 doi: 10.1038/ngeo1680 García-Ibáñez et al. 2016 Ocean acidification in the subpolar North Atlantic: rates and mechanisms controlling pH changes . Biogeosciences, 13, 3701–3715, 2016, doi:10.5194/bg-13-3701-2016

---

## Referee Comment (RC2) · Are Olsen (Referee) · 25 Apr 2019

This contribution presents the trends in a very impressive set of data from the western subpolar North Atlantic, collected within the framework of the SURATLANT program. While the paper presents the data and trends more or less adequately, it fails in properly attributing them to (climatic + oceanographic + biological) drivers. These aspects are unclear, speculative, and somewhat confused. As such, major revisions are required.

My major comments are:

1. Samples are collected two to four times a year. My sense is that with such rare sampling one might be very prone to getting false trends because of differences in the timing of the data collection, also in combination with timing of the spring bloom. For

example, the strong increase in region C+D+E summer DIC in 2001 -2008 might be generated by earlier sampling/later spring bloom.

While some effort seems to have been taken to deal with this; reconstructed data; it is not quite clear how this is done, and how well it works. Some improvement is needed in this description (page 5 lines 1-19). Is it possible to test this scheme by using data from a year with sampling in all of the months used (e.g. Jan-March)? At least for SST and SSS this can be done as there should be continuous TSG data available, and for fCO2 there are data available from the VOSs Nuka Arctica and Atlantic Companion, which crosses the study area. These can be retrieved from SOCAT and used to test the method.

2. More broadly, it would be interesting to know how representative the data are for large-scale interannual phenomena. This can be tested. For example, the trends in SST can be compared with objectively analysed SST from NOAA. The trends in SSS can be compared with some renanalysis model output. And again, for the time representativeness, the continuous data from the TSG, which collects data on all crossings, should be used. It might also be worthwhile to look at remotely sensed Chl a, to evaluate if there are concominant trends in surface ocean primary production, e.g. a loss in production from 2001 -2008 such as the DIC data seem to indicate and was suggested in the manuscript.

3. The attribution section is not well done. In particular, I strongly suggest the authors to explicitly account for salinity changes in the pCO2 driver decomposition following the method by Keeling et al. (2004) and recently used in the subpolar North Atlantic by Fröb et al. (2019). The reason is that dillution may completely overwhelm increase of DIC expected from uptake of anthropogenic CO2.

4. Further, showing trends in salinity normalised DIC and TA is worthwhile, but make sure to use the correct method for salinity normalising as described in Friis et al. (2003).

5. Also, there is a lot of mentioning of the change in the air-sea CO2 difference. But this

is not illustrated, which leaves a lot to the readers imagination. I therefore recommend to show the actual air-sea fCO2 difference in Fig. 3. You can make room for this by removing one of the panels for Omega (Calcite/Aragonite), as there is no need to show both.

6. Uncertainties are not properly dealt with. For example, the uncertainty in the SST of 0.1 degrees C results in an fCO2 uncertainty of 2 microatmospheres. There are also uncertainties in DIC and TA. These errors need to be propagated to calculated fCO2, pH etc. This can be done using the most recent CO2SYS from James Orr as this includes error propagation. It is available from GitHub. The errors in fCO2 (and the others) can be propagated to the trends using Monte Carlo.

7. I am in particular concerned with the fact that the large summer fCO2 increase in region B 2001-2008 are basically caused by the 'reconstructed' data of 2001 and 2008. The actual observations are pretty steady. How confident are you in these reconstructed values?

8. As the other reviewer, I think this contribution confuses anthropogenic and natural CO2. The trends that are observed in pCO2 does not say any on anthropogenic carbon uptake. Further, the North Atlantic is not a big sink of anthropogenic CO2 because the air-sea flux is large. The air-sea flux is a combination of natural and anthropogenic CO2 fluxes. Horizontal advection is likely a big source of anthropogenic CO2 to the North Atlantic. Hence, the data that are presented only informs about the changes in surface pCO2 and air-sea CO2 flux, not on the North Atlantic sink for anthropogenic CO2. This needs to be considered both in the introduction and in the discussion.

9. I think the dicussion is pretty dissapointing. It is basically a recap of the results + some more exploration of these, combined with some speculation based on published litterature. There are no attempts to analyse the relation between the observed trends and likely drivers (for which data exist) – such as NAO or AMV indices, winter/summer mixed layer depths (Argo data), primary production (remotely sensed ocean color), and

SGP strenght (SPG index). This should be done.

10. Finally, Metzl et al (2010) suggested deep mixing as the cause for the sharp winter fCO2 trend in 2001-2008. Fröb et al (2019) demonstrated unequivocally that deep mixing leads to strong increases in winter fCO2 in NAO positive years because of intensified deep mixing (bringing remineralised DIC to the surface). In light of this, it is interesting that the smallest winter fCO2 trends from SURATLANT are observed in the period 2008-2017, as it is well known that this has been a period of rather large deep mixing compared to 2001-2008. I am not sure how this can be reconciled, i.e. the large fCO2 trend in the period of little deep mixing 2001-2008 vs the small fCO2 trend in the period of frequent deep mixing in 2008-2017. It seems worthwhile to dvelve into this. While doing so, keep in mind that deep-mixing events seemed (in Fröb et al., 2019) to cause year-to-year anomalies and not so much an anomalous trend. Therefore, I recommend to reconsider the use of three periods - the trend in each of these can be strongly affected by the fCO2 in the start and/or end year, which might just be an anomalous year. Therefore the authors might be doing themselves a disfavor by sticking to the three periods, which are largely defined because of 'historical'/'traditional' reasons related to sampling and previous SURATLANT papers.

Other comments:

Page 2, line 4. Takahashi's estimates of air-sea CO2 flux cannot be equated to anthropogenic CO2 flux.

Page 2, line 6. I don't think any has ascribed the variations in the North Atlantic CO2 sink to climate change. Careful with such statements.

Page 2, line 21. 'based on Total Alkalinity (TA) and Dissolved Inorganic Carbon (DIC) observations,..'

Page 3, line 7. 'which', not 'witch'.

Page 3, line 14-15. 'to this end', not 'to this aim'.

Page 3, line 15. 'Iceland', not 'Island'

Page 3, line 17. These acronyms have already been defined.

Page 4, equation 1. I think it would be useful to recap the accuracy of this relationship, and how well it works for the data that are presented here.

Page 4. Atmospheric fCO2 calculation. What atmospheric pressure was used?

Page 4, decomposition equation. As mentioned above, please use the method that explicitly accounts for the effects of dillution/concentration. Note also, that the equation as written is wrong. The dX/dt term should be dX/dz, where z is the driver in question. You further need to explain what values you used for these sensitivities, and how they were derived.

Page 5, line 5. How many times were data excluded from box B because of SSS being outside the 34-35 range?

Page 5. Also, please consider the grouping of these regions for the trend analysis From Figure 2 SSS and TA, E and D appears quite similar while C and B both appear different. My sense this that combining E and D, while keeping C and B seperate might be the best approach.

Page 5, lines 10-19. Please collect this information in one section. Please also write that these are the reconstructed values in Fig. 3 (If I understand correctly the e.g Jan/March values adjusted to February are the 'reconstructed' values in Fig 3).

Page 6, lines 2-3. As mentioned above I am not convinced that these are 'pluriannual' trends. You might be doing yourself a disfavor by splitting the data into three time periods. It might make more sense to look at the timeseries as a whole and instead look at anomalies from the long-term trend. In particular this should be done from 2001 onwards. Consider to relate anomalies to mixed layer depths similar to Fröb et al. (2019).

Page 6, lines 7 onwards. Here a panel showing air-sea fCO2 difference, as suggested above would help, as it will make the discusssion more quantitative.

Page 6, line 20. I think 'near-stagnation' is the wrong word here. In winter the increase appears significant, at 1 uatm/yr. Generally, please make the summary of the results quantative.

Page 7, line 11. Remove 'Thereafter'

Page 7, line 15-16. Enclose 'much faster than the atmospheric signal', with commas. And... 'suggest larger productivity in the beginning of the period than at the end'. BTW this can and should be checked with Chl a data.

Page 7, line 19, replace 'than' with 'to'

Page 7, line 21-22. It is interesting that the slower increase in fCO2 is associated with strengthening of the winds and enhanced deep mixing. As noted above, Metzl et al., (2010) suggested deep mixing as the cause for the larger increase in the 2001-2008 period. Recently Fröb et al. (2019) found anomously high fCO2 during years of deep mixing. What is suggested here, is thus at odds with these papers. This needs to be explored or revised.

Page 7, line 25. Set the '2' as subscript.

Page 8, line 9 -10. The link between the trends in the carbon system and NAO+AMV that are described here are not backed up with any statistics. It comes across as very speculative. The statements needs to be backed up with for example correlation analyses.

Page 8, line 11. Fröb et al (2018) shows in particular a large increase of anthropogenic DIC inventory in deep mixing years, and tendencies for a loss of natural carbon. Fröb et al. (2019) shows a outgassing of CO2 during deep mixing years. The coupling between the inventory changes and the variability in fluxes has yet to be made. Some discussion around this would be interesting.

Page 9, line 5. 'Makes it difficult to predict the evolution of CO2 uptake..' I suggest to read the paper by Li et at 2016, on decadal predictions of North Atlantic CO2 uptake, this might provide some relevant information.

References (not cited in the paper) Fröb, F., Olsen, A., Becker, M., Chafik, L., Johannessen, T., Reverdin, G., and Omar, A.: Wintertime fCO(2) Variability in the Subpolar North Atlantic Since 2004, Geophys Res Lett, 46, 1580-1590, 2019.

Li, H. M., Ilyina, T., Muller, W. A., and Sienz, F.: Decadal predictions of the North Atlantic CO2 uptake, Nat Commun, 7, 2016.

---

## Author Comment (AC1) · 23 Jun 2019

Leseurre, C., Lo Monaco, C., Reverdin, G., Metzl, N., Fin, J., Olafsdottir, S., and Racapé, V.: Ocean carbonate system variability in the North Atlantic Subpolar surface water (1993–2017), Biogeosciences Discuss., https://doi.org/10.5194/bg-2019-119, in review, 2019.

Review 1 (anonymous):

We thank the reviewer for her/his fast review, comments and questions that will be taken into account when revising the manuscript. Below we list our responses before preparing a revised manuscript.

"General Comments: This short manuscript describes the different trends in CO2 fugacity (fCO2) in the North Atlantic Subpolar Gyre (50°N-64°N) considering three different periods based on the observations of the long-term monitoring program SURAT-LANT. Reverdin et al. (2018) have previously described these observations in an article in ESSD. Changes in pH and aragonite saturation, variables quasilinear dependent on CO2 fugacity, are also described. As shown in the manuscript itself (p.2 l.25) is this an extension of the previous analyses made by Corbière et al., 2007; Metzl et al., 2010, there was practically nothing new. The new data given in figure 3 only represents 1/3 of the complete serie. A large part of the data and the half of figures come from the article by Reverdin et al. (2018)".

Response:

The reviewer seems disappointed with the new data and results that would offer "practically nothing new" regarding previous work. In previous studies we used data from 1993-2003 (Corbière et al., 2007) to investigate for the first time the interannual to decadal changes observed in the NASPG with the SURATLANT data. This first analysis was then complemented by Metzl et al. (2010) with new data (2004-2008) that present significant variations compared to 1993-2003 with a winter focus in order to avoid to potential biases due to biological activity in the summer season. In particular, Metzl et al. (2010) indicate very rapid change of fCO2 (up to 7 $\mu$atm/yr in 2001-2008) highlighting "the need for continued long‐term sea surface ocean observations of carbon properties (DIC, TA and fCO2)". The SURATLANT data for 1993-2007 (or 1993-2010) were also associated with other observations to better evaluate pCO2 and air-sea flux variability at regional and larger scale in the North Atlantic (Schuster et al 2009; 2013; Watson et al., 2009; Mc Kinley et al 2011; Fay and Mc Kinley, 2013) or to describe the seasonality of sea surface $\delta$13CDIC (Racapé et al., 2014).

In the present study, we added ten years of data (2008-2017) which offers new and complementary results. The recent paper by Reverdin et al (ESSD, 2018a) aimed at describing in detail the methods, accuracy and data for the full period 1993-2017 (with some data revisited and corrected), including some preliminary results of seasonal cycles and long-term trends for a few properties; however, especially for ESSD Journal, Reverdin et al (2018a) did not investigate internal processes or external forcing resulting in the observed temporal changes of the properties (SST, SSS, nutrients, TA and DIC, $\delta$18O and $\delta$13C). This ESSD publication was accompanied with files of the data publicly available (at SeaNoe and OCADS). The present paper is aimed at better understanding the changes in the carbonate system over the whole period (1993-2017) and in different regions and periods, including summer-time (not evaluated in Metzl et al., 2010) as well as pH, Omega trends not described in previous work. Concerning the figures, as we used the same Box definition selected by Reverdin et al (2018a), we used the same figures for simplicity. In the revised manuscript we will prepare a new single figure (probably showing all data collection for DIC/TA, boxes boundaries and the main circulation.

"The description of the seasonal cycles shown by Reverdin et al. (2018) are different from those shown here without showing the reason for it. This puts in doubt that the interannual changes given for winter and summer cannot be affected by a poor quantification of the annual cycle in part due to low temporal coverage that can generate aliasing problems".

Response:

In Reverdin et al (2018a) the seasonal cycles were presented for all Boxes (including the southern box A) and for salinity normalized DIC and TA. Here, we have chosen to present in more detail the seasonality for boxes B, C, D, E and for all properties related to the carbonate system (including fCO2, pH and Omega not shown in Reverdin et al 2018a) and properties involved in the analysis of trends and drivers (SST, SSS, DIC, TA). For TA and DIC we show the mean seasonal cycle but not normalized as in Reverdin et al. Also, as we have no observations for all years in February (winter) or July (summer), the climatological seasonal cycles are used to group data of different months (such as JFM and JJA) in order to best estimate the seasonal trend (e.g. projecting observed January anomaly in February). Thus, we briefly introduce and describe the seasonality in section 4.1. The figure is required to indicate the seasonal variability that we have to correct for when combining data of different months. This is not so well known for the carbonate properties (model evaluations used for interannual to decadal analysis do not seem yet to correctly represent the processes involved in its seasonality, Pilcher et al., 2015, Mc Kinley et al., 2017). In addition, the seasonal cycles presented for the boxes B, C, D, and E contributed to decide to merge boxes C, D, E for the trend analysis. Of course, we fully agree with the reviewer that the seasonal cycle is not the main aim of the study, which is the estimation of the trends and the analysis of key drivers that is discussed in the core of the paper.

"The manuscript tries to describe the main drivers associated with CO2 chemistry using the same methodology shown in Metzl et al. (2010) for fCO2 and also García-Ibañez et al (2016) for pH. However, the relationship between the drivers and the main processes occurring in North Subpolar gyre is unfortunately very poorly developed. Partly because the authors seem to be unaware of key articles that have demonstrated the main patterns of variation linked to the NAO (Thomas et al. 2008, Keller et al. 2012, Schuster et al. 2013 and Pérez et al. 2013). In terms of acidification rates, the article by Garcia-Ibañez et al. (2016) is also ignored".

Thank you for recalling these references and suggestion to add a more in-depth discussion on the link with NAO (also suggested by reviewer 2).

Response about articles:

We were aware of these studies as they used in part SURATLANT data for 1993-2004 (or 1993-2005) to validate ocean models or reconstructed pCO2 fields (Thomas et al 2008; Ullman et al., 2009; Keller et al., 2012; Signorini et al., 2012; Rodenbeck et al., 2013, 2014). The SURATLANT data for 1993-2005 or 1993-2007 (fCO2 calculated from TA/DIC) were also used to complement fCO2 underway observations in the North Atlantic to better evaluate seasonal and decadal variations of pCO2 and air-sea CO2 fluxes in this region (Watson et al., 2009; Schuster et al 2009, 2013; Mc Kinley et

al. 2011; Fay and Mc Kinley, 2013). We will add some of these references (including Pérez et al., 2010, 2013; Garcia-Ibanez et al., 2016) in the introduction as well as in the discussion of the results we obtained in the NASPG for the period 1993-2017.

Response about NAO:

NAO was previously recognized a possible link of the rapid fCO2 increase when NAO shifted from positive to negative phase in 1995-1996 (Corbière et al., 2007). However, this was not confirmed for the period 2001-2007 when NAO did not vary so much around neutral value (Metzl et al., 2010). A possible explanation is that the observed variations of fCO2 in the NASPG, especially rapid trends such as +7 $\mu$atm/yr observed in 2001-2007, are driven by superimposed processes linked to climatic signals such as NAO and AMO, and both should be taken into account, as well as other processes involved in the NASPG ocean circulation, ventilation and vertical mixing. AMO, based on SST, is a long-term multidecadal signal (sometimes better called AMV) that experienced a gradual progressive increase from negative values in the 70s to positive values in the early 2000s, and remains positive and relatively stable in 2002-2017. NAO based on sea level atmospheric pressure gradient, shows much shorter variability, with highs and lows occurring in its winter record at interannual to decadal periods. We think that a direct relationship of the variability of pCO2 or CO2 uptake in the North Atlantic with NOA is still ambiguous (e.g. Takahashi et al., 2009; Schuster et al., 2013; Mc Kinley et al. 2017) and the detection of pCO2 changes with climate variability is still challenging from observations (at least for the period we investigated 1993-2017). However, for long-term multi-decadal variability the link between AMO and pCO2 change in the NASPG appears more robust (Breeden and McKinley, 2016; Landschützer et al., 2019).

Summary of NAO and model results cited:

Ocean models can help to understand the link between NAO and biogeochemical cycles, but results from models are still controversial. Keller et al (2012) who investigated

several simulations (6 Earth systems models and for winter only, i.e. not the productive season), conclude that on-site entrainment in the subpolar gyre (mixing, upwelling) is the main driver of carbon sink variability as opposed to advection (as suggested by Thomas et al. 2008). In their model Thomas et al (2008) suggest that negative or neutral NAO conditions result in a substantial decline in $CO_2$ uptake for years 1997-2004 along the NAC and in the eastern subpolar gyre. However, in another modeling study of the North Atlantic, Ullman et al. (2009) conclude that the air-sea $CO_2$ flux increased over the 1992–2006 and this is due to the increasing atmospheric $CO_2$ and not to long-term variability in the physical climate or biogeochemistry. During the transition of NAO (from positive to neutral), their model simulates a decline of the convection and vertical DIC supply to the surface in the subpolar region, counteracting the increase of $pCO_2$ due to warming. This leads to a small net $pCO_2$ increase (compared to atmospheric trend) and increasing $CO_2$ sink. Interestingly in the NASPG, the DIC decrease of -0.75 $\mu$mol/kg/yr in winter is more pronounced in the model (Ullman et al., 2009) than in the SURATLANT data (for 1992-2005). That might explain why observations suggest a reduced $CO_2$ sink after the NAO shift in the mid-90s (Corbière et al., 2007), while the model suggest an increasing sink.

Keller et al., (2012) show that simulations with different coupled models lead to different results, and the response to NAO seems modest: typical NAO-driven variations at large-scale are +/- 10 $\mu$mol/kg for surface DIC and TA, and +/- 8 $\mu$atm for delta-$pCO_2$. Depending on the model, the change of $pCO_2$ in the subpolar region varies between +2/-4 $\mu$atm (for NAO+) and +16/-12 $\mu$atm (for NAO-), that is even the sign of the response is different between models. Such low variations, if real, are rather difficult to extract from observations and thus the results of the models difficult to validate. Thus, Keller et al (2012) conclude that although the interannual variability in the North Atlantic is largest in the subpolar gyre, the magnitude and responses of the carbon uptake to NAO significantly differ between the models (recall that this conclusion holds only for winter).

These model studies and their controversial results (level of variability and processes at play) show that there is still more work to be performed to understand the link between NAO and biogeochemistry. As a matter of fact, in a recent analysis on the link between NAO and biology in the North Atlantic, Mc Kinley et al. (2018) conclude that "nowhere is the NAO correlated with biomass variability and that more investigation of the links between North Atlantic climate and biomass variability is clearly warranted". The same is true for biogeochemistry and carbon cycle and we believe that the new data on the carbonate system we analyzed in our submitted paper along the SURATLANT line should offer new information on this issue, even if the main conclusion is that over 1993-2017 there is NO direct link detected between fCO2 and DIC trends with NAO in this region of the NASPG. Of course, at shorter time scale (1-2 years) we might recognize such a link as described below (and see also the large changes in regions with changes in convection, such as in the western Irminger Sea, Fröb et al., 2018).

Summary of observations with new data related or not with the NAO:

The update made for the last decade 2008-2017 (data not included in our previous work, Corbière et al., 2007; Metzl et al., 2010) adds observations obtained during a strong negative NAO in 2010 and a positive NAO phase in 2015.

The 2010 event was associated with a warming and freshening (and low density) found in both SURATLANT discrete sampling (in August 2010) and monthly reconstructed Binned products (Reverdin et al 2018b). In August 2010 we observed low DIC (and also high $\delta$13CDIC as discussed by Racapé et al 2014), a signal also revealed in high Chl-a concentrations (identified from MODIS data). However, as this was associated with a warming (observed positive SST anomaly up to +1.5 °C), the fCO2 (and pH) values were not very different from previous summers, illustrating the competitive effect of warming and higher production on fCO2 for this event. We also note that in summer 2010, DIC/TA was also sampled during the OVIDE-2010 cruise (in late June). In the NASPG, surface DIC concentrations for OVIDE-2010 were around 2070-2090 $\mu$mol/kg, i.e. just between SURATLANT data obtained in early June (2100-2110 $\mu$mol/kg) and in

mid-August (2050-2070 $\mu$mol/kg); for TA, concentrations were the same for all cruises. This confirms the summer 2010 anomaly (low DIC) apparently associated with higher productivity during the negative NAO phase, but with no significant impact on fCO2 and trends.

We also identified a cold year in 2015 (SST anomaly around -1 °C), when NAO was in a positive phase: for that year, DIC was higher in winter (near the maximum observed on average in winter in our time series) but again, as the temperature also lowers fCO2, the fCO2 value were not so different from other winters, now illustrating the competitive effect of cooling and deep mixing on fCO2.

With the new data introduced in this manuscript for 2008-2017 that corresponds to a period when NAO presents large IAV compared to 1995-2007, we observed more variability in the DIC data (and fCO2, pH) that might result in less clear trends over 5-10 years. We have identified two specific years (2010 and 2015) that experienced very low NAO (-3 in 2010) and high NAO (+2 in 2015), leading respectively to observed warming (cooling), freshening (saltier) and low (high) DIC; these anomalies could be explained by an increase in productivity in 2010 and deeper mixing in 2015. As these anomalies have been clearly recognized, we have tested the sensitivity of the trends analysis with and without these NAO events. Not surprisingly, for the 2015 anomaly and because we have no winter data after 2015, we derived significant different trends when 2015 is or not considered for the period 2008-2015. For example, for the northern boxes CDE the winter trends evaluated for years 2008-2015 were +1.1 $\mu$mol/kg/yr for DIC and 0.6 $\mu$atm/yr for fCO2. If we restrict to the period 2008-2014, trends become negative, i.e. -0.14 $\mu$mol/kg/yr for DIC and -0.8 $\mu$atm/yr for fCO2. We thus have to be careful when selecting (and interpreting) the periods. On the other hand, if we test the sensitivity of the trends for summer season in 2008-2017 (with or without the 2010 anomaly), results are basically the same. These specific results recall that fCO2 trends (and here also for DIC) are highly sensitive to the choice of starting and ending years as was illustrated by Mc Kinley et al (2011).

We also tested the impact of these NAO events (high and low) on the long-term trends, 1993-2017 for summer and 1994-2015 for winter. In that case, the results are more robust. For example, for the northern boxes CDE the winter trends evaluated for years 1994-2015 were +0.6 $\mu$mol/kg/yr for DIC and +1.4 $\mu$atm/yr for fCO2. If we restrict to the period 1994-2014, trends are lower, i.e. +0.5 $\mu$mol/kg/yr for DIC and +1.3 $\mu$atm/yr for fCO2. In both cases, the DIC trend appears close to the increase due to anthropogenic uptake (estimated around +0.6 to +0.8 $\mu$mol/kg/yr in this region; see our response to other reviewer comments below). For summer, the trend in 1993-2017 with or without the 2010 NAO anomaly leads to the same results: +1 $\mu$mol/kg/yr for DIC and +2 $\mu$atm/yr for fCO2. In that case, the NAO has no effect on the trends. For the full period, the summer trends appear faster than derived from winter data and this needs to be clarified. This is also why it is important to separate the full period in 3 parts (as presented in the submitted paper) to better investigate the process involved, especially with the cooling and freshening observed after 2008. If the NAO events are relatively well characterized with our observations and could impact the trends when evaluated over 5-10 years, for long-term trends (24 years) this seems a secondary effect (as also suggested from long-term ocean simulations, Breeden and McKinley 2016).

As a final test, we investigated the SOCAT data (version V6, Bakker et al, 2016) in this region corresponding to the NAO events. Unfortunately, there is no fCO2 data for July-August 2010 that could be used to support the low DIC concentrations we observed in August 2010, but also in late June 2010 during the OVIDE-2010 cruise. However, for 2015 (high NAO), SOCAT data suggest relatively high fCO2 around 390-415 $\mu$atm in Jan-Feb 2015 in the NASPG, not far from our average value of around 400 $\mu$atm. A comparison with SOCAT fCO2 data for the full period would be interesting but this is beyond the scope of our analysis mainly based on DIC/TA observations. This would be a topic for another publication shared with SOCAT data providers in the north Atlantic.

Based on our results and sensitivity analysis described above, we will introduce the NAO events in the revision and discuss how NAO may be linked (or not) to the CO2

trends. We thank again the reviewer to highlight this issue in her/his comment.

"Very imprecise in the description of the processes involved, mixing anthropogenic factor with natural processes without a clear target. The manuscript cites the well-known key processes of water mass transformations. Besides, the drivers are disjointed way with any change with the water column chemistry. It continuously mixes the anthropogenic and non-anthropogenic changes or flows of CO2. Methodology poorly described. For interannual estimates the seasonal variability is not eliminated, which calls into question whether rates of change can be affected by changes in the annual cycle or biases in the relatively low frequency of observations".

Response: Thank you to highlight the anthropogenic versus non-anthropogenic signal issue (same comment by reviewer 2, Are Olsen). As the DIC and pCO2 variability are large in surface waters, detection of anthropogenic CO2 (C-ant) is difficult (if time-series are limited) and data-based methods such as C0, TrOCA, delta-C*... not suitable to quantify C-ant in surface waters. Indeed, longer-term time-series are required to separate natural and anthropogenic (or climate induced) signals (e.g. at least 30 years in the North Atlantic subpolar gyre, McKinley et al, 2011). We have evaluated the anthropogenic concentrations (C-ant) in this region based on subsurface Glodap-V2 data (Olsen et al., 2016) and TrOCA method, but we did not introduce the results in the present paper as they correspond to a different period (and only for summer). However, in the revision, as recommended by reviewers, we will introduce the trends of C-ant evaluated in subsurface from different methods (our calculations and recently from Gruber et al., 2019). In short, based on Glodap-V2 data in the NASPG we estimate C-ant trend of +0.7 $\mu$mol/kg/yr at subsurface (150-200m) for the period 1997-2011. This would explain 75% of the DIC trend of +0.9 $\mu$mol/kg/yr observed at subsurface for the same period. On the other hand, using the new e(MLR*) method, Gruber et al (2019) evaluate accumulation of C-ant from 1994 to 2007 in the global ocean. In the North Atlantic and specifically along the SURATLANT line, the accumulated C-ant is +8.5 (+/- 1.7) $\mu$mol/kg in the layer 150-200m. This signal is rather homogeneous at depth

(150-200m) but with a small gradient between southern (6 $\mu$mol/kg) and northern (10 $\mu$mol/kg) NASPG regions. This would correspond to a trend of between +0.6 and +0.7 $\mu$mol/kg/yr, very close to our C-ant estimate based on Glodap-V2. If we assume that the subsurface C-ant trend results are correct and also valid for the surface layer, the e(MLR*) method lead to a very homogeneous C-ant accumulation in the layer 0-50m along the SURATLANT line (55N-64N), of +10.1 (+/- 0.8) $\mu$mol/kg between 1994 and 2007, i.e. about +0.8 $\mu$mol/kg/yr. Interestingly, this is in the range of the long-term (1993-2017) DIC surface trends that we report (table 2 in our paper) between 0.6 and 0.7 $\mu$mol/kg/yr in boxes B,C,D,E (depending the season). However, our observations also show that DIC trends could be very different for short periods and north/south regions (see all trends listed in Table 2), and occasionally DIC decreases over time as opposed to C-ant (e.g. 1993-1997 winter, this study; Metzl et al., 2010; Ullman et al 2009). Based on the C-ant estimates, we will also evaluate and discuss the trends of DIC-Cant (C-nat), i.e. the natural part of the signal in the revision.

Figure 1 and 2 (attached) show the anthropogenic and natural contributions of DIC to pH change. This overview is obtained with the C-ant set at 0.7 $\mu$mol / kg / yr (in each region, period, season). We note that the results presented in boxes C and D-E are substantially the same as when we group them in C-D-E.

Specific remarks:

"P.1 Line 25 "As a consequence, the future evolution of air-sea CO2 fluxes, pH and the saturation state of surface waters with regards to aragonite and calcite remain highly uncertain in this region". This is a very weak point of the manuscript, since despite analyzing the drivers does not allow them to make future evolutions".

Response: we agree that this was not very clear. Although we found significant trends over 24 years, the variability we observed between different periods suggest that one cannot extrapolate the results in the future. This will be revised accordingly.

"P.2 Lines 3-4 "covering 5% of the global surface ocean, is responsible for 20% of the

oceanic uptake of anthropogenic CO2 (Khatiwala et al., 2013), with a mean annual air-sea CO2 flux estimated at 0.27 PgC/yr (Takahashi et al., 2009)." This is misleading. North Atlantic accumulates 20% anthropogenic CO2. The uptake can be produced in the subtropical regions and transported northwards. The rate given by Takahashi et al. 2009 included a big component of natural CO2 mostly due to the cooling of northward advected subtropical water and by biological carbon fixation in the subpolar gyre (Thomas et al. 2008; Perez et al. 2013)".

Response: Reviewer is correct. Sentence will be revised: "The North Atlantic is one of the strongest ocean sinks for natural (Takahashi et al., 2009) and anthropogenic atmospheric CO2 (Sabine et al. 2004; Khatiwala et al., 2013)." At that stage in the introduction, we don't really need to specify number such as 0.27 PgC/yr (in our analysis we do not evaluate an integrated flux over the domain).

"P.5 Line 25 "The seasonal changes in fCO2 and pH are anticorrelated" This a consequence of the marine carbonic system when the alkalinity variability is so low as occurs in the North Atlantic".

Response: This is correct: Revised following: "As the alkalinity seasonality is low, the seasonal changes in fCO2 and pH are anti-correlated". "P.6 Line 27 "But it is due to a large increase in DIC rather than warming (Table 2), and as a consequence, it is accompanied by a rapid decrease in Omega. Why? Reduction of the vertical winter missing by cooling typical of the Irminger?"

Response: We are not sure to understand the reviewer question but maybe the reviewer was confused with numbers listed in the table (we apologize, there was a mistake in our original Table for CDE Omega-Ar trend, should read 0.005, not -0.069). In this section we show that in summer 1993-1997 (not in winter) the fCO2 increase is particularly fast (10 to 12 $\mu$atm/yr) in both regions, box B and CDE, but for a different reason: For CDE (north) this was related to a warming, whereas for box B (south) this was explained by DIC increase. Contrary to Box CDE, Omega decreases in box B

in summer 1993-1997. The interpretation is however limited to a short period 1993-1997. The DIC increase in Box B may be related to change in advection or productivity, but with data in hand we cannot separate these processes (and there were too few nutrients data for this period). We thus only conclude on the contrasting effect in the north (warming) and the south (DIC) to explain the rapid fCO2 trend in summer and the north/south trends in Omega. We will rewrite the section to clarify.

"P.6 Line 30 "a need to further investigate the drivers of TA variability, which seem partially decoupled from surface": However, Reverdin et al. 2018 show a perfect linear regression between TA and salinity, and the manuscript used this relationship to fill the gap of many observations without a second carbonic system variable".

Response: The reviewer is correct. We used the TA/S relation established by Reverdin et al (2018a) when TA data were not available in 1993-1997 and occasionally when TA data were flagged "doubt or bad" for other years. For a regional view (as described in Reverdin et al), the TA/S relation was first obtained for salinity above 34. These authors also note: "For the lower salinities found on the Newfoundland shelf, different sources of freshwater (from the Arctic or resulting from continental or sea ice melt inputs) contribute to deviations from the relation." For the southern region (box A and box B for some periods when salinity is low), one should be careful when using this relation. This is also why in this paper, we did not describe the long-term trend in the southern region (box A), which requires a specific analysis. In addition, one cannot exclude the impact of blooms of coccolithophorids observed in the NASPG as variability of such blooms may also change TA concentrations with resulting deviations from the TA/S regional relation. Recently, Loveday and Smyth (2018) show significant differences of such blooms over decades in the North Atlantic. An interesting topic would be to re-explore TA distribution and TA/S relations for different periods. This is why we recall that a specific study of TA spatio-temporal distribution and drivers of TA variability should be performed, but this is beyond the scope of the present analysis. We will rewrite the section.

[Figure]

"P.8 Line 20 "saturation with respect to calcium carbonate ()". This is a speculative addendum given the strong uncertainties and possible aliasing due to the seasonal coverture of the data".

Response: we agree that the future evolution of saturation is somewhat speculative. Based on the trends observed for different periods, one can get very different projections for future scenario and caution should be taken when using specific periods to extrapolate the results. Based on the observed trends we only suggest a range of years to reach undersaturation. We will revise this section.

"Fig 1 and Fig 2 come from Reverdin et al. 2018". Response: this is correct: as we used the same Box definition selected by Reverdin et al (2018), we used the same figures for simplicity. In the revised manuscript we will prepare a new figure (probably showing all data collection for DIC/TA, boxes boundaries and the main circulation).

References:

Thomas, et al. 2008. Changes in the North Atlantic Oscillation influence CO2 uptake in the North Atlantic over the past 2 decades, Global Biogeochem. Cycles, 22, GB4027, doi:10.1029/2007GB003167.

Keller et al. 2012 Variability of the ocean carbon cycle in response to the North Atlantic Oscillation. Tellus B 2012, 64, 18738,http://dx.doi.org/10.3402/tellusb.v64i0.18738

Schuster et al. 2013 An assessment of the Atlantic and Arctic sea–air CO2 fluxes, 1990–2009. Biogeosciences, 10, 607-627, https://doi.org/10.5194/bg-10-607-2013.

Perez et al. 2013, Atlantic Ocean CO2 uptake reduced by weakening of the meridional overturning circulation Nature Geoscience, 6, doi:10.1038/ngeo1680.

García-Ibáñez et al. 2016 Ocean acidification in the subpolar North Atlantic: rates and mechanisms controlling pH changes. Biogeosciences, 13, 3701–3715, 2016, doi:10.5194/bg-13-3701-2016

Reference added in the response: Bakker, D. C. E., Pfeil, B., Landa, C. S., Metzl, N., O'Brien, K. M., Olsen, A., et al, 2016: A multi-decade record of high-quality fCO2 data in version 3 of the Surface Ocean CO2 Atlas (SOCAT), Earth Syst. Sci. Data, 8, 383-413, doi:10.5194/essd-8-383-2016, 2016.

Breeden, M. L. and McKinley, G. A.: Climate impacts on multidecadal pCO2 variability in the North Atlantic: 1948–2009, Biogeosciences, 13, 3387-3396, https://doi.org/10.5194/bg-13-3387-2016, 2016.

Corbière, A., N. Metzl, G. Reverdin, C. Brunet and T. Takahashi, 2007. Interannual and decadal variability of the oceanic carbon sink in the North Atlantic subpolar gyre. Tellus B, Vol. 59, issue 2, 168-179, DOI: 10.1111/j.1600-0889.2006.00232.

Fay, A. R., and G. A. McKinley, 2013, Global trends in surface ocean pCO2 from in situ data, Global Biogeochem. Cycles, 27, doi:10.1002/gbc.20051.

Fröb, F., Olsen, A., Pérez, F. F., García-Ibáñez, M. I., Jeansson, E., Omar, A., and Lauvset, S. K., 2018, Inorganic carbon and water masses in the Irminger Sea since 1991, Biogeosciences, 15, 51-72, http://doi.org/10.5194/bg-15-51-2018.

Fröb, F., Olsen, A., Becker, M., Chafik, L., Johannessen, T., Reverdin, G., and Omar, A.: Wintertime fCO2 Variability in the Subpolar North Atlantic Since 2004, Geophys Res Lett, 46, 1580-1590, 2019.

García-Ibáñez, M. I., Zunino, P., Fröb, F., Carracedo, L. I., Ríos, A. F., Mercier, H., Olsen, A., and Pérez, F. F.: Ocean acidification in the subpolar North Atlantic: rates and mechanisms controlling pH changes, Biogeosciences, 13, 3701-3715, https://doi.org/10.5194/bg-13-3701-2016, 2016.

Gruber, N., D. Clement, B. R. Carter, R. A. Feely, S. van Heuven, M. Hoppema, M. Ishii, R. M. Key, A. Kozyr, S. K. Lauvset, C. Lo Monaco, J. T. Mathis, A. Murata, A. Olsen, F. F. Perez, C. L. Sabine, T. Tanhua, and R. Wanninkhof (2019). The oceanic sink for anthropogenic CO2 from 1994 to 2007, Science vol. 363 (issue 6432), pp. 1193-1199.

DOI: 10.1126/science.aau5153.

Keller K., F. Joos, C. Raible, V. Cocco, T. Frolicher, J. Dunne, M. Gehlen, L. Bopp, J. Orr, J. Tjiputra, C. Heinze, J. Segscheider, T. Roy and N. Metzl, 2012. Variability of the Ocean Carbon Cycle in Response to the North Atlantic Oscillation. Tellus B, 64, 18738, doi:10.3402/tellusb.v64i0.18738.

Landschützer, P., Ilyina, T., & Lovenduski, N. S. (2019). Detecting regional modes of variability in observation-based surface ocean pCO2. Geophysical Research Letters, 46. https://doi.org/10.1029/2018GL081756.

Loveday, B. R. and Smyth, T.: A 40-year global data set of visible-channel remote-sensing reflectance and coccolithophore bloom occurrence derived from the Advanced Very High-Resolution Radiometer catalogue, Earth Syst. Sci. Data, 10, 2043-2054, https://doi.org/10.5194/essd-10-2043-2018, 2018.

McKinley, G. A., A. R. Fay, T. Takahashi and N. Metzl, 2011. Convergence of atmospheric and North Atlantic carbon dioxide trends on multidecadal timescales. Nature Geoscience. doi:10.1038/NGEO1193

McKinley, G.A., A.R. Fay, N. Lovenduski, and D.J. Pilcher, 2017. Natural variability and anthropogenic trends in the ocean carbon sink. Annual Review Marine Science. doi: 10.1146/annurev-marine-010816-060529

McKinley, G. A., Ritzer, A. L., and Lovenduski, N. S.: Mechanisms of northern North Atlantic biomass variability, Biogeosciences, 15, 6049-6066, https://doi.org/10.5194/bg-15-6049-2018, 2018.

Metzl, N., A Corbière, G. Reverdin, A. Lenton, T. Takahashi, A. Olsen, T. Johannessen, D. Pierrot, R. Wanninkhof , S. R. Ólafsdóttir, J. Olafsson and M. Ramonet, 2010. Recent acceleration of the sea surface fCO2 growth rate in the North Atlantic subpolar gyre (1993-2008) revealed by winter observations, Global Biogeochem. Cycles, 24, GB4004, doi:10.1029/2009GB003658.

Olsen, A., R. M. Key, S. van Heuven, S. K. Lauvset, A. Velo, X. Lin, C. Schirnick, A. Kozyr, T. Tanhua, M. Hoppema, S. Jutterström, R. Steinfeldt, E. Jeansson, M. Ishii, F. F. Pérez & T. Suzuki, 2016. An internally consistent data product for the world ocean: the Global Ocean Data Analysis Project, version 2 (GLODAPv2), Earth Syst. Sci. Data, 8, 297-323, doi:10.5194/essd-8-297-2016.

Pérez, F. F., Vázquez-Rodríguez, M., Mercier, H., Velo, A., Lherminier, P., and Ríos, A. F.: Trends of anthropogenic CO2 storage in North Atlantic water masses, Biogeosciences, 7, 1789-1807, https://doi.org/10.5194/bg-7-1789-2010, 2010.

Pérez et al. 2013, Atlantic Ocean CO2 uptake reduced by weakening of the meridional overturning circulation Nature Geoscience, 6 doi: 10.1038/ngeo1680.

Pilcher, D. J., Brody, S. R., Johnson, L., Bronselaer, B., 2015. Assessing the abilities of CMIP5 models to represent the seasonal cycle of surface ocean pCO2. Journal of Geophysical Research Oceans, 120, 7, 4625–4637, doi: 10.1002/2015JC010759.

Racapé, V., N. Metzl, C. Pierre, G. Reverdin, P.D. Quay, S. R. Olafsdottir, 2014. The seasonal cycle of the d13CDIC in the North Atlantic Subpolar Gyre. Biogeosciences, 11, 6, 1683-1692, doi:10.5194/bg-11-1683-2014.

Reverdin, G., Metzl, N., Olafsdottir, S., Racapé, V., Takahashi, T., Benetti, M., Valdimarsson, H., Benoit-Cattin, A., Danielsen, M., Fin, J., Naamar, A., Pierrot, D., Sullivan, K., Bringas, F., and Goni, G.: SURATLANT: a 1993–2017 surface sampling in the central part of the North Atlantic subpolar gyre, Earth Syst. Sci. Data, 10, 1901-1924, https://doi.org/10.5194/essd-10-1901-2018, 2018a.

Reverdin, G., Valdimarsson, H., Alory, G., Diverres, D., Bringas, F., Goni, G., Heilmann, L., Chafik, L., Szekely, T., and Friedman, A. R.: North Atlantic subpolar gyre along predetermined ship tracks since 1993: a monthly data set of surface temperature, salinity, and density, Earth Syst. Sci. Data, 10, 1403-1415, https://doi.org/10.5194/essd-10-1403-2018, 2018b.

Rödenbeck, C., R.F. Keeling, D.C.E. Bakker, N. Metzl, A. Olsen, C.Sabine, and M. Heimann, 2013. Global surface-ocean $pCO_2$ and sea-air $CO_2$ flux variability from an observation-driven ocean mixed-layer scheme. Ocean Science, 9, 193-216. doi:10.5194/osd-9-193-2013.

Rödenbeck, C., D. C. E. Bakker, N. Metzl, A. Olsen, C. Sabine, N. Cassar, F. Reum, R. F. Keeling, and M. Heimann, 2014. Interannual sea–air $CO_2$ flux variability from an observation-driven ocean mixed-layer scheme. Biogeosciences, 11, 4599-4613, 2014 doi:10.5194/bg-11-4599-2014.

Schuster, U., A.J. Watson, N. Bates, A. Corbière, M. Gonzalez-Davila, N.Metzl, D. Pierrot and M. Santana-Casiano, 2009. Trends in North Atlantic sea surface $pCO_2$ from 1990 to 2006. Deep-Sea Res II, doi:10.1016/j.dsr2.2008.12.011.

Schuster, U., G. A. McKinley, N. Bates, F. Chevallier, S. C. Doney, A. R. Fay, M. González-Dávila, N. Gruber, S. Jones, J. Krijnen, P. Landschützer, N. Lefèvre, M. Manizza, J. Mathis, N. Metzl, A. Olsen, A. F. Rios, C. Rödenbeck, J. M. Santana-Casiano, T. Takahashi, R. Wanninkhof, and A. J. Watson, 2013. An assessment of the Atlantic and Arctic sea-air $CO_2$ Fluxes, 1990-2009. Biogeosciences, 10, 607-627, doi:10.5194/bg-10-607-2013.

Signorini, S. R. , S. Häkkinen, K. Gudmundsson, A. Olsen, A. M. Omar, J. Olafsson, G. Reverdin, S. A. Henson, C. R. McClain and D. L. Worthen, 2012. The Role of Phytoplankton Dynamics in the Seasonal and Interannual Variability of Carbon in the Subpolar North Atlantic: Âň A Modeling Study. Geosci. Model Dev., 5, 683-707, doi:10.5194/gmd-5-683-2012

Takahashi, T., S C. Sutherland, R.Wanninkhof, C. Sweeney, R.A. Feely, D. Chipman, B. Hales, G. Friederich, F. Chavez, A. Watson, D. Bakker, U. Schuster, N.Metzl, H.Y. Inoue, M. Ishii, T. Midorikawa, C.Sabine, M. Hoppema, J.Olafsson, T. Amarson, B.Tilbrook, T. Johannessen, A. Olsen, R. Bellerby, Y. Nojiri, C.S. Wong, B. Delille, N. Bates and H. De Baar, 2009. Climatological Mean and Decadal Change in Surface

Ocean pCO2, and Net Sea-air CO2 Flux over the Global Oceans. Deep-Sea Res II, doi:10.1016/j.dsr2.2008.12.009.

Thomas, H., A.E.F. Prowe, I.D. Lima, S.C. Doney, R. Wanninkhof, R.J. Greatbatch, U. Schuster and A. Corbière, 2008. Changes in the North Atlantic Oscillation influence CO2 uptake in the North Atlantic over the past two decades, Global Biogeochem. Cycles, 22, GB4027, doi:10.1029/2007GB003167.

Ullman, D. J., G.A. McKinley, V. Bennington and S. Dutkiewicz, 2009. Trends in the North Atlantic carbon sink: 1992-2006, Global Biogeochem. Cycles, 23, GB4011, doi:10.1029/2008GB003383.

Watson, A.J., U. Schuster, D .C. E. Bakker, N. Bates, A. Corbiere, M. Gonzalez-Davila, T. Freidrich, J. Hauck, C. Heinze, T. Johannessen, A. Koertzinger, N. Metzl, J. Olafsson, A. Olsen, A. Oschlies, X. Padin, B. Pfeil, A. Rios, M Santana-Casiano, T. Steinhoff, M. Telszewski, D. W. R. Wallace, R. Wanninkhof, 2009. Tracking the variable North Atlantic sink for atmospheric CO2, Science ,326, 1391, doi:10.1126/science.1177394.

[Figure]

Figure 1. Effect of the changes in total (filled), anthropogenic (doted) and natural (empty) DIC to the trends in surface pH during summer for the three periods and different boxes (B, C, D-E and C-D-E).

**Fig. 1.**

[Figure]

Figure 2. Effect of the changes in total (filled), anthropogenic (doted) and natural (empty) DIC to the trends in surface pH during winter for the three periods and different boxes (B, C, D-E and C-D-E).

**Fig. 2.**

---

## Author Comment (AC2) · 23 Jun 2019

Leseurre, C., Lo Monaco, C., Reverdin, G., Metzl, N., Fin, J., Olafsdottir, S., and Racapé, V.: Ocean carbonate system variability in the North Atlantic Subpolar surface water (1993–2017), Biogeosciences Discuss., https://doi.org/10.5194/bg-2019-119, in review, 2019.

Review 2 (Are Olsen):

"This contribution presents the trends in a very impressive set of data from the western subpolar North Atlantic, collected within the framework of the SURATLANT program. While the paper presents the data and trends more or less adequately, it fails in properly attributing them to (climatic + oceanographic + biological) drivers. These aspects are

unclear, speculative, and somewhat confused. As such, major revisions are required. My major comments are:"

We thank Are Olsen for his fast review, comments and questions that will be taken into account when revising the manuscript. Below we list our responses before preparing a revised manuscript.

1. Samples are collected two to four times a year. My sense is that with such rare sampling one might be very prone to getting false trends because of differences in the timing of the data collection, also in combination with timing of the spring bloom. For example, the strong increase in region C+D+E summer DIC in 2001 -2008 might be generated by earlier sampling/later spring bloom.

"While some effort seems to have been taken to deal with this; reconstructed data; it is not quite clear how this is done, and how well it works. Some improvement is needed in this description (page 5 lines 1-19). Is it possible to test this scheme by using data from a year with sampling in all of the months used (e.g. Jan-March)? At least for SST and SSS this can be done as there should be continuous TSG data vailable, and for $fCO_2$ there are data available from the VOSs Nuka Arctica and Atlantic Companion, which crosses the study area. These can be retrieved from SOCAT and used to test the method".

Response: We fully agree with the reviewer. For the analysis of trends, one should take care of the timing of sampling especially in summer (as well as different tracks for some transects). This is why previous studies focused on winter data (e.g. Metzl et al., 2010; Fröb et al., 2019). Here, in addition to winter 2008-2017 not previously analyzed, we also tried for the first time to evaluate the trends in summer (June-July-August), i.e. after the spring bloom that generally occurs in April-May-June (with a date variable between regions and dates). Do we detect (or not) different view on the trends when comparing winter and summer? On average for 1993-2017, we observed that DIC and $fCO_2$ increase (and pH decrease) in both winter and summer (but with different rates).

Results are much more different when trends are limited to short periods.

Concerning the "strong increase in region C+D+E summer DIC in 2001 -2008". This is apparently driven by low DIC observed in summer 2001 and high DIC in summer 2007 (medium concentrations observed in 2004-2005, but no data in 2006). Note that for this particular summer 2007 transect samples were taken in early August 2007 and thus correction to standardize to mid-July should not introduce suspicious bias. As suggested by the reviewer, we had a look at the Chl-a time-series (from MODIS) but for now we did not observe a significant difference for the summer 2007 (e.g. low productivity). Also, in summer 2008, DIC and nutrients were significantly lower than in 2007. At that stage we have no firm conclusion to explain the rapid increase of DIC (and fCO2) for summer 2001-2007. Note that in winter 2001-2007 we also found rapid fCO2 increase.

In years when observations were not collected in July or February (selected months for summer and winter trend and driver's analysis), we use data collected in previous or next month by correcting for the climatological trend between successive months. As suggested by the reviewer we will confirm this scheme with the more regularly-observed SST and SSS data, based on the binned monthly products constructed by Reverdin et al. (2018b).

We also checked the SOCAT data in the region for specific years (e.g. the summer 2010 anomaly). A comparison with SOCAT fCO2 data for the full period would be very interesting but this is beyond the scope of our analysis mainly based on DIC/TA observations. This would be a topic for another publication shared with SOCAT data providers in the North Atlantic.

2. More broadly, it would be interesting to know how representative the data are for large-scale interannual phenomena. This can be tested. For example, the trends in SST can be compared with objectively analysed SST from NOAA. The trends in SSS can be compared with some reanalysis model output. And again, for the time representativeness, the continuous data from the TSG, which collects data on all crossings, should be used. It might also be worthwhile to look at remotely sensed Chl a, to evaluate if there are concomitant trends in surface ocean primary production, e.g. a loss in production from 2001 -2008 such as the DIC data seem to indicate and was suggested in the manuscript.

Response: as suggested by the reviewer, we will compare the SST and SSS trends with monthly fields along AX-2 line constructed by Reverdin et al. (2018b). We found coherent view on the timing and variability of surface properties (more clearly seen in SST and SSS anomalies). This will be added in the revision. We will also use SST reanalysis to confirm the trends for the selected periods. We also started to investigate Chl-a data (from MODIS). Preliminary results suggest that there are some Chl-a trends that might qualitatively explain the variability of DIC (more specifically variations of DIC-natural derived from DIC corrected to anthropogenic signal). To quantify the impact of biological processes on DIC and fCO2, a more specific study should be performed based on primary production (not only biomass). On this issue, Bennington et al (2009) used an ocean model and satellite Chl-a to explore the effect of biological activity on carbon uptake in the North Atlantic and found that biological variability "is not sufficient to be a first-order control on annual subpolar air-sea CO2 flux variability." This might be explored in more detail with our observations (including nutrients) but for another publication. As a first step, we explored the winter and summer trends of natural N-DIC over 1993-2017 and found no significant trend in summer and winter, suggesting that biological activity variability has a small impact on the DIC and fCO2 long-term changes. This analysis from in-situ data confirms the conclusion from ocean models (Bennington et al, 2009).

The main signal that we can interpret based on satellite Chl-a is for the warm summer 2010 with high Chl-a observed (probably linked to NAO signal) and that could explain part of the DIC changes during that summer. However, this event has little impact on the trends for this specific period (summer 2008-2017). There are no summer 2010

SOCAT data, but the low summer 2010 DIC is confirmed with independent DIC and TA OVIDE-2010 observations.

3. The attribution section is not well done. In particular, I strongly suggest the authors to explicitly account for salinity changes in the pCO2 driver decomposition following the method by Keeling et al. (2004) and recently used in the subpolar North Atlantic by Fröb et al. (2019). The reason is that dilution may completely overwhelm increase of DIC expected from uptake of anthropogenic CO2.

Response: In the revised manuscript, we will consider separately the effect of salinity and N-DIC, N-TA following the reviewer's suggestion.

4. Further, showing trends in salinity normalised DIC and TA is worthwhile, but make sure to use the correct method for salinity normalising as described in Friis et al. (2003).

Response: we will add trend analysis in salinity normalized DIC and TA. In previous work we found about the same trends in 2001-2007 for TA/N-TA (negative) and DIC/N-DIC (positive) but for winter only (Metzl et al 2010). New results suggest that for summer 2001-2007, the trends are also about the same for DIC and N-DIC or TA and N-TA. However, as salinity is decreasing in 2008-2017 (also previously documented by Reverdin et al 2018a,b; Tesdal et al 2018; Fröb et al 2019), trends for N-TA and N-DIC are significantly different compared to TA and DIC for the last decade. This supports the reviewer's suggestion to show both DIC/TA and N-DIC/N-TA trends and discuss processes that drive the observed changes. Of course, we follow the same approach as Friis et al. to estimate the normalized DICA and TA (but with equation in Reverdin et al. (2018a)). Thanks to highlight this issue.

5. Also, there is a lot of mentioning of the change in the air-sea CO2 difference. But this is not illustrated, which leaves a lot to the readers imagination. I therefore recommend to show the actual air-sea fCO2 difference in Fig. 3. You can make room for this by removing one of the panels for Omega (Calcite/Aragonite), as there is no need to show both.

Response: this is a good point and we will add a figure of air-sea fCO2 differences. Replacing Omega is a nice idea, but we think a separate figure should be added to present air-sea CO2 gradients for the full period and seasons.

6. Uncertainties are not properly dealt with. For example, the uncertainty in the SST of 0.1 degrees C results in an fCO2 uncertainty of 2 $\mu$atm. There are also uncertainties in DIC and TA. These errors need to be propagated to calculated fCO2, pH etc. This can be done using the most recent CO2SYS from James Orr as this includes error propagation. It is available from GitHub. The errors in fCO2 (and the others) can be propagated to the trends using Monte Carlo.

Response: In Table 2, significant trends are presented with (*). But, as in some cases, there are no trends (this can not be determined by the Student's method), so we chose a maximum error criterion (but maybe not adapted as suggested by Are Olsen). We will correct that with the most recent CO2SYS (from James Orr), thank you for this pertinent comment.

7. I am in particular concerned with the fact that the large summer fCO2 increase in region B 2001-2008 are basically caused by the 'reconstructed' data of 2001 and 2008. The actual observations are pretty steady. How confident are you in these reconstructed values?

Response: the large summer fCO2 increase in 2001-2008 is observed in all boxes. Reviewer is correct that starting (2001) and ending (2007) points were reconstructed for July from data obtained in August (for both boxes B and CDE). If we don't standardize the data to July, the "summer" (June+July+August) trend in Box B would be +6.8 $\mu$atm/yr (instead of 8.5 when normalizing to July). The same test for box CDE leads to fCO2 increase of +11.2 $\mu$atm/yr (instead of 11.5 when normalizing to July). We should note that for August 2007, TA data were doubtful and we used reconstructed TA from salinity for this specific cruise. Overall, we are confident that in 2001-2007 the fCO2 increased much faster than in the atmosphere.

8. As the other reviewer, I think this contribution confuses anthropogenic and natural CO2. The trends that are observed in pCO2 does not say any on anthropogenic carbon uptake. Further, the North Atlantic is not a big sink of anthropogenic CO2 because the air-sea flux is large. The air-sea flux is a combination of natural and anthropogenic CO2 fluxes. Horizontal advection is likely a big source of anthropogenic CO2 to the North Atlantic. Hence, the data that are presented only informs about the changes in surface pCO2 and air-sea CO2 flux, not on the North Atlantic sink for anthropogenic CO2. This needs to be considered both in the introduction and in the discussion.

Response: Thank you to highlight the anthropogenic versus non-anthropogenic signal issue (same remark by reviewer 1). We therefore address the same response below.

As the DIC and pCO2 variability are large in surface waters, detection of anthropogenic CO2 (C-ant) is difficult (if time-series are limited) and data-based methods such as C0, TrOCA, delta-C*... not suitable to quantify C-ant in surface waters. Indeed, longer-term time-series are required to separate natural and anthropogenic (or climate induced) signals (e.g. at least 30 years in the North Atlantic subpolar gyre, McKinley et al, 2011). We have evaluated the anthropogenic concentrations (C-ant) in this region based on subsurface Glodap-V2 data (Olsen et al., 2016) and TrOCA method, but we did not introduce the results in the present paper as they correspond to a different period (and only for summer). However, in the revision, as recommended by reviewers, we will introduce the trends of C-ant evaluated in subsurface from different methods (our calculations and recently from Gruber et al., 2019). In short, based on Glodap-V2 data in the NASPG we estimate C-ant trend of +0.7 $\mu$mol/kg/yr at subsurface (150-200m) for the period 1997-2011. This would explain 75% of the DIC trend of +0.9 $\mu$mol/kg/yr observed at subsurface for the same period. On the other hand, using the new e(MLR*) method, Gruber et al (2019) evaluate accumulation of C-ant from 1994 to 2007 in the global ocean. In the North Atlantic and specifically along the SURATLANT line, the accumulated C-ant is +8.5 (+/- 1.7) $\mu$mol/kg in the layer 150-200m. This signal is rather homogeneous at depth (150-200m) but with a small gradient between southern

(6 $\mu$mol/kg) and northern (10 $\mu$mol/kg) NASPG regions. This would correspond to a trend of between +0.6 and +0.7 $\mu$mol/kg/yr, very close to our C-ant estimate based on Glodap-V2. If we assume that the subsurface C-ant trend results are correct and also valid for the surface layer, the e(MLR*) method lead to a very homogeneous C-ant accumulation in the layer 0-50m along the SURATLANT line (55N-64N), of +10.1 (+/- 0.8) $\mu$mol/kg between 1994 and 2007, i.e. about +0.8 $\mu$mol/kg/yr. Interestingly, this is in the range of the long-term (1993-2017) DIC surface trends that we report (table 2 in our paper) between 0.6 and 0.7 $\mu$mol/kg/yr in boxes B,C,D,E (depending the season). However, our observations also show that DIC trends could be very different for short periods and north/south regions (see all trends listed in Table 2), and occasionally DIC decreases over time as opposed to C-ant (e.g. 1993-1997 winter, this study; Metzl et al., 2010; Ullman et al 2009). Based on the C-ant estimates, we will also evaluate and discuss the trends of DIC-Cant (C-nat), i.e. the natural part of the signal in the revision. We will discuss the results in relation with biological activity as also suggested by reviewer (using remote sensing Chl-a).

Figure 1 and 2 (attached) show the anthropogenic and natural contributions of DIC to pH change. This overview is obtained with the C-ant set at 0.7 $\mu$mol / kg / yr (in each region, period, season). We note that the results presented in boxes C and D-E are substantially the same as when we group them in C-D-E.

9. I think the discussion is pretty disappointing. It is basically a recap of the results + some more exploration of these, combined with some speculation based on published literature. There are no attempts to analyses the relation between the observed trends and likely drivers (for which data exist) – such as NAO or AMV indices, winter/summer mixed layer depths (Argo data), primary production (remotely sensed ocean color), and SGP strength (SPG index). This should be done.

Response: We agree that a further discussion should be attempted to link the observed changes of surface properties with regional or large scale forcing such as NAO, AMV. The same comment was addressed by reviewer 1. We therefore address the same

response below.

About NAO:

NAO was previously recognized a possible link of the rapid fCO2 increase when NAO shifted from positive to negative phase in 1995-1996 (Corbière et al., 2007). However, this was not confirmed for the period 2001-2007 when NAO did not vary so much around neutral value (Metzl et al., 2010). A possible explanation is that the observed variations of fCO2 in the NASPG, especially rapid trends such as +7 $\mu$atm/yr observed in 2001-2007, are driven by superimposed processes linked to climatic signals such as NAO and AMO, and both should be taken into account, as well as other processes involved in the NASPG ocean circulation, ventilation and vertical mixing. AMO, based on SST, is a long-term multidecadal signal (sometimes better called AMV) that experienced a gradual progressive increase from negative values in the 70s to positive values in the early 2000s, and remains positive and relatively stable in 2002-2017. NAO based on sea level atmospheric pressure gradient, shows much shorter variability, with highs and lows occurring in its winter record at interannual to decadal periods. We think that a direct relationship of the variability of pCO2 or CO2 uptake in the North Atlantic with NOA is still ambiguous (e.g. Takahashi et al., 2009; Schuster et al., 2013; Mc Kinley et al. 2017) and the detection of pCO2 changes with climate variability is still challenging from observations (at least for the period we investigated 1993-2017). However, for long-term multi-decadal variability the link between AMO and pCO2 change in the NASPG appears more robust (Breeden and McKinley, 2016; Landschützer et al., 2019).

Summary of NAO and model results cited:

Ocean models can help to understand the link between NAO and biogeochemical cycles, but results from models are still controversial. Keller et al (2012) who investigated several simulations (6 Earth systems models and for winter only, i.e. not the productive season), conclude that on-site entrainment in the subpolar gyre (mixing, upwelling) is

the main driver of carbon sink variability as opposed to advection (as suggested by Thomas et al. 2008). In their model Thomas et al (2008) suggest that negative or neutral NAO conditions result in a substantial decline in CO2 uptake for years 1997-2004 along the NAC and in the eastern subpolar gyre. However, in another modeling study of the North Atlantic, Ullman et al. (2009) conclude that the air-sea CO2 flux increased over the 1992–2006 and this is due to the increasing atmospheric CO2 and not to long-term variability in the physical climate or biogeochemistry. During the transition of NAO (from positive to neutral), their model simulates a decline of the convection and vertical DIC supply to the surface in the subpolar region, counteracting the increase of pCO2 due to warming. This leads to a small net pCO2 increase (compared to atmospheric trend) and increasing CO2 sink. Interestingly in the NASPG, the DIC decrease of -0.75 $\mu$mol/kg/yr in winter is more pronounced in the model (Ullman et al., 2009) than in the SURATLANT data (for 1992-2005). That might explain why observations suggest a reduced CO2 sink after the NAO shift in the mid-90s (Corbière et al., 2007), while the model suggest an increasing sink.

Keller et al., (2012) show that simulations with different coupled models lead to different results, and the response to NAO seems modest: typical NAO-driven variations at large-scale are +/- 10 $\mu$mol/kg for surface DIC and TA, and +/- 8 $\mu$atm for delta-pCO2. Depending on the model, the change of pCO2 in the subpolar region varies between +2/-4 $\mu$atm (for NAO+) and +16/-12 $\mu$atm (for NAO-), that is even the sign of the response is different between models. Such low variations, if real, are rather difficult to extract from observations and thus the results of the models difficult to validate. Thus, Keller et al (2012) conclude that although the interannual variability in the North Atlantic is largest in the subpolar gyre, the magnitude and responses of the carbon uptake to NAO significantly differ between the models (recall that this conclusion holds only for winter).

These model studies and their controversial results (level of variability and processes at play) show that there is still more work to be performed to understand the link between

[Figure]

NAO and biogeochemistry. As a matter of fact, in a recent analysis on the link between NAO and biology in the North Atlantic, Mc Kinley et al. (2018) conclude that "nowhere is the NAO correlated with biomass variability and that more investigation of the links between North Atlantic climate and biomass variability is clearly warranted". The same is true for biogeochemistry and carbon cycle and we believe that the new data on the carbonate system we analyzed in our submitted paper along the SURATLANT line should offer new information on this issue, even if the main conclusion is that over 1993-2017 there is NO direct link detected between fCO2 and DIC trends with NAO in this region of the NASPG. Of course, at shorter time scale (1-2 years) we might recognize such a link as described below (and see also the large changes in regions with changes in convection, such as in the western Irminger Sea, Fröb et al., 2018).

Summary of observations with new data related or not with the NAO:

The update made for the last decade 2008-2017 (data not included in our previous work, Corbière et al., 2007; Metzl et al., 2010) adds observations obtained during a strong negative NAO in 2010 and a positive NAO phase in 2015.

The 2010 event was associated with a warming and freshening (and low density) found in both SURATLANT discrete sampling (in August 2010) and monthly reconstructed Binned products (Reverdin et al 2018b). In August 2010 we observed low DIC (and also high $\delta$13CDIC as discussed by Racapé et al 2014), a signal also revealed in high Chl-a concentrations (identified from MODIS data). However, as this was associated with a warming (observed positive SST anomaly up to +1.5 °C), the fCO2 (and pH) values were not very different from previous summers, illustrating the competitive effect of warming and higher production on fCO2 for this event. We also note that in summer 2010, DIC/TA was also sampled during the OVIDE-2010 cruise (in late June). In the NASPG, surface DIC concentrations for OVIDE-2010 were around 2070-2090 $\mu$mol/kg, i.e. just between SURATLANT data obtained in early June (2100-2110 $\mu$mol/kg) and in mid-August (2050-2070 $\mu$mol/kg); for TA, concentrations were the same for all cruises. This confirms the summer 2010 anomaly (low DIC) apparently associated with higher

productivity during the negative NAO phase, but with no significant impact on fCO2 and trends.

We also identified a cold year in 2015 (SST anomaly around -1 °C), when NAO was in a positive phase: for that year, DIC was higher in winter (near the maximum observed on average in winter in our time series) but again, as the temperature also lowers fCO2, the fCO2 value were not so different from other winters, now illustrating the competitive effect of cooling and deep mixing on fCO2.

With the new data introduced in this manuscript for 2008-2017 that corresponds to a period when NAO presents large IAV compared to 1995-2007, we observed more variability in the DIC data (and fCO2, pH) that might result in less clear trends over 5-10 years. We have identified two specific years (2010 and 2015) that experienced very low NAO (-3 in 2010) and high NAO (+2 in 2015), leading respectively to observed warming (cooling), freshening (saltier) and low (high) DIC; these anomalies could be explained by an increase in productivity in 2010 and deeper mixing in 2015. As these anomalies have been clearly recognized, we have tested the sensitivity of the trends analysis with and without these NAO events. Not surprisingly, for the 2015 anomaly and because we have no winter data after 2015, we derived significant different trends when 2015 is or not considered for the period 2008-2015. For example, for the northern boxes CDE the winter trends evaluated for years 2008-2015 were +1.1 $\mu$mol/kg/yr for DIC and 0.6 $\mu$atm/yr for fCO2. If we restrict to the period 2008-2014, trends become negative, i.e. -0.14 $\mu$mol/kg/yr for DIC and -0.8 $\mu$atm/yr for fCO2. We thus have to be careful when selecting (and interpreting) the periods. On the other hand, if we test the sensitivity of the trends for summer season in 2008-2017 (with or without the 2010 anomaly), results are basically the same. These specific results recall that fCO2 trends (and here also for DIC) are highly sensitive to the choice of starting and ending years as was illustrated by Mc Kinley et al (2011).

We also tested the impact of these NAO events (high and low) on the long-term trends, 1993-2017 for summer and 1994-2015 for winter. In that case, the results are more

robust. For example, for the northern boxes CDE the winter trends evaluated for years 1994-2015 were +0.6 $\mu$mol/kg/yr for DIC and +1.4 $\mu$atm/yr for fCO2. If we restrict to the period 1994-2014, trends are lower, i.e. +0.5 $\mu$mol/kg/yr for DIC and +1.3 $\mu$atm/yr for fCO2. In both cases, the DIC trend appears close to the increase due to anthropogenic uptake (estimated around +0.6 to +0.8 $\mu$mol/kg/yr in this region; see our response to other reviewer comments below). For summer, the trend in 1993-2017 with or without the 2010 NAO anomaly leads to the same results: +1 $\mu$mol/kg/yr for DIC and +2 $\mu$atm/yr for fCO2. In that case, the NAO has no effect on the trends. For the full period, the summer trends appear faster than derived from winter data and this needs to be clarified. This is also why it is important to separate the full period in 3 parts (as presented in the submitted paper) to better investigate the process involved, especially with the cooling and freshening observed after 2008. If the NAO events are relatively well characterized with our observations and could impact the trends when evaluated over 5-10 years, for long-term trends (24 years) this seems a secondary effect (as also suggested from long-term ocean simulations, Breeden and McKinley 2016).

As a final test, we investigated the SOCAT data (version V6, Bakker et al, 2016) in this region corresponding to the NAO events. Unfortunately, there is no fCO2 data for July-August 2010 that could be used to support the low DIC concentrations we observed in August 2010, but also in late June 2010 during the OVIDE-2010 cruise. However, for 2015 (high NAO), SOCAT data suggest relatively high fCO2 around 390-415 $\mu$atm in Jan-Feb 2015 in the NASPG, not far from our average value of around 400 $\mu$atm. A comparison with SOCAT fCO2 data for the full period would be interesting but this is beyond the scope of our analysis mainly based on DIC/TA observations. This would be a topic for another publication shared with SOCAT data providers in the north Atlantic.

Based on our results and sensitivity analysis described above, we will introduce the NAO events in the revision and discuss how NAO may be linked (or not) to the CO2 trends. We thank again Are Olsen to highlight this issue in his comment.

10. Finally, Metzl et al (2010) suggested deep mixing as the cause for the sharp winter fCO2 trend in 2001-2008. Fröb et al (2019) demonstrated unequivocally that deep mixing leads to strong increases in winter fCO2 in NAO positive years because of intensified deep mixing (bringing remineralised DIC to the surface). In light of this, it is interesting that the smallest winter fCO2 trends from SURATLANT are observed in the period 2008-2017, as it is well known that this has been a period of rather large deep mixing compared to 2001-2008. I am not sure how this can be reconciled, i.e. the large fCO2 trend in the period of little deep mixing 2001-2008 vs the small fCO2 trend in the period of frequent deep mixing in 2008-2017. It seems worthwhile to delve into this. While doing so, keep in mind that deep-mixing events seemed (in Fröb et al., 2019) to cause year-to-year anomalies and not so much an anomalous trend. Therefore, I recommend to reconsider the use of three periods - the trend in each of these can be strongly affected by the fCO2 in the start and/or end year, which might just be an anomalous year. Therefore, the authors might be doing themselves a disfavor by sticking to the three periods, which are largely defined because of 'historical'/'traditional' reasons related to sampling and previous SURATLANT papers.

Response: Reviewer is partially correct. Not surprisingly, the trends for each period strongly depend on starting and ending points. This somehow recalls the sensitivity analysis presented by McKinley et al. for decadal trends (2011, their figure 2a). The 3 periods we selected were based on (i) available DIC and TA data, (ii) variability of SST and salinity (more specifically trends of SST and SSS anomalies as derived from monthly binned products, Reverdin et al 2018) and (iii) fCO2 (and pH) observed changes.

We also agree that deep-mixing events (for winter) or high/low productivity (in summer) would drive large anomalies for specific years, but might not impact much the multi-year trends. As a matter of fact, we identified some years with significant anomalies (e.g. summer 2010 or winter 2015 probably linked to NAO) that might change the trend if these events are or not taken into account (i.e. if these events were just not observed and missing in the time-series). We will discuss more clearly in the revised manuscript

that in winter 2015 (positive NAO), and because we have no winter observations after 2015, this specific year leads to a relatively large trend. If high DIC and fCO2 in winter 2015 are not taken into account the trends for winter 2008-2014 are significantly different. Interestingly, in a recent analysis based on SOCAT data, Denvil-Sommer et al (2019) suggests large variability in the NASPG (more specifically the NASPSS biome, Fig S5) for years 2001-2015, especially after 2013 (but with different responses between data-based methods). The observations that will be conducted during the next 5 years or so would support (or not) what we have learned from data presently available. On the opposite, if we filtered the summer 2010 anomaly, the trends in summer for 2008-2017 are almost the same. Finally, we note that if the trends are evaluated for the full period (1993-2017), the winter 2015 anomaly has less impact. In the revision, as suggested by reviewer, we propose to show first the long-term trend and then separate the period to better explain the drivers and contrasting response between region and periods. We will also investigate mixed-layer variability based on either in-situ data (ARGO) or models outputs.

Figure 3 (attached) show a link the same variability of temperature observed for two different data-set, in like to AMO. Regarding the depth of the mixing layer, the index presented in Figure 4 (very low frequency) does not seem conclusive, in contrast to the temperature. Thus, these results seem to support again that the NAO does not have too much impact on trends, which reinforces our previous conclusions.

Other comments:

"Page 2, line 4. Takahashi's estimates of air-sea CO2 flux cannot be equated to anthropogenic CO2 flux.

Response: The reviewer is correct (same comment as reviewer 1). The sentence will be revised as: "The North Atlantic is one of the strongest ocean sinks for natural (Takahashi et al., 2009) and anthropogenic atmospheric CO2 (Sabine et al. 2004; Khatiwala et al., 2013)." At that stage in the introduction, we don't really need to specify

number such as 0.27 PgC/yr (in our analysis we do not evaluate integrated flux over the domain).

"Page 2, line 6. I don't think any has ascribed the variations in the North Atlantic CO2 sink to climate change. Careful with such statements."

Response: Mc Kinley et al 2011 found that in the North Atlantic subtropical gyre, the increase of pCO2 by a warming trend is partially due to anthropogenic forcing (i.e. interpreted as climate change). For clarity, we will revise the following: However, repeated observations have shown important variations in this natural sink for anthropogenic CO2 in response to climate variability (Corbière et al., 2007; Metzl et al., 2010; McKinley et al., 2011; Landschützer et al., 2013).

"Page 2, line 21. "based on Total Alkalinity (TA) and Dissolved Inorganic Carbon (DIC) observations [. . .]".

Response: will be corrected

"Page 3, line 7. 'which', not 'witch'."

Response: will be corrected

"Page 3, line 14-15. 'to this end', not 'to this aim'."

Response: will be corrected

"Page 3, line 15. 'Iceland', not 'Island'."

Response: will be corrected

"Page 3, line 17. These acronyms have already been defined."

Response: will be corrected

"Page 4, equation 1. I think it would be useful to recap the accuracy of this relationship, and how well it works for the data that are presented here."

Response: The TA/S relation was established by Reverdin et al (2018) and used here when TA data were not available in 1993-1997 and occasionally when TA data were flagged "dubious or bad". For a regional view (as described in Reverdin et al), the TA/S relation was obtained for salinity above 34. These authors also note: "For the lower salinities found on the Newfoundland shelf, different sources of freshwater (from the Arctic or resulting from continental or sea ice melt inputs) contribute to deviations from the relation." For the southern region (box A and box B for some periods when salinity is low), one should be careful when using this relation. This is also why in this paper, we did not describe the long-term trend in the southern region (box A) which would require a dedicated analysis. In addition, one cannot exclude the impact of blooms of coccolithophorids observed in the NASPG, as variability of such blooms may also impact on TA concentrations and thus deviate from the TA/S regional relation. Recently, Loveday and Smyth (2018) show significant differences of such blooms over decades in the North Atlantic. An interesting topic would be to re-explore TA distribution and TA/S relations for different periods. This is why we recall that a specific study of TA spatiotemporal distribution and drivers of TA variability should be performed, but beyond the scope of the present analysis. Details of the TA/S relation, including comparisons with other relations were presented in Reverdin et al., 2018, Appendix figure B1, B2, B3).

"Page 4. Atmospheric fCO2 calculation. What atmospheric pressure was used?"

Response: We used standard pressure (1 atm).

"Page 4, decomposition equation. As mentioned above, please use the method that explicitly accounts for the effects of dilution/concentration. Note also, that the equation as written is wrong. The dX/dt term should be dX/dz, where z is the driver in question. You further need to explain what values you used for these sensitivities, and how they were derived."

Response: Correct. Equation will be rewritten.

"Page 5, line 5. How many times were data excluded from box B because of SSS being outside the 34-35 range?"

Response: Over 470 samples in Box-B (50-54N) for the full period with DIC data, 399 data were in the salinity range 34-35 (38 data excluded for S<34 and 33 data excluded for S>35).

"Page 5. Also, please consider the grouping of these regions for the trend analysis from Figure 2 SSS and TA, E and D appears quite similar while C and B both appear different. My sense this that combining E and D, while keeping C and B separate might be the best approach."

Response: Trends and contributions have been tested accordingly (see table 1 and figure 5). Figure 5 show the decomposition of the trends in surface pH only (the same observation between boxes C, D-E and C-D-E is observed). Trends averaged one boxes C and D-E are overall similar to within the error bars. In all cases, they illustrate the large differences in the trends between the three different periods that match what we commented in the previous draft of the paper.

"Page 5, lines 10-19. Please collect this information in one section. Please also write that these are the reconstructed values in Fig. 3 (If I understand correctly the e.g Jan/March values adjusted to February are the 'reconstructed' values in Fig 3).

Response: The reviewer is correct. For the trend analysis we standardized winter to February and summer to July. Reconstructed values were adjusted based on the mean seasonal cycle. Of course, when no observation is available for the previous or the next month, there is no reconstructed value (i.e. gaps in the time series for some years).

"Page 6, lines 2-3. As mentioned above I am not convinced that these are 'pluriannual' trends. You might be doing yourself a disfavor by splitting the data into three time periods. It might make more sense to look at the timeseries as a whole and instead look at anomalies from the long-term trend. In particular this should be done from

2001 onwards. Consider to relate anomalies to mixed layer depths similar to Fröb et al. (2019)."

Response: In the revision, we propose to show first the long-term trend and then separate the period to better explain the drivers and contrasting response between region and periods. We will also explore mixed-layer depth as suggested.

"Page 6, lines 7 onwards. Here a panel showing air-sea fCO2 difference, as suggested above would help, as it will make the discussion more quantitative."

Response: Good point. We will add a separate figure to present air-sea dCO2 difference for the full period and seasons.

"Page 6, line 20. I think 'near-stagnation' is the wrong word here. In winter the increase appears significant, at 1 $\mu$atm/yr. Generally, please make the summary of the results quantative."

Response: This will be reformulated (depending on the new results based on revised calculations for the new Box definition).

"Page 7, line 11. Remove 'Thereafter'."

Response: Will be corrected.

"Page 7, line 15-16. Enclose 'much faster than the atmospheric signal', with commas and: 'suggest larger productivity in the beginning of the period than at the end'. BTW this can and should be checked with Chl a data."

Response: There is no Chl-a measurements in SURATLANT and Chl-a from remote sensing started in 1998. Unfortunately, we cannot explore a link between DIC and Chl-a trends for the period 1993-1997. From our results we can only suggest "an increase in productivity at the beginning of the period compared to the end" to explain why DIC was lower in 1993.

"Page 7, line 19, replace 'than' with 'to'."

Response: will be corrected

"Page 7, line 21-22. It is interesting that the slower increase in fCO2 is associated with strengthening of the winds and enhanced deep mixing. As noted above, Metzl et al., (2010) suggested deep mixing as the cause for the larger increase in the 2001-2008 period. Recently Fröb et al. (2019) found anomously high fCO2 during years of deep mixing. What is suggested here, is thus at odds with these papers. This needs to be explored or revised." Response: This is an interesting difference between the region further west investigated by Fröb et al. (2019) and the central gyre/Reykjanes Ridge region that is investigated here. Contrary to the western Irminger Sea, the area surveyed by SURATLANT does not encounter deep convection. Winter convection in the northern part of the SURATLANT area is limited to mode waters, such as the Reykjanes Ridge mode water (Thierry et al., 2007). These waters vary interannually, but not necessarily in phase with the deep convection of the western Irminger Sea. Furthermore, they only reach as deep as 400 to 650m depending on the year, and with a potential density close to 27.51. This is much shallower and less dense that the deep winter mixing in the western Irminger Sea) that was observed to reach the middle ocean layer (same density as Labrador Sea water). This depth-range difference can induce strong differences in the properties mixed to the surface. We would also like to comment that there is a difference between the effect of local winter mixing that will be strongly modulated with NAO (at least in central and northern part of SURATLANT) and the longer-term trends that are mostly explored here.

"Page 7, line 25. Set the '2' as subscript."

Response: will be corrected

Page 8, line 9 -10. The link between the trends in the carbon system and NAO+AMV that are described here are not backed up with any statistics. It comes across as very speculative. The statements need to be backed up with for example correlation analyses.

Response: we will study that.

"Page 8, line 11. Fröb et al (2018) shows in particular a large increase of anthropogenic DIC inventory in deep mixing years, and tendencies for a loss of natural carbon. Fröb et al. (2019) shows an outgassing of $CO_2$ during deep mixing years. The coupling between the inventory changes and the variability in fluxes has yet to be made. Some discussion around this would be interesting."

Response: we will try to add a discussion on this interesting issue, but this will also depend on revised calculations and if possible, link with MLD. To be done.

"Page 9, line 5. 'Makes it difficult to predict the evolution of $CO_2$ uptake...' I suggest to read the paper by Li et at 2016, on decadal predictions of North Atlantic $CO_2$ uptake, this might provide some relevant information."

Response: Regarding the prediction of carbon uptake, Li et al 2016 is a very good suggestion. The comparison with Li et al 2016 is however not clear as these authors indicate that (from simulations) when the mixing is enhanced in the western SPG this leads to more $CO_2$ uptake, whereas weaker mixing in the eastern SPG leads to less $CO_2$ uptake (their figure 1b and Supp Fig 1b). On this topic, we would also refer to Couldrey et al 2019.

References (not cited in the paper):

Fröb, F., Olsen, A., Becker, M., Chafik, L., Johannessen, T., Reverdin, G., and Omar, A.: Wintertime fCO(2) Variability in the Subpolar North Atlantic Since 2004, Geophys Res Lett, 46, 1580-1590, 2019.

Li, H. M., Ilyina, T., Muller, W. A., and Sienz, F.: Decadal predictions of the North Atlantic $CO_2$ uptake, Nat Commun, 7, 2016.

References listed in response to reviewer 2: Bakker, D. C. E., Pfeil, B., Landa, C. S., Metzl, N., O'Brien, K. M., Olsen, A., et al, 2016: A multi-decade record of high-quality fCO2 data in version 3 of the Surface Ocean CO2 Atlas (SOCAT), Earth Syst. Sci.

Data, 8, 383-413, doi:10.5194/essd-8-383-2016, 2016

Bennington, V., McKinley, G. A., Dutkiewicz, S. & Ullman, D. What does chlorophyll variability tell us about export and air-sea CO2 flux variability in the North Atlantic? Global Biogeochem. Cycles 23, GB3002 (2009).

Breeden, M. L. and McKinley, G. A.: Climate impacts on multidecadal pCO2 variability in the North Atlantic: 1948–2009, Biogeosciences, 13, 3387-3396, https://doi.org/10.5194/bg-13-3387-2016, 2016.

Corbière, A., N. Metzl, G. Reverdin, C. Brunet and T. Takahashi, 2007. Interannual and decadal variability of the oceanic carbon sink in the North Atlantic subpolar gyre. Tellus B, Vol. 59, issue 2, 168-179, DOI: 10.1111/j.1600-0889.2006.00232 Couldrey, M. P., Oliver, K. I. C., Yool, A., Halloran, P. R. and Achterberg, E. P.(2019). Drivers of 21st Century carbon cycle variability in the NorthAtlantic Ocean. Biogeosciences Discussions, 1 33. doi:10.5194/bg 2019 16. Fay, A. R., and G. A. McKinley, 2013, Global trends in surface ocean pCO2 from in situ data, Global Biogeochem. Cycles, 27, doi:10.1002/gbc.20051.

Fröb, F., Olsen, A., Becker, M., Chafik, L., Johannessen, T., Reverdin, G., and Omar, A.: Wintertime fCO(2) Variability in the Subpolar North Atlantic Since 2004, Geophys Res Lett, 46, 1580-1590, 2019.

García-Ibáñez, M. I., Zunino, P., Fröb, F., Carracedo, L. I., Ríos, A. F., Mercier, H., Olsen, A., and Pérez, F. F.: Ocean acidification in the subpolar North Atlantic: rates and mechanisms controlling pH changes, Biogeosciences, 13, 3701-3715, https://doi.org/10.5194/bg-13-3701-2016, 2016.

Gruber, N., D. Clement, B. R. Carter, R. A. Feely, S. van Heuven, M. Hoppema, M. Ishii, R. M. Key, A. Kozyr, S. K. Lauvset, C. Lo Monaco, J. T. Mathis, A. Murata, A. Olsen, F. F. Perez, C. L. Sabine, T. Tanhua, and R. Wanninkhof (2019). The oceanic sink for anthropogenic CO2 from 1994 to 2007, Science vol. 363 (issue 6432), pp. 1193-1199.
DOI: 10.1126/science.aau5153

Keller K., F. Joos, C. Raible, V. Cocco, T. Frolicher, J. Dunne, M. Gehlen, L. Bopp, J. Orr, J. Tjiputra, C. Heinze, J. Segscheider, T. Roy and N. Metzl, 2012. Variability of the Ocean Carbon Cycle in Response to the North Atlantic Oscillation. Tellus B, 64, 18738, doi:10.3402/tellusb.v64i0.18738.

Landschützer, P., Ilyina, T., & Lovenduski, N. S. (2019). Detecting regional modes of variability in observation-based surface ocean pCO2. Geophysical Research Letters, 46. https://doi.org/10.1029/2018GL081756

Loveday, B. R. and Smyth, T.: A 40-year global data set of visible-channel remote-sensing reflectances and coccolithophore bloom occurrence derived from the Advanced Very High Resolution Radiometer catalogue, Earth Syst. Sci. Data, 10, 2043-2054, https://doi.org/10.5194/essd-10-2043-2018, 2018.

McKinley, G. A., A. R. Fay, T. Takahashi and N. Metzl, 2011. Convergence of atmospheric and North Atlantic carbon dioxide trends on multidecadal timescales. Nature Geoscience. doi:10.1038/NGEO1193

McKinley, G.A., A.R. Fay, N. Lovenduski, and D.J. Pilcher, 2017. Natural variability and anthropogenic trends in the ocean carbon sink. Annual Review Marine Science. doi: 10.1146/annurev-marine-010816-060529

McKinley, G. A., Ritzer, A. L., and Lovenduski, N. S.: Mechanisms of northern North Atlantic biomass variability, Biogeosciences, 15, 6049-6066, https://doi.org/10.5194/bg-15-6049-2018, 2018.

Metzl, N., A Corbière, G. Reverdin, A. Lenton, T. Takahashi, A. Olsen, T. Johannessen, D. Pierrot, R. Wanninkhof , S. R. Ólafsdóttir, J. Olafsson and M. Ramonet, 2010. Recent acceleration of the sea surface fCO2 growth rate in the North Atlantic subpolar gyre (1993-2008) revealed by winter observations, Global Biogeochem. Cycles, 24, GB4004, doi:10.1029/2009GB003658.

Olsen, A., R. M. Key, S. van Heuven, S. K. Lauvset, A. Velo, X. Lin, C. Schirnick, A. Kozyr, T. Tanhua, M. Hoppema, S. Jutterström, R. Steinfeldt, E. Jeansson, M. Ishii, F. F. Pérez & T. Suzuki, 2016. An internally consistent data product for the world ocean: the Global Ocean Data Analysis Project, version 2 (GLODAPv2), Earth Syst. Sci. Data, 8, 297-323, doi:10.5194/essd-8-297-2016.

Pérez, F. F., Vázquez-Rodríguez, M., Mercier, H., Velo, A., Lherminier, P., and Ríos, A. F.: Trends of anthropogenic CO2 storage in North Atlantic water masses, Biogeosciences, 7, 1789-1807, https://doi.org/10.5194/bg-7-1789-2010, 2010.

Pérez et al. 2013, Atlantic Ocean CO2 uptake reduced by weakening of the meridional overturning circulation Nature Geoscience, 6 doi: 10.1038/ngeo1680

Pilcher, D. J., Brody, S. R., Johnson, L., Bronselaer, B., 2015. Assessing the abilities of CMIP5 models to represent the seasonal cycle of surface ocean pCO2. Journal of Geophysical Research Oceans, 120, 7, 4625–4637, doi: 10.1002/2015JC010759.

Racapé, V., N. Metzl, C. Pierre, G. Reverdin, P.D. Quay, S. R. Olafsdottir, 2014. The seasonal cycle of the d13CDIC in the North Atlantic Subpolar Gyre. Biogeosciences, 11, 6, 1683-1692, doi:10.5194/bg-11-1683-2014.

Reverdin, G., Metzl, N., Olafsdottir, S., Racapé, V., Takahashi, T., Benetti, M., Valdimarsson, H., Benoit-Cattin, A., Danielsen, M., Fin, J., Naamar, A., Pierrot, D., Sullivan, K., Bringas, F., and Goni, G.: SURATLANT: a 1993–2017 surface sampling in the central part of the North Atlantic subpolar gyre, Earth Syst. Sci. Data, 10, 1901-1924, https://doi.org/10.5194/essd-10-1901-2018, 2018a.

Reverdin, G., Valdimarsson, H., Alory, G., Diverres, D., Bringas, F., Goni, G., Heilmann, L., Chafik, L., Szekely, T., and Friedman, A. R.: North Atlantic subpolar gyre along predetermined ship tracks since 1993: a monthly data set of surface temperature, salinity, and density, Earth Syst. Sci. Data, 10, 1403-1415, https://doi.org/10.5194/essd-10-1403-2018, 2018b.

Rödenbeck, C., R.F. Keeling, D.C.E. Bakker, N. Metzl, A. Olsen, C.Sabine, and M. Heimann, 2013. Global surface-ocean pCO2 and sea-air CO2 flux variability from an observation-driven ocean mixed-layer scheme. Ocean Science, 9, 193-216. doi:10.5194/osd-9-193-2013.

Rödenbeck, C., D. C. E. Bakker, N. Metzl, A. Olsen, C. Sabine, N. Cassar, F. Reum, R. F. Keeling, and M. Heimann, 2014. Interannual sea–air CO2 flux variability from an observation-driven ocean mixed-layer scheme. Biogeosciences, 11, 4599-4613, 2014 doi:10.5194/bg-11-4599-2014.

Schuster, U., A.J. Watson, N. Bates, A. Corbière, M. Gonzalez-Davila, N.Metzl, D. Pierrot and M. Santana-Casiano, 2009. Trends in North Atlantic sea surface pCO2 from 1990 to 2006. Deep-Sea Res II, doi:10.1016/j.dsr2.2008.12.011

Schuster, U., G. A. McKinley, N. Bates, F. Chevallier, S. C. Doney, A. R. Fay, M. González-Dávila, N. Gruber, S. Jones, J. Krijnen, P. Landschützer, N. Lefèvre, M. Manizza, J. Mathis, N. Metzl, A. Olsen, A. F. Rios, C. Rödenbeck, J. M. Santana-Casiano, T. Takahashi, R. Wanninkhof, and A. J. Watson, 2013. An assessment of the Atlantic and Arctic sea-air CO2 Fluxes, 1990-2009. Biogeosciences, 10, 607-627, doi:10.5194/bg-10-607-2013.

Signorini, S. R. , S. Häkkinen, K. Gudmundsson, A. Olsen, A. M. Omar, J. Olafsson, G. Reverdin, S. A. Henson, C. R. McClain and D. L. Worthen, 2012. The Role of Phytoplankton Dynamics in the Seasonal and Interannual Variability of Carbon in the Subpolar North Atlantic:Âň A Modeling Study. Geosci. Model Dev., 5, 683-707, doi:10.5194/gmd-5-683-2012

Takahashi, T., S C. Sutherland, R.Wanninkhof, C. Sweeney, R.A. Feely, D. Chipman, B. Hales, G. Friederich, F. Chavez, A. Watson, D. Bakker, U. Schuster, N.Metzl, H.Y. Inoue, M. Ishii, T. Midorikawa, C.Sabine, M. Hoppema, J.Olafsson, T. Amarson, B.Tilbrook, T. Johannessen, A. Olsen, R. Bellerby, Y. Nojiri, C.S. Wong, B. Delille, N. Bates and H. De Baar, 2009. Climatological Mean and Decadal Change in Surface

Ocean pCO2, and Net Sea-air CO2 Flux over the Global Oceans. Deep-Sea Res II, doi:10.1016/j.dsr2.2008.12.009

Tesdal, J.E., Abernathey, R.P., Goes, J.I., Gordon, A.L., Haine, T.W.N., 2018. Salinity trends within the upper layers of the Subpolar North Atlantic. Journal of Climate 31, 2675{2698. doi:10.1175/JCLI-D-17-0532.1.

Thierry, V., de Boisséson, E., Mercier, H. Interannual variability of the Subpolar Mode Water properties over the Reykjanes Ridge during 1990-2006. Journal of Geophysical Research, 113, doi:10.1029/2007JC004443.

Thomas, H., A.E.F. Prowe, I.D. Lima, S.C. Doney, R. Wanninkhof, R.J. Greatbatch, U. Schuster and A. Corbière, 2008. Changes in the North Atlantic Oscillation influence CO2 uptake in the North Atlantic over the past two decades, Global Biogeochem. Cycles, 22, GB4027, doi:10.1029/2007GB003167

Ullman, D. J., G.A. McKinley, V. Bennington and S. Dutkiewicz, 2009. Trends in the North Atlantic carbon sink: 1992-2006, Global Biogeochem. Cycles, 23, GB4011, doi:10.1029/2008GB003383.

Watson, A.J., U. Schuster, D .C. E. Bakker, N. Bates, A. Corbiere, M. Gonzalez-Davila, T. Freidrich, J. Hauck, C. Heinze, T. Johannessen, A. Koertzinger, N. Metzl, J. Olafsson, A. Olsen, A. Oschlies, X. Padin, B. Pfeil, A. Rios, M Santana-Casiano, T. Steinhoff, M. Telszewski, D. W. R. Wallace, R. Wanninkhof, 2009. Tracking the variable North Atlantic sink for atmospheric CO2 , Science ,326, 1391, doi:10.1126/science.1177394.

[Figure]

[Figure]

Figure 1. Effect of the changes in total (filled), anthropogenic (doted) and natural (empty) DIC to the trends in surface pH during summer for the three periods and different boxes (B, C, D-E and C-D-E).

**Fig. 1.**

[Figure]

Figure 2. Effect of the changes in total (filled), anthropogenic (doted) and natural (empty) DIC to the trends in surface pH during winter for the three periods and different boxes (B, C, D-E and C-D-E).

**Fig. 2.**

[Figure]

Figure 3. a) Temperature potential of Modal Water (Reykjanes Ride, cross), associated with NAO index (grey bar). Thierry et al. 2007; extended data, V. Thierry pers. comm. 19/06/2019. b) Sea Surface Temperature of C-D-E region (Jan-Fev, dot) estimate with AX2-binned (Reverdin et al. 2018b)

**Fig. 3.**

[Figure]

Figure 4. Index of variability of the thickness of the mode water layer (indicative of winter mixing layer (MLD), Reykjanes Rides. Thierry et al. 2007; extended data, V. Thierry pers. comm. 19/06/2019).

**Fig. 4.**

[Figure]

Figure 5. Decomposition of the trends in surface pH for winter (a,b,c) and summer (d,e,f). The effect of the changes in SSS, SST, TA and DIC is shown for the three periods.

**Fig. 5.**

Table 1. Trends (per year) evaluated from data presented in the manuscript (with some corrections). Here we show results for boxes B, C-D-E (as in the sub. paper) plus boxes C and D-E (as suggested by the reviewer). In bold are represented significant trends.

| | | box | SST (°C/yr) | SSS (/yr) | TA (µmol/kg/yr) | DIC (µmol/kg/yr) | pH (/yr) | fCO₂ (µatm/yr) | Ωca (/yr) | Ωar (/yr) |
|---|---|---|---|---|---|---|---|---|---|---|
| 1993-1996 | summer | B | -0.10 ±0.16 | 0.041 ±0.034 | 1.8 ±1.6 | **9.0 ±2** | **-0.0143 ±0.0031** | **12.0 ±2.6** | **-0.107 ±0.015** | **-0.069 ±0.01** |
| | | C.D.E | **0.68 ±0.06** | **0.032 ±0.028** | 2.2 ±1.3 | 2.4 ±2.3 | **-0.0123 ±0.0031** | **10.6 ±2.4** | 0.003 ±0.025 | 0.005 ±0.016 |
| | | C | **0.58 ±0.06** | -0.020 ±0.015 | -0.6 ±0.7 | 2.3 ±2 | **-0.0155 ±0.003** | **13.6 ±2.6** | -0.038 ±0.023 | -0.022 ±0.015 |
| | | D.E | **0.72 ±0.08** | 0.059 ±0.04 | 3.6 ±1.9 | 2.4 ±3.5 | **-0.0103 ±0.0046** | **8.8 ±3.5** | 0.026 ±0.038 | 0.020 ±0.024 |
| 1994-1997 | winter | B | -0.19 ±0.3 | **0.045 ±0.032** | **2.2 ±1.5** | 0.1 ±1.8 | 0.0075 ±0.0037 | -7.2 ±3.6 | 0.028 ±0.022 | 0.017 ±0.015 |
| | | C.D.E | 0.06 ±0.1 | -0.009 ±0.015 | **-0.5 ±0.6** | **-0.9 ±0.5** | 0.0002 ±0.001 | -0.4 ±1 | 0.006 ±0.01 | 0.004 ±0.006 |
| | | C | 0.07 ±0.17 | -0.003 ±0.021 | -0.5 ±1 | **-1.7 ±1.3** | **0.0021 ±0.0019** | **-2.3 ±1.9** | 0.017 ±0.015 | 0.011 ±0.01 |
| | | D.E | 0.05 ±0.1 | **-0.012 ±0.013** | **-0.5 ±0.6** | **-0.4 ±0.5** | -0.0007 ±0.0011 | 0.5 ±1.2 | 0.000 ±0.01 | 0.000 ±0.007 |
| 2001-2007 | summer | B | 0.07 ±0.09 | **-0.030 ±0.013** | **-4.3 ±0.7** | 0.2 ±1.2 | **-0.0105 ±0.0012** | **8.5 ±1** | **-0.068 ±0.015** | **-0.043 ±0.01** |
| | | C. D. E | 0.01 ±0.04 | **0.017 ±0.008** | **-1.3 ±0.5** | **4.9 ±0.7** | **-0.0130 ±0.0013** | **11.5 ±1.1** | **-0.093 ±0.011** | **-0.059 ±0.007** |
| | | C | -0.07 ±0.08 | **0.012 ±0.009** | **-1.2 ±0.6** | **5.1 ±1** | **-0.0125 ±0.0015** | **11.2 ±1.3** | **-0.095 ±0.015** | **-0.061 ±0.01** |
| | | D.E | **0.05 ±0.04** | **0.019 ±0.008** | **-1.4 ±0.7** | **4.8 ±0.9** | **-0.0133 ±0.0019** | **11.6 ±1.5** | **-0.092 ±0.015** | **-0.059 ±0.01** |
| 2002-2007 | winter | B | 0.04 ±0.11 | **-0.020 ±0.008** | -0.2 ±0.4 | **2.1 ±1.2** | **-0.0061 ±0.0014** | **6.2 ±1.3** | -0.031 ±0.018 | -0.020 ±0.011 |
| | | C. D. E | -0.05 ±0.07 | -0.002 ±0.012 | **-1.9 ±0.7** | **1.6 ±0.4** | **-0.0074 ±0.0009** | **7.0 ±0.9** | **-0.051 ±0.006** | **-0.032 ±0.004** |
| | | C | **-0.19 ±0.09** | **-0.015 ±0.011** | **-2.9 ±0.9** | **2.2 ±0.5** | **-0.0088 ±0.0013** | **7.9 ±1.3** | **-0.074 ±0.01** | **-0.047 ±0.006** |
| | | D.E | 0.02 ±0.07 | 0.004 ±0.016 | -1.3 ±0.8 | **1.4 ±0.6** | **-0.0065 ±0.0011** | **6.3 ±1.2** | **-0.039 ±0.005** | **-0.024 ±0.003** |
| 2008-2017 | summer | B | -0.10 ±0.06 | -0.012 ±0.009 | **1.4 ±0.7** | -0.3 ±0.8 | **0.0052 ±0.0013** | **-4.3 ±1.1** | **0.026 ±0.011** | **0.016 ±0.007** |
| | | C. D. E | **-0.10 ±0.03** | **-0.012 ±0.004** | **2.0 ±0.3** | **1.4 ±0.5** | 0.0027 ±0.001 | -2.0 ±0.8 | 0.010 ±0.008 | 0.006 ±0.005 |
| | | C | **-0.14 ±0.05** | **-0.014 ±0.005** | **1.6 ±0.4** | 1.2 ±0.7 | 0.0030 ±0.0012 | -2.5 ±1.1 | 0.005 ±0.01 | 0.003 ±0.006 |
| | | D.E | **-0.07 ±0.05** | **-0.013 ±0.004** | **2.2 ±0.3** | 1.5 ±0.7 | 0.0021 ±0.0014 | -1.5 ±1.2 | 0.011 ±0.013 | 0.007 ±0.008 |
| 2008-2015 | winter | B | 0.02 ±0.07 | -0.004 ±0.009 | 0.4 ±0.5 | 1.3 ±0.9 | -0.0023 ±0.0012 | 2.2 ±1.2 | -0.011 ±0.01 | -0.007 ±0.006 |
| | | C. D. E | -0.07 ±0.04 | -0.007 ±0.005 | 0.3 ±0.4 | **1.1 ±0.4** | -0.0007 ±0.0005 | 0.6 ±0.5 | -0.011 ±0.005 | -0.007 ±0.003 |
| | | C | -0.11 ±0.05 | -0.006 ±0.008 | 0.4 ±0.6 | 1.4 ±0.6 | -0.0008 ±0.0008 | 0.6 ±0.8 | -0.015 ±0.006 | -0.010 ±0.004 |
| | | D.E | -0.05 ±0.03 | **-0.007 ±0.003** | 0.3 ±0.4 | 0.9 ±0.5 | -0.0006 ±0.0006 | 0.5 ±0.6 | **-0.008 ±0.005** | **-0.005 ±0.003** |
| 1993-2017 | summer | B | **0.05 ±0.02** | **-0.003 ±0.002** | 0.1 ±0.2 | **0.5 ±0.2** | **-0.0017 ±0.0004** | **1.5 ±0.3** | -0.005 ±0.003 | -0.003 ±0.002 |
| | | C | 0.02 ±0.01 | **0.003 ±0.001** | **0.4 ±0.1** | **1.0 ±0.2** | **-0.0019 ±0.0004** | **1.7 ±0.3** | **-0.009 ±0.003** | **-0.005 ±0.002** |
| | | D.E | **0.04 ±0.01** | **0.003 ±0.002** | **0.3 ±0.1** | **0.9 ±0.2** | **-0.0020 ±0.0004** | **1.9 ±0.3** | **-0.009 ±0.003** | **-0.005 ±0.002** |
| | | C.D.E | **0.03 ±0.01** | **0.003 ±0.001** | **0.3 ±0.1** | **0.9 ±0.1** | **-0.0020 ±0.0004** | **1.8 ±0.2** | **-0.009 ±0.002** | **-0.005 ±0.001** |
| | | B. C. D. E | **0.04 ±0.01** | 0.000 ±0.002 | **0.2 ±0.1** | **0.7 ±0.1** | **-0.0018 ±0.0002** | **1.7 ±0.2** | **-0.007 ±0.002** | **-0.004 ±0.001** |
| 1993-2015 | winter | B | **0.02 ±0.02** | 0.003 ±0.002 | 0.1 ±0.1 | **0.6 ±0.2** | **-0.0016 ±0.0003** | **1.5 ±0.3** | **-0.006 ±0.002** | **-0.004 ±0.001** |
| | | C | -0.02 ±0.01 | **0.005 ±0.002** | 0.0 ±0.1 | **0.9 ±0.1** | **-0.0018 ±0.0002** | **1.7 ±0.1** | **-0.012 ±0.002** | **-0.007 ±0.001** |
| | | D.E | **0.02 ±0.01** | **0.005 ±0.001** | **0.2 ±0.1** | **0.5 ±0.1** | **-0.0013 ±0.0001** | **1.3 ±0.1** | **-0.005 ±0.001** | **-0.003 ±0.001** |
| | | C.D.E | **0.01 ±0.01** | **0.005 ±0.001** | 0.1 ±0.1 | **0.6 ±0.1** | **-0.0014 ±0.0001** | **1.4 ±0.1** | **-0.007 ±0.001** | **-0.004 ±0.001** |
| | | B. C. D. E | **0.01 ±0.01** | **0.004 ±0.001** | 0.1 ±0.1 | **0.6 ±0.1** | **-0.0015 ±0.0001** | **1.5 ±0.1** | **-0.007 ±0.001** | **-0.004 ±0.001** |

**Fig. 6.**

---

## Author Response (AR1)

This letter lists the main changes we have made in the manuscript to take into account the two reviews and provides a list of all relevant changes made in the manuscript.

More details of our reply to the reviewer comments are given on the BGD site.

In the initial submitted manuscript the results were presented following only two sections: Trends in region C-D-E and Trends in region B. Following the suggestions from the reviewers and to clarify the results and discussion, the sections of the manuscript have been revised and rearranged as follows:

Introduction
Material and Methods
      2.1 Data collection and measurements
      2.2 Calculations of the carbonate system parameters and contributions
      2.3 Data selection: regions and seasons
Results
      3.1 Seasonal cycle
      3.2 Long-term trend and anthropogenic $CO_2$
      3.3 Winter trends at different periods
      3.4 Summer trends at different periods
Discussion
Conclusion and perspectives

**1. Data**

A reviewer seems disappointed with the new data and results that would offer "practically nothing new" with respect to previous work. In the present study, we added ten years of data (2008-2017) which offer new and complementary results. The paper by Reverdin et al. (ESSD, 2018b) aimed at describing in detail the methods, accuracy and data for the full period 1993-2017 (with some data revisited and corrected): Reverdin et al. (2018b) did not investigate internal processes or external forcing resulting in the observed temporal changes of the properties (SST, SSS, nutrients, TA and DIC, $\delta^{18}O$ and $\delta^{13}C$). The present paper is aimed at better understanding the changes in the carbonate system over the whole period (1993-2017) and for different regions and periods, including summer-time (not evaluated in previous work, Metzl et al., 2010) as well as for pH and Omega trends. This is more clearly stated in the revision. Note that the new data for 2008-2017 were obtained when NAO and AMO variability were large and this is now discussed in more details as recommended by both reviewers. We also think that although challenging, it is important to start evaluating and, when possible, explain the trends observed during the summer season.

**Seasonal cycle and the selection of periods:**

A reviewer questioned whether the seasonal cycle should be presented in the submitted manuscript. We agree with the reviewer that the seasonal cycle is not a prime objective of the study. However, for the trend analysis, as we don't always have observations in February (winter) or July (summer), the climatological seasonal cycle is used to group data of different months (such as JFM and JJA) in order to best estimate the trends and drivers for different seasons. We thus briefly introduce and describe the seasonality in section 3.1. In years when observations were not collected in July or February (the months selected to represent summer and winter trend), we use data collected in the previous or next month by correcting for the climatological seasonal change between successive months. As suggested by the reviewers we verified that this can be done with the more regularly-observed SST and SSS data, based on the binned monthly products constructed by Reverdin et al. (2018a). We also used this new binned-product to select the three periods investigated in more detail (1993-1997, 2001-2007 and 2008-2017), in connection with NAO and AMO indices and added a new figure to present both SST, SSS anomalies and NAO/AMO indices (new Fig. 3).

**2. Method**

Are Olsen suggested to explicitly take into account salinity changes in the $pCO_2$ driver decomposition. In the revised manuscript, we considered a new equation of attribution for SST, SSS, DIC and TA effects on pH, $fCO_2$ and $\Omega$. We now consider separately the effect of salinity and sDIC, sTA (DIC and TA normalized) as recommended. Therefore, we have added trend estimates for sDIC and sTA in Table 2, present sDIC and sTA in Figure 4, and added the effects of sDIC and sTA in Figure 5 (as drivers).

Also, as mentioned by Are Olsen, uncertainties were not properly dealt with. To correct this error, we used the most recent CO2SYS software provided by James Orr (as this includes error propagation), and propagated the errors in $fCO_2$ (and others terms) for the trends using a Monte Carlo method. For that, we used the measurement errors of each input parameter (SST, SSS, DIC, TA, nutrients).

**3. Results and discussion (processes, NAO, AMO):**

A reviewer found that the relationship between the drivers and the main processes occurring in North Subpolar gyre was unfortunately very poorly developed partly because "the authors seem to be unaware of key articles that have demonstrated the main patterns of variation linked to the NAO (Thomas et al. 2008, Keller et al. 2012, Schuster et al. 2013 and Pérez et al. 2013)". In terms of acidification rates, the article by Garcia-Ibañez et al. (2016) was also ignored.

As recommended by the two reviewers we have extended the discussion including the link (or the absence of a link) with NAO and AMO and added new references as suggested, either based on observations or models. Note that recent publications, not available when we submitted our manuscript, have also been added to better discuss and compare our results (e.g. Lebehot et al., 2019; Omar et al., 2019; Macovei et al., 2019; Holliday et al., 2019; Fröb et al 2019)

With respect to the processes, we discussed in more detail the different drivers for each period and when possible referred to observed change in productivity (for summer) or convection/mixing (for winter). In addition, as we observed significant changes in the TA trends, we discussed this result in more details in the revised version and suggest a possible link with calcification (although not yet quantify but might be relevant for further studies).

**4. Anthropogenic versus non-anthropogenic signal issue**

The reviewers asked to better separate the anthropogenic and non-anthropogenic signals. The first version of this paper did not separate anthropogenic and natural signals of $CO_2$. In the new version, we have estimated and compared the anthropogenic signal of DIC in the NASPG based on different data and methods (Glodap-V2, Gruber et al., 2019). This is now described and discussed in Section 3.2 for the long-term trend analysis (1993-2017) as well as in the Discussion, section 4. We also evaluate and show the impact of both anthropogenic (C-ant) and natural (DIC-nat) signals to explain the drivers and a better identification of the period when the natural variability dominates. A new figure (Fig. 6) is now added to show these effects for long-term, each period and each region.

**5. Figures and Tables**

In the revised manuscript we have prepared:

A new single figure (Fig. 1) as recommended, which summarized the SURATLANT cruises track and boxes considered for the trend analysis.

In Figure 2 we have added the atmospheric $fCO_2$ to highlight the ocean $CO_2$ source/sink seasonality.

A new Figure 3 presents the SST and SSS anomalies for winter along the SURATLANT track (based on binned products constructed by Reverdin et al. (2018a). This also shows the 3 different periods selected for the trend analysis. Figure 3c also includes the NAO and AMO index during this period and used in the description of the results and in the discussion section.

In Figure 4 we added a figure of the air-sea $fCO_2$ differences (Fig. 4h) as suggested by Are Olsen to illustrate the change of the ocean $CO_2$ sink that is discussed in more detail in the revised manuscript (including comparison with other recent results, Lebehot et al., 2019; Denvil-Sommer et al., 2019 among others).

In Figure 4 we also added normalized DIC and TA (Fig. 4d, f).

In Figure 5 we now add the effects of sDIC and sTA for each period and region.

A new Figure 6 presents the decomposition of the trends for pH according to the effect of the changes in total, natural and anthropogenic DIC.

Table 2: All values have been recalculated according to the error analysis and selection of data (e.g. for summer 2007-2017 the data in July 2013 are not taken into account; this is explained in the manuscript).

Table 2: Trends for sDIC and sTA have been added.

[revised manuscript text omitted]

---

## Referee Report (RR1)

I thank Leseurre et al for providing the revised manuscript. While clearly improved, there are still some issues that needs to be dealt with before I can recommend that the manuscript is accepted for publication.

**Comments on responses to major issues 1-10 in initial review.**

1. *Testing the scheme to reconstruct February and July data.*

   I proposed to test this scheme using the high frequency SST and SSS data collected by the TSG. This has not been done. While a figure showing the trends in higher frequency TSG SST and SSS data have been added to the manuscript (Fig 3), this does not constitute an actual test of the reconstruction scheme. Please add and discuss a side-by-side comparison of trends in SST and SSS as measured by the TSG in February and July, vs. those shown in Fig 4 based on a mixture of measured and reconstructed data. This could be in the form of a figure or a table.

   I see that there are clear discrepancies between the SST and SSS trends in Fig 3 and 4 for some of the time periods. The 1993-1997 period, the data in Fig 3 shows increasing SST and SSS, while for the corresponding period in Fig 4, SST has no trend and SSS is decreasing. The winter SST trends for the second period (2001-2007) are also different (increasing in Fig 3 decreasing in Fig 4).

2. *Comparison with trends in mapped products*

   I asked for a comparison with SST trends from the NOAA objectively analysed product; this would shed light on how representative the data (in Fig. 4) are of large-scale interannual phenomena. In their response the authors states this will be included: "We will also use SST reanalysis to confirm the trends for the selected periods". This has not been included (i.e. comparison to regional SST trends from NOAA OI SST product (https://climatedataguide.ucar.edu/climate-data/sst-data-noaa-optimal-interpolation-oi-sst-analysis-version-2-oisstv2-1x1)

Overall, for comment 1 & 2. I would recommend to add a subsection in the Results section, between current 3.1 and 3.2, that compares the SST and SSS trends as presented in Fig 4, with those from higher frequency TSG data and also regional SST trends from the NOAA OI product. This would, before anything else, inform readers about how accurate the trends shown in Fig 4 are, and the extent to which they represent larger scale interannual variations.

3. *pCO2 driver decomposition*

   It is good to see that a decomposition of the drivers of pCO2 and pH trends have now been included. However, please include citations for the equation (e.g. Keeling et al (2004) and Fröb et al. (2019)). Also, it is not clear what parts the columns for DIC/sDIC and TA/sTA in Fig 5 show.

Specifically, is DIC the total DIC driver while sDIC is the part of the DIC driver not related to salinity? Please specify this in the caption.

4. *Trends in salinity normalised DIC and TA*

   I am happy with the response and actions

5. *Inclusion of figure with trends in air-sea pCO2 difference*

   This is fine

6. *Uncertainties*

   Please explain in the methods section how the uncertainty of the trends presented in Fig 4 was determined.

7. *Trend 2001-2008 and reconstructed data*

   OK, would be worthwhile to mention this

8. *Anthropogenic vs natural carbon*

   This has been addressed nicely

9. *Discussion*

   This is now very exhaustive

10. *Long term trend*

    OK

**New comments**

**Page 1 line 18:** 'the trend of ...CO2....is slightly less than the atmospheric....and the pH decrease.' Odd sentence, please revise.

**Page 1 line 28:** Consider using 'multiannual' instead of 'pluriannual'

**Page 2 line 3:** replace 'indices' with 'values'

**Page 2 line 33-43:** The paper by Goris et al: Constraining projection-based estimates of the future North Atlantic carbon uptake, *Journal of Climate, 31*, 3959-3978, 2018, shows clearly that many ESM struggle to represent mixing correctly in the NASPG, which render them unable to reproduce the actual trends.

**Page 3 line 7-9:** It is not so much the shoaling of the lysocline that is the problem, but that these northern surface waters have initially low CaCO3 saturation because of high natural (preindustrial) DIC concentrations because of low temperatures giving high CO2 solubility. Please revise.

**Page 3 line 35**: Replace 'conducts' with 'conduits'

**Page 5, line 5**: State the pressure used, i.e. 1013.25 hPa

**Page 5**: Please add a subsection on how uncertainties were determined: the uncertainties of the trend slopes, the uncertainties of the driver decomposition (i.e. the 1000 random perturbations) and any other uncertainty calculation. For the uncertainties in the trend regression, how did you deal with the uncertainty of each point in the y direction (e.g. the SST values for each year has a standard deviation associated with them shown as error bars in Fig 4(a), how was this accounted for when calculating the regression and its uncertainty? As far as I know this is not straightforward).

**Page 6 line 5:** You used the seasonal climatology to adjust data from January/March and June/August to February and July, not 'to complement winter and summer'. Please state here the typical magnitude of these adjustments.

**Page 6 line 16-17:** Salinity is also low in the green box, but DIC is not correspondingly lower. I.e. the low DIC in the southern (red) box are not only a result of the low salinities, please explain more.

**Page 6 line 30:** 'Section (3.3)' should be 'Section 2.3'

**Page 6:** It would be good to include the uncertainty of the trends that are stated.

**Page 7:** Use of TrOCA method. The TrOCA method has issues and it is not very accurate. See for example https://www.biogeosciences.net/7/723/2010/bg-7-723-2010.pdf. Some cautionary remarks would be appropriate, for example that you believe it is robust enough to determine Cant trends even if the absolute values are questionable.

**Page 7 line 7:** The DIC trend of 0.9 from GLODAPv2 – was this something you calculated. What were the lat/lon bounds

**Page 7 line 7:** GLODAPv2, not GLODAP-V2

**Page 7 line 16:** replace 'very homogeneous' with 'a'

**Page 7 lines 11-18.** I don't understand some of these numbers. From Gruber et al., 2019 you find a Cant increase between 1994 and 2007 of 8.5 μmol/kg. This gives a trend of 0.65 μmol/kg yr. You assume this applies for the surface as well, to give a total increase of 10.1 μmol/kg for 1994-2007. But why the change?, if

the subsurface trend from Gruber applies in the surface layer, this would give 8.5 µmol/kg there as well, not 10.1 µmol/kg.

My sense is that the Cant increase from Gruber et al (0.65 µmol/kg yr ) is similar to the one you estimate using TrOCA (0.6 µmol/kg yr), but both of these are lower than the observed DIC increase in SURATLANT (0.7-0.9 µmol/kg yr) and estimate from GLODAPv2 (0.9 µmol/kg yr). So altogether, the DIC is increasing slightly faster than explained by Cant alone. Please revise and simplify.

**Page 8 line 11:** Please simplify to: Because interannual variability is more pronounced in summer than in winter because of the added influence of biological activity.

**Page 8 line 14-15:** Please include citations to these studies that have evaluated trends in different seasons.

**Page 9 line 16:** Mention the source of these independent pCO2 observations (ship, principal investigators, cite any papers by PIs that describe the data).

**Page 10 line 14:** 'last decade , 2001-2017…', better with 'last period, 2008-2017 for summer and 2008-2015 for winter'.

**Page 10 line 29-30:** The case for low primary production in 1996, is very weak. 1996 was characterised by very negative NAO index. Another year with very low NAO index was 2010, and in that year primary production was exceptionally high. So if anything – by analogy – I would expect strong primary production in 1996.

**Page 11 line 2-4**: Attribution of summer 2005 - 2007 high fCO2 values. Low productivity and/or deep (winter) mixing suggested. This can and should be checked with satellite Chl and Argo MLD data. For the record, Fröb et al., 2016 (Irminger Sea deep convection injects oxygen and anthropogenic carbon to the ocean interior, *Nature Communications*, 7:13244, 2016) shows that winter mixing was *not* deep in these years in the central-west Irminger Sea. Therefore primary production is the more likely candidate.

**Page 11, line 18-20.** This is cherry picking, what about the earlier time periods when the discrete data does not reflect the trends in higher frequency TSG data (such as winter 1993-1998 noted earlier in this review)? The extent to which these discrete samples reflect larger scale variations should be investigated using a systematic approach – not an ad hoc as is currently done.

**Page 11 line 31.** Write as GLODAPv2.2019

**Page 12 line 32-33.** This is also related to the way models do/don't represent winter mixing, see earlier cited Goris et al paper.

**Page 13 line 10.** Last sentence needs revision.

**Page 14 lines 1-10.** The 2010 primary production event is nicely discussed in Henson et al, Unusual subpolar North Atlantic phytoplankton bloom in 2010, JGR, 2013

**Page 14 line 23-24.** Please name ship (Skogafoss) and principal investigator (Wanninkhof NOAA/AOML) of these data (it is always nice to give specific and concrete credit to the correct people)

**Page 14-15,** 'Such large changes in pH trends … would not have been resolved when using fCO2 data and TA/S relation instead of TA measurements'. I am not convinced this is actually the case. pH determined from fCO2 and TA is not very sensitive to TA, but depends most on fCO2, which in itself reflects underlying DIC/TA changes; pH and fCO2 are strongly correlated. I think such large changes could be visible in pH determined from fCO2 and TA (from TA-S relationship). You may want to check with a simple calculation before making such a statement. (i.e. combine the calculated fCO2 with TA from TA-S relationship and calculate pH trends.)

**Page 15, line 11** You state 'indirect methods' but refer to only one, Denvil-Sommer et al., 2019. This generalisation does not seem correct unless you compare with other mapped pCO2/flux products.

---

## Author Response (AR2)

Leseurre, C., Lo Monaco, C., Reverdin, G., Metzl, N., Fin, J., Olafsdottir, S., and Racapé, V.: Ocean carbonate system variability in the North Atlantic Subpolar surface water (1993–2017), Biogeosciences Discuss., https://doi.org/10.5194/bg-2019-119, in review, 2019.

5    **Referee #2 Are Olsen**                                           **First Major Revision**

**1) New responses to comments on responses to major issues 1-10 in initial review:**

"I thank Leseurre et al for providing the revised manuscript. While clearly improved, there are still some
10   issues that needs to be dealt with before I can recommend that the manuscript is accepted for publication."

We thank again Are Olsen for his fast-new review and comments that will be considered when revising the manuscript. Below we list our responses before preparing a revised manuscript.

15   "**1. Testing the scheme to reconstruct February and July data.**

I proposed to test this scheme using the high frequency SST and SSS data collected by the TSG. This has not been done. While a figure showing the trends in higher frequency TSG SST and SSS data have been added to the manuscript (Fig 3), this does not constitute an actual test of the reconstruction scheme. Please add and discuss a side-by-side comparison of trends in SST and SSS as measured by the TSG in
20   February and July, vs. those shown in Fig 4 based on a mixture of measured and reconstructed data. This could be in the form of a figure or a table.

I see that there are clear discrepancies between the SST and SSS trends in Fig 3 and 4 for some of the time periods. The 1993-1997 period, the data in Fig 3 shows increasing SST and SSS, while for the
25   corresponding period in Fig 4, SST has no trend and SSS is decreasing. The winter SST trends for the second period (2001-2007) are also different (increasing in Fig 3 decreasing in Fig 4)."

**Response**: We had thoroughly tested the scheme to reconstruct February and July data using the high frequency SST and SSS data collected by the TSG, but this was indeed not included in the new version of the article. We suggest adding a slide-by-side comparison of the reconstructed data and trends added
30   with SST and SSS measured by the TSG as a supplement (cf attached figure S1-1-2 and table S1).

"**2. Comparison with trends in mapped products.**

I asked for a comparison with SST trends from the NOAA objectively analysed product; this would shed light on how representative the data (in Fig. 4) are of large-scale interannual phenomena. In their
35   response the authors states this will be included: "We will also use SST reanalysis to confirm the trends for the selected periods". This has not been included (i.e. comparison to regional SST trends from NOAA OI SST product (https://climatedataguide.ucar.edu/climate-data/sst-data-noaaoptimal-interpolation-oi-sst-analysis-version-2-oisstv2-1x1).

Overall, for comment 1 & 2. I would recommend to add a subsection in the Results section, between
40   current 3.1 and 3.2, that compares the SST and SSS trends as presented in Fig 4, with those from higher frequency TSG data and also regional SST trends from the NOAA OI product. This would, before anything else, inform readers about how accurate the trends shown in Fig 4 are, and the extent to which they represent larger scale interannual variations."

**Response**: Actually, we omitted to include a comparison of our SST trends with SST trends from the NOAA objectively analysed product. We are sorry for this oversight, which will be corrected by supplementary material in the revised manuscript. We will present the mapped regional SST trends (from NOAA OI SST product) for the three periods (winter and summer, corresponding to February and July; cf attached figure S2-a-b). All together, they are rather similar (along the track) to the ones from the high-frequency SST analyses along the track that were presented on figure 3, and provide some idea of spatial coherence of these SST trends.

On the other hand, there are some larger differences between the SST trends of Figure 3 or from NOAA OI-SST V2 product and the ones constructed with the data in this paper. This is particularly the case for short periods such as in 1993-1996 (or 1994-1997), as is expected, in particular for SST, due to aliasing in Suratlant data of the rather large higher frequency variability, such as between successive months. Also, it does not overwhelm the longer-term variability for the two other periods, such as the cooling in both seasons during the last period. As was commented in Reverdin et al. (2018), this sampling-induced error on the trend is less of an issue for surface salinity.

"**3. pCO2 driver decomposition**

It is good to see that a decomposition of the drivers of pCO2 and pH trends have now been included. However, please include citations for the equation (e.g. Keeling et al (2004) and Fröb et al. (2019)). Also, it is not clear what parts the columns for DIC/sDIC and TA/sTA in Fig 5 show. Specifically, is DIC the total DIC driver while sDIC is the part of the DIC driver not related to salinity? Please specify this in the caption."

**Response**: Again, we thank you for your insightful comments. We will include citations of Keeling et al. (2004) and Fröb et al. (2019) for equation (2).

In Fig. 5, DIC (or TA) is the total DIC (or TA) driver and sDIC (or sTA) is the part of DIC (or TA) driver not related to salinity. We are aware that this point is not very clear, and will rewrite Fig. 5 caption.

"**6. Uncertainties**

Please explain in the methods section how the uncertainty of the trends presented in Fig 4 was determined.

**Response**: This will be added in section 2.2 and explained as: "The uncertainty of the trends presented in Fig. 4 was determined in two ways. For the measured parameters (SST, SSS, DIC, TA) the uncertainties represent the standard deviations associated with the means. For the calculated parameters, the uncertainties consider the errors associated with each measurement of the calculation parameters (Orr et al., 2018)."

"**7. Trend 2001-2008 and reconstructed data**

OK, would be worthwhile to mention this"

**Response**: We summarize the last discussion on this subject and we are agreeing to mention this test in the revised manuscript.

[Are Olsen: "I am in particular concerned with the fact that the large summer fCO2 increase in region B 2001-2008 are basically caused by the 'reconstructed' data of 2001 and 2008. The actual observations are pretty steady. How confident are you in these reconstructed values?"

Leseurre et al.: "The large summer fCO2 increase in 2001-2008 is observed in all boxes. Reviewer is correct that starting (2001) and ending (2007) points were reconstructed for July from data obtained in August (for both boxes B and CDE). If we don't standardize the data to July, the "summer" (June+July+August) trend in Box B would be +6.8 µatm/yr (instead of 8.5 when normalizing to July). The same test for box CDE leads to fCO2 increase of +11.2 µatm/yr (instead of 11.5 when normalizing to July). We should note that for August 2007, TA data were doubtful and we used reconstructed TA from salinity for this specific cruise. Overall, we are confident that in 2001-2007 surface fCO2 increased much faster than in the atmosphere."]

This will be rewritten as: "Despite having reconstructed July from the data obtained in August (for boxes B and CDE) both for the starting (2001) and ending (2007) years, this result is fairly robust. Indeed, if we do not normalize summer to July, but directly include the June / July / August data, we also observe a rapid increase in $fCO_2$ of +6.2 µatm yr$^{-1}$."

**2) Responses to new comments about this second version**

Page 1 line 16: 'the trend of …CO2….is slightly less than the atmospheric….and the pH decrease.' Odd sentence, please revise.

**Response**: This will be reformulated as: 'Over the full period (1993-2017) pH decreases (-0.0017 yr$^{-1}$) and fugacity of CO2 (fCO2) increases (+1.70 µatm yr$^{-1}$). The trend of fCO2 in surface water is slightly less than the atmospheric rate (+1.96 µatm yr$^{-1}$).'

Page 1 line 28: Consider using 'multiannual' instead of 'pluriannual'

**Response:** This will be corrected. Thank you!

Page 2 line 3: replace 'indices' with 'values'

**Response:** This will be corrected.

Page 2 line 33-43: The paper by Goris et al: Constraining projection-based estimates of the future North Atlantic carbon uptake, Journal of Climate, 31, 3959-3978, 2018, shows clearly that many ESM struggle to represent mixing correctly in the NASPG, which render them unable to reproduce the actual trends.

**Response**: We will rewrite those lines as: "Such signal is not captured by current ESM CMIP5 models (Tjiputra et al., 2014; Lebehot et al., 2019) likely due to inadequate representation of biogeochemical cycles (here DIC and/or TA) and in part to bad representation of mixing leading to $pCO_2$ seasonality that is opposed to observations (Goris et al., 2018). This leads to uncertainties on the evolution of the oceanic $CO_2$ uptake in the North Atlantic in the future (Lebehot et al., 2019). A better knowledge of DIC and TA trends (not only $fCO_2$) is needed to correct and validate biogeochemical representation in ESM 35 models."

Page 3 line 7-9: It is not so much the shoaling of the lysocline that is the problem, but that these northern surface waters have initially low CaCO3 saturation because of high natural (preindustrial) DIC concentrations because of low temperatures giving high CO2 solubility. Please revise.

**Response:** Thank you for this comment. This will be corrected.

Page 3 line 35: Replace 'conducts' with 'conduits'

**Response**: This will be corrected.

Page 5, line 5: State the pressure used, i.e. 1013.25 hPa.

5   **Response**: This will be added.

Page 5: Please add a subsection on how uncertainties were determined: the uncertainties of the trend slopes, the uncertainties of the driver decomposition (i.e. the 1000 random perturbations) and any other uncertainty calculation. For the uncertainties in the trend regression, how did you deal with the

10   uncertainty of each point in the y direction (e.g. the SST values for each year has a standard deviation associated with them shown as error bars in Fig 4(a), how was this accounted for when calculating the regression and its uncertainty? As far as I know this is not straightforward).

**Response**: We are going to add a comment concerning the uncertainty in each of the following cases: uncertainties of annual means, trends and driver decomposition.

15   The annual means for measured parameters (SST, SSS, DIC and TA; Fig. 4a to 4f) was determined with standard deviation associated with the averaging of the individual data available. For the calculated parameters (pH, $fCO_2$, $\Omega_{Ar}$; Fig. 4d to 4j), the uncertainties consider the errors associated with each measurement of the calculation parameters (Orr et al., 2018). The uncertainty on the trends is linked to the discrete sampling of spatio-temporal variability within the season that we wish to analyze. The

20   uncertainty on the trends is linked either to first, the discrete sampling of a spatial-temporal variability within the season analyzed, or second, to the interannual variability (real or linked to insufficient sampling). If the uncertainty associated with the annual averages is low, then the trend uncertainty will be low and will correspond to the first part. On the other hand, in the case where the interannual (real) variability is high, the uncertainty will correspond to the second part which will decrease the trend

25   significance.

The uncertainty on the driver decomposition, was evaluated by performing 1000 random perturbations within the range of the standard deviation of the observed trends in SST, SSS, TA and DIC.

Page 6 line 5: You used the seasonal climatology to adjust data from January/March and June/August

30   to February and July, not 'to complement winter and summer'. Please state here the typical magnitude of these adjustments.

**Response**: We should have used "to adjust" rather than "to complement". This will be reformulated as "When no data are available in February or July, we adjust the representative winter and summer data based on the deviations from the seasonal cycle observed in January or March (for winter) and June or

35   August (for summer)."

We were thinking about writing an equation in the revised manuscript in order to have a better understanding on how the data were reconstructed. For example, this equation would be:

$$SST_{month(A)}^{year} = SST_{month(A)}^{clim} + ( SST_{month(B)}^{year} - SST_{month(B)}^{clim})$$

Here, month(A) corresponds to February or July, month(B) to January or March (for winter), June or

40   August (for summer) and clim to climatology.

Page 6 line 16-17: Salinity is also low in the green box, but DIC is not correspondingly lower. I.e. the low DIC in the southern (red) box are not only a result of the low salinities, please explain more.

**Response**: That is right. Temperature is also a factor that (indirectly) impacts DIC. In fact, atmospheric $CO_2$ dissolves less in warmer regions. This will be corrected.

Page 6 line 30: 'Section (3.3)' should be 'Section 2.3'

**Response**: This will be corrected.

Page 6: It would be good to include the uncertainty of the trends that are stated.

**Response**: This will be added as: "Using data over 24 years (and all seasons) they estimate an increase in DIC of +0.77 (+/- 0.11) µmol kg$^{-1}$ yr$^{-1}$, a fCO$_2$ trend of +1.95 (+/-0.12) µatm yr$^{-1}$ and a pH decrease of -0.0021 (+/- 0.0001) yr$^{-1}$."

Page 7: Use of TrOCA method. The TrOCA method has issues and it is not very accurate. See for example https://www.biogeosciences.net/7/723/2010/bg-7-723-2010.pdf. Some cautionary remarks would be appropriate, for example that you believe it is robust enough to determine Cant trends even if the absolute values are questionable.

**Response**: We are aware that the TrOCA method, as other data-based methods, does not always reproduce well the absolute values (e.g. Vazquez-Rodriguez et al., 2009 for a comparison including the North Atlantic). However, as stated by Are Olsen, we think that they can be used to estimate the trend: is it 0.1 µmol/kg/yr or 3 µmol/kg/yr? Our calculations suggest a trend of nearly 0.6 µmol/kg/yr at subsurface, an evaluation performed before the publication by Gruber et al (2019). Interestingly, for 1994-2007, we find a trend coherent with Gruber et al for the same region. For this paper, we think that presenting these results (TrOCA and Gruber) is sufficient to separate natural versus anthropogenic signal in our pH and fCO2 trend analysis.

Page 7 line 7: The DIC trend of 0.9 from GLODAPv2 – was this something you calculated. What were the lat/lon bounds.

**Response**:

Yes, we calculate the subsurface (150-200m) trend for both DIC and Cant (with TrOCA) using the same Glodapv2 stations selected in the region 54°N-64°N / 40°W-20°W. We also looked at O$_2$, Nutrients and TALK trends but that were not significant.

Page 7 line 7: GLODAPv2, not GLODAP-V2

**Response**: This will be corrected. Sorry for this mistake.

Page 7 line 16: replace 'very homogeneous' with 'a'

**Response**: This will be corrected.

Page 7 lines 11-18. I don't understand some of these numbers. From Gruber et al., 2019 you find a Cant increase between 1994 and 2007 of 8.5 µmol/kg. This gives a trend of 0.65 µmol/kg yr. You assume this applies for the surface as well, to give a total increase of 10.1 µmol/kg for 1994-2007. But why the change? if the subsurface trend from Gruber applies in the surface layer, this would give 8.5 µmol/kg there as well, not 10.1 µmol/kg.

My sense is that the Cant increase from Gruber et al (0.65 μmol/kg yr-1) is similar to the one you estimate using TrOCA (0.6 μmol/kg yr), but both of these are lower than the observed DIC increase in SURATLANT (0.7-0.9 μmol/kg yr) and estimate from GLODAPv2 (0.9 μmol/kg yr). So altogether, the DIC is increasing slightly faster than explained by Cant alone. Please revise and simplify.

5    **Response**: Thank you to highlight this point. We understand the confusion this raised and we need to reformulate the discussion. The TrOCA method is not suitable to evaluate Cant in surface layers and we can only conclude that the subsurface tend is of +0.6 µmol/kg/yr. We then compared this value with the independent estimate from Gruber (2019) in the same layer and region, who found the very close trend of 0.65 µmol/kg/yr. This similarity is reassuring. Now, because we are interested in comparing the

10    surface trend of DIC from Suratlant with Cant trend at the surface, we will assume that the results from Gruber (2019) are also valid at the surface. In their products, Gruber et al 2019 also offer results for the surface layer and we used their results in the layer 0-50m in our region to evaluate the mean Cant accumulation of 10.1 µmol/kg between 1994 and 2007: here at the surface this leads to Cant trend of +0.8 µmol/kg/yr, i.e. slightly higher than in the layer restricted to 150-200m. The Cant surface trend

15    from Gruber is within the range of DIC trend from Suratlant (range 0.7 to 0.9 µmol/kg/yr for long-term analysis). Overall, these comparisons (TroCA, Gruber, etc..) suggest that Cant explains a significant part of the observed DIC surface trend and similarly for pH (as presented in Figure 6 for long-term 1993-2017).

20    Page 8 line 11: Please simplify to: "Because interannual variability is more pronounced in summer than in winter because of the added influence of biological activity."

**Response**: This will be corrected.

Page 8 line 14-15: Please include citations to these studies that have evaluated trends in different

25    seasons.

**Response**: They will be added as: "Only few observational studies conducted at high latitudes (Olafsson et al., 2009; Munro et al., 2015; Wakita et al 2017) showed that the trends of DIC, fCO2 and pH are seasonally different and thus driven by different processes not yet fully explained."

30    Page 9 line 16: Mention the source of these independent $pCO_2$ observations (ship, principal investigators, cite any papers by PIs that describe the data).

**Response**: The $pCO_2$ observations we refer to in this section were listed in Metzl et al (2010). We don't think it is useful to recall them here.

35    Page 10 line 14: 'last decade, 2001-2017…', better with 'last period, 2008-2017 for summer and 2008-2015 for winter'.

**Response**: This will be corrected.

Page 10 line 29-30: The case for low primary production in 1996, is very weak. 1996 was characterized

40    by very negative NAO index. Another year with very low NAO index was 2010, and in that year primary production was exceptionally high. So, if anything – by analogy – I would expect strong primary production in 1996.

**Response**:

Yes, you are right. We have attempted to test this high productivity assumption, but we lack data for 1996. Indeed, we have no SURATLANT nutrient data for summer 1996; and there are no satellite observations before 1998. We also tried to check the high DIC and fCO2 for 1996 with GLODAPv2 and SOCAT but there are no data available for this period to compare with SURATLANT in the same locations. On the other hand, a study by Martinez et al (2016) based on CPR data and focused in the NASPG (50°N-60°N) suggests negative anomalies of Chl-a around 1996-1997. However, we notice that the regions selected by these authors are not exactly those we are selecting (our Boxes) and it is difficult to clearly identify if Chl-a decreased in 1996. In order to confirm the results presented by Martinez et al, we recently requested (22/3/20) Phytoplankton Color Index (PCI) data from CPR (monthly mean in CPR boxes) in the Suratlant regions (https://www.cprsurvey.org/data/our-data/). As shown in Figure R1, one observes low PCI in 1996 compared to several highs PCI recorded in 1994-1995; that would suggest less production occurring in 1996 and might explain the higher DIC concentrations discussed in our paper. Such preliminary view based on monthly mean PCI calls for deeper investigations using original CPR data but beyond the scope of our present analysis; this would require to reprocess the original CPR data along the Suratlant lines (not only monthly mean products as presented in Fig R1), an analysis to share with colleagues familiar with CPR data to interpret the results.

Regarding your comment, we will add the reference to Martinez et al and change the text following:

"We notice that in 1993-1996 NAO changed from a positive to negative phase and AMO progressively increased in the nineties (Fig. 3c) but no particular anomalies were revealed in the winter observations. Thus, we have no straight explanation for the fast DIC, fCO2 and pH changes observed during the summers 1993-1996, except that relatively higher DIC in 1996 might have resulted from a decrease in primary production compared with to previous years. This is rather speculative as we have no other information on nutrients o or on the biological activity (e.g. no nutrient data for the summer 1996 and not input from remote sensing data at the beginning of the nineties). However, using CPR data over 1960-2010 in both western and eastern NASPG regions, Martinez et al. (2016) identified negative anomalies of Chl-a around 1996-1997 after a sharp increase of Chl-a during over 1986–1995."

Page 11 line 2-4: Attribution of summer 2005 - 2007 high fCO2 values. Low productivity and/or deep (winter) mixing suggested. This can and should be checked with satellite Chl and Argo MLD data. For the record, Fröb et al., 2016 (Irminger Sea deep convection injects oxygen and anthropogenic carbon to the ocean interior, Nature Communications, 7:13244, 2016) shows that winter mixing was not deep in these years in the central-west Irminger Sea. Therefore, primary production is the more likely candidate.

**Response**: It is true that the amount of chlorophyll-a seen by satellite is less important over the period 2004-2007 in the northern region of the SURATLANT transect. This is shown on Figure R2 during the period 2002-2010. North of 58°N, the chlorophyll-a content is lower over the 2004-2007 period than after 2007.

Page 11, line 18-20. This is cherry picking, what about the earlier time periods when the discrete data does not reflect the trends in higher frequency TSG data (such as winter 1993-1998 noted earlier in this review)? The extent to which these discrete samples reflect larger scale variations should be investigated using a systematic approach – not an ad hoc as is currently done.

**Response**: This will be corrected as: "During 2008-2017, the progressive cooling in the NASPG, found here in both winter and summer data (Fig. 4a) is a large-scale signal (Fig. 3a, Robson et al., 2016; Reverdin et al., 2018a)".

45

Page 11 line 31. Write as GLODAPv2.2019

**Response**: This will be corrected.

Page 12 line 32-33. This is also related to the way models do/don't represent winter mixing, see earlier cited Goris et al paper.

**Response**: We will take it into account.

Page 13 line 10. Last sentence needs revision.

**Response**: "However, the length of this period is short to interpret trends highly sensitive to the starting and ending years" will be revised as: "However, the length of the period is short and trend results are very sensitive to interannual anomalies, especially in the first and last years".

Page 14 lines 1-10. The 2010 primary production event is nicely discussed in Henson et al, Unusual subpolar North Atlantic phytoplankton bloom in 2010, JGR, 2013.

**Response**: Thank you. This will be added as "Indeed, Henson et al. (2013) showed that the physical forcing caused by the very negative NAO recorded in winter 2009-2010 stimulated spring blooms and not the eruption of the volcano Eyjafjallajökull in Iceland which erupted in spring 2010, depositing large amounts of iron in the North Atlantic Subpolar."

Page 14 line 23-24. Please name ship (Skogafoss) and principal investigator (Wanninkhof NOAA/AOML) of these data (it is always nice to give specific and concrete credit to the correct people)

**Response**: We agree. This will be added.

We fully agree that it is important to credit data providers, which is not always the case in many papers, especially when using SOCAT data for global investigations. Note that these cruises were specifically used by Reverdin et al. (2018) to compare calculated and measured fCO2 (see Fig A1 in Reverdin 2018), but we did not recall these results in the present manuscript and just refer to expocodes.

Page 14-15, "Such large changes in pH trends … would not have been resolved when using fCO2 data and TA/S relation instead of TA measurements". I am not convinced this is actually the case. pH determined from fCO2 and TA is not very sensitive to TA, but depends most on fCO2, which in itself reflects underlying DIC/TA changes; pH and fCO2 are strongly correlated. I think such large changes could be visible in pH determined from fCO2 and TA (from TA-S relationship). You may want to check with a simple calculation before making such a statement. (i.e. combine the calculated fCO2 with TA from TA-S relationship and calculate pH trends.)

**Response**: We were referring to calculating pH from observations of $fCO_2$ and TA reconstructed from salinity. Obviously, if we couple calculated $fCO_2$ (from measured TA and DIC) and reconstructed TA, the changes in pH observed with the TA / DIC pair will also be observed (see Table R1). Unfortunately, we don't use observations of oceanic $fCO_2$ on SURATLANT. Thus, we choose to delete this sentence in the conclusion.

Page 15, line 11 You state 'indirect methods' but refer to only one, Denvil-Sommer et al., 2019. This generalization does not seem correct unless you compare with other mapped pCO2/flux products.

**Response**: The figure S5h in Denvil-Sommer et al., 2019 compares several (four) indirect methods.

**Reference added:**

Goris, N., Tjiputra, J. F., Olsen, A., Schwinger, J., Lauvset, S. K. and Jeansson, E.: Constraining projection-based estimates of the future North Atlantic carbon uptake, Journal of Climate, 31, 3959-3978, doi:10.1175/JCLI-D-17-0564.1, 2018.

Henson, S. A., Painter, S. C., Holliday, N. P., Stinchcombe, M. C., and Giering, S. L. C.: Unusual subpolar NorthAtlantic phytoplankton bloom in 2010: Volcanic fertilization or North Atlantic Oscillation?, J. Geophys. Res. Oceans, 118, 4771–4780, doi:10.1002/jgrc.20363, 2013.

Martinez, E., Raitsos, D. E., and Antoine, D.: Warmer, deeper, and greener mixed layers in the North Atlantic subpolar gyre over the last 50 years. Glob. Chang. Biol. 22, 604–612. doi:10.1111/gcb.13100, 2016.

Munro, D. R., N. S. Lovenduski, T. Takahashi, B. B. Stephens, T. Newberger, and C. Sweeney.: Recent evidence for a strengthening CO2 sink in the Southern Ocean from carbonate system measurements in the Drake Passage (2002–2015), Geophys. Res. Lett., 42, doi:10.1002/2015GL065194, 2015.

[Figure]

**Figure S1**. Difference of SST (a,b) and SSS (c,d) between Suratlant discrete data and data based on the Binned products constructed by Reverdin et al. (2018a) and updated data available at https://doi.org/10.6096/SSS-BIN-NASG, last access 25/03/2020. Data obtained between 1993 and 2017 during summer (in red, a-c) and 1994 and 2015 during winter (in blue, b-d) in box B (open symbols) and by combining all data in boxes C, D, E (filled symbols). Data from February and July are indicated with circles and the reconstructed data are depicted with triangles.

[Figure]

**Figure S2-1**. Regional SST trends maps from NOAA OI.v2 SST products available in ASCII at http://rda.ucar.edu/datasets/ds277.0/, last access 6/03/2020; for the three periods during summer, corresponding to July for a), b), c) and mean of June, July, August for d), e), f).

[Figure]

**Figure S2-2**. Regional SST trends maps from NOAA OI.v2 SST products available in ASCII at http://rda.ucar.edu/datasets/ds277.0/, last access 6/03/2020; for the three periods during winter, corresponding to February for a), b), c) and mean of January, February, March for d), e), f).

**Table S1.** SST and SSS trends based on Suratlant data (Table 2) and on the Binned products constructed by Reverdin et al. (2018a) available at https://doi.org/10.6096/SSS-BIN-NASG, last access 25/03/2020. In bold are represented the significant trends (Student test).

| period | season | box | SST °C (yr$^{-1}$) | | SSS (yr$^{-1}$) | |
|---|---|---|---|---|---|---|
| | | | Suratlant | Binned | Suratlant | Binned |
| 1993-2017 | summer | B | **0.05 ± 0.00** | 0.01 ± 0.02 | **-0.003 ± 0.000** | -0.003 ± 0.005 |
| | | C.D.E | **0.03 ± 0.00** | 0.04 ± 0.02 | **0.003 ± 0.000** | 0.001 ± 0.003 |
| | winter | B | **0.02 ± 0.00** | -0.02 ± 0.02 | **0.003 ± 0.000** | -0.003 ± 0.005 |
| | | C.D.E | **0.01 ± 0.00** | 0.01 ± 0.02 | **0.005 ± 0.000** | **0.005 ± 0.002** |
| 1993-1997 | summer | B | **-0.09 ± 0.02** | 0.16 ± 0.36 | **0.042 ± 0.001** | -0.025 ± 0.083 |
| | | C.D.E | **0.60 ± 0.01** | **0.77 ± 0.19** | **0.004 ± 0.001** | -0.012 ± 0.012 |
| | winter | B | **-0.19 ± 0.02** | 0.15 ± 0.29 | **0.045 ± 0.001** | 0.024 ± 0.063 |
| | | C.D.E | 0.00 ± 0.01 | 0.24 ± 0.09 | **-0.018 ± 0.001** | 0.001 ± 0.040 |
| 2001-2007 | summer | B | **0.04 ± 0.01** | 0.04 ± 0.07 | **-0.027 ± 0.000** | -0.017 ± 0.013 |
| | | C. D. E | -0.01 ± 0.00 | 0.07 ± 0.08 | **0.013 ± 0.000** | 0.005 ± 0.013 |
| | winter | B | 0.04 ± 0.01 | 0.10 ± 0.05 | **-0.019 ± 0.001** | -0.013 ± 0.022 |
| | | C. D. E | **-0.05 ± 0.01** | 0.10 ± 0.07 | -0.002 ± 0.000 | 0.013 ± 0.008 |
| 2008-2017 | summer | B | **-0.09 ± 0.00** | 0.01 ± 0.06 | **-0.013 ± 0.000** | -0.001 ± 0.018 |
| | | C. D. E | **-0.11 ± 0.00** | -0.11 ± 0.06 | **-0.013 ± 0.000** | **-0.018 ± 0.008** |
| | winter | B | 0.02 ± 0.01 | -0.03 ± 0.07 | **-0.004 ± 0.000** | -0.028 ± 0.018 |
| | | C. D. E | **-0.05 ± 0.00** | -0.11 ± 0.05 | **-0.006 ± 0.000** | -0.007 ± 0.005 |

[Figure]

[Figure]

**Figure R1**. (a) Map of the North Atlantic regionalization of CPR data available at https://www.cprsurvey.org/data/our-data/, last access 20/03/2020. (b) Monthly mean Phytoplankton Color Index (PCI) for boxes B6, B7, C6, C7, D6, D7, D8 over the period 1991-2000.

[Figure]

[Figure]

**Figure R2**. Variability of chlorophyll-a seen by satellite over the period 2002-2010 for the Suratlant tracks. Data used: dataset-oc-glo-bio-multi-l4-chl_4km_monthly-rep, available on http://marine.copernicus.eu, last access 13/03/2020.

**Table R1.** pH trends (per year) calculated by two ways: from TA and DIC measurements or from TA reconstructed with measured salinity and calculated fCO$_2$ (from TA and DIC). 
[revised manuscript text omitted]